# Training shallow ReLU networks on noisy data using hinge loss: when do we overfit and is it benign?

*Erin George, *Michael Murray, William Swartworth, Deanna Needell
Department of Mathematics, UCLA, CA, USA
[egeo,mmurray,wswartworth,deanna]@math.ucla.edu
*Equal contribution

## Abstract

We study benign overfitting in two-layer ReLU networks trained using gradient descent and hinge loss on noisy data for binary classification. In particular, we consider linearly separable data for which a relatively small proportion of labels are corrupted or flipped. We identify conditions on the margin of the clean data that give rise to three distinct training outcomes: benign overfitting, in which zero loss is achieved and with high probability test data is classified correctly; non-benign overfitting, in which zero loss is achieved but test data is misclassified with probability lower bounded by a constant; and non-overfitting, in which clean points, but not corrupt points, achieve zero loss and again with high probability test data is classified correctly. Our analysis provides a fine-grained description of the dynamics of neurons throughout training and reveals two distinct phases: in the first phase clean points achieve close to zero loss, in the second phase clean points oscillate on the boundary of zero loss while corrupt points either converge towards zero loss or are eventually zeroed by the network. We prove these results using a combinatorial approach that involves bounding the number of clean versus corrupt updates across these phases of training.

## 1 Introduction

Conventional machine learning wisdom suggests that the generalization error of a complex model will typically be worse versus a simpler model when both are trained to interpolate data. Indeed, the bias-variance trade-off implies that although choosing a complex model is advantageous in terms of approximation error, it comes at the price of an increased risk of overfitting. The traditional solution to managing this trade-off is to use some form of regularization, allowing the optimizer to select a predictor from a rich class of functions while at the same time encouraging it to choose one that is in some sense simple. However, in recent years this perspective has been challenged by the observation that deep learning models, trained with minimal if any form of regularization, can almost perfectly interpolate noisy data with nominal cost to their generalization performance (Zhang et al., 2017; Belkin et al., 2018b, 2019). This phenomenon is referred to as *benign overfitting*.

Following these empirical observations, a line of research has emerged aiming to theoretically characterize the conditions under which various machine learning models, trained to zero loss on noisy data, obtain, at least asymptotically, optimal generalization error. To date, the majority of analyses in this regard have focused primarily on linear models, including linear regression (Bartlett et al., 2020; Muthukumar et al., 2020; Wu & Xu, 2020; Chatterji & Long, 2021; Zou et al., 2021; Hastie et al., 2022; Koehler et al., 2021; Wang et al., 2021a; Chatterji & Long, 2022; Cao et al., 2021; Shamir, 2022), logistic regression (Chatterji & Long, 2021; Muthukumar et al., 2021; Wang et al., 2021b) and kernel regression (Belkin et al., 2018a; Mei & Montanari, 2019; Liang & Rakhlin, 2020; Liang et al., 2019). With regards to understanding benign overfitting in neural networks, in the

Neural Tangent Kernel (NTK) regime (Jacot et al., 2018) the prediction of a neural network is well approximated via kernel regression (Adlam & Pennington, 2020). However, this regime typically requires unrealistically large network width and fails to capture feature learning. Indeed, and despite being the initial source of inspiration, an understanding of when and how neural networks benignly overfit in the rich, feature learning regime is not well understood.

## 1.1 Contributions and related work

In this work we study benign overfitting in the context of binary classification for two-layer ReLU networks, trained using gradient descent and hinge loss, on label corrupted, linearly separable data. There are a number of recent and or concurrent works which prove benign overfitting results in a similar setting Frei et al. (2022, 2023); Xu & Gu (2023); Cao et al. (2022); Kou et al. (2023); Kornowski et al. (2023), however, we emphasize that these exclusively study exponentially tailed losses, notably the popular logistic loss. Benign overfitting is intimately related to the notion of *implicit bias*, the preference of an algorithm for selecting minimizers with certain properties over others. The implicit bias of homogeneous networks trained with gradient descent on an exponentially tailed loss from a low initial loss is known to converge in direction to a Karush-Kuhn-Tucker (KKT) point of the associated max-margin problem Lyu & Li (2020); Ji & Telgarsky (2020). This implies at least intuitively a certain bias towards margin maximization. In a recent work Frei et al. (2023) it is shown that if the input data is sufficiently orthogonal then a shallow, leaky ReLU network evaluated on such a KKT point is equivalent to a particular linear classifier. Moreover, and under additional data assumptions, the authors show such networks benignly overfit. Another recent paper Kornowski et al. (2023) uses a similar approach to derive benign overfitting results for ReLU networks and also provides a description of the transition between benign and tempered overfitting in the univariate input case. To the best of our knowledge, equivalent results on the implicit bias of homogeneous networks trained with non-exponentially tailed losses are not characterized. Furthermore, training a linear classifier with an exponential versus non-exponential tailed loss is known to result in a different implicit bias, with the non-exponential tailed loss potentially inducing convergence in direction to a classifier with a poor margin Ji et al. (2020). As a result, a priori it is not clear if and how the choice of hinge loss impacts the propensity for a shallow ReLU network to overfit.

There are two main existing lines of work which study benign overfitting in neural networks outside of the kernel regime. Concerning perhaps the most relevant line of prior work to our own, Frei et al. (2022) consider a smooth, leaky ReLU activation function, train the network using the logistic instead of the hinge loss and assume the data is drawn from a mixture of well-separated sub-Gaussian distributions. The key result of this work is that given a sufficient number of iterations of GD, then the network will interpolate the noisy training data while also achieving minimax optimal generalization error up to constants in the exponents. A concurrent work Xu & Gu (2023) extends this result to more general activation functions including ReLU, relaxes the assumptions on the noise distribution to being centered with bounded logarithmic Sobolev constant, and also improves the convergence rate. As highlighted in Xu & Gu (2023), the fact that ReLU is non-smooth and non-leaky significantly complicates the analysis of both the convergence and generalization. A second line of work (Cao et al., 2022; Kou et al., 2023) studies benign overfitting in two-layer convolutional as opposed to feedforward neural networks. Whereas here and in Frei et al. (2022); Xu & Gu (2023) each data point is modeled as the sum of a signal and noise component, in Cao et al. (2022); Kou et al. (2023) the signal and noise components lie in disjoint patches. The weight vector of each neuron is applied to both patches separately and a non-linearity, such as ReLU, is applied to the resulting pre-activation. In this setting, the authors prove interpolation of the noisy training data and derive conditions on the clean margin under which the network benignly vs non-benignly overfits. We emphasize that the data model studied in this work is very different to the setting we study here, and as a result we primarily restrict our comparison to that with Frei et al. (2022) and the concurrent work Xu & Gu (2023). Finally, in regard to optimizing shallow ReLU networks using hinge loss, a line of work (Brutzkus et al., 2018; Wang et al., 2019; Yang et al., 2021) studies the convergence of gradient descent on generic, linearly separable data without label corruptions. These works also require additional assumptions, notably leaky ReLU instead of ReLU, insertion of noise into the optimization algorithm or changes to the loss function.

Before we discuss our contributions we remark that a previous work Mallinar et al. (2022) describes and experimentally explores a taxonomy of overfitting: benign overfitting, where the generalization error is optimal; catastrophic overfitting, where the generalization error is close to random chance;

and tempered overfitting, which lies in between. In this work, we do not consider the full breadth of this taxonomy, and use the terms "non-benign overfitting" or equivalently "harmful overfitting" to refer to overfitting that may be either tempered or catastrophic. We now summarize our contributions: in particular, under certain assumptions on the model hyperparameters, we prove conditions on the clean margin resulting in the three distinct training outcomes highlighted below. We remark also that the prior works discussed primarily focus on deriving positive benign overfitting results.

1. **Benign overfitting:** Theorem 3.1 provides conditions under which the training loss converges to zero and bounds the generalization error, showing that it is asymptotically optimal. This result is analogous to those of Frei et al. (2022) and Xu & Gu (2023) but for the hinge instead of logistic loss.

2. **Non-benign overfitting:** Theorem 3.6 provides conditions under which the network achieves zero training loss while generalization error is bounded below by a constant. Unlike Frei et al. (2022) and Xu & Gu (2023), this is not due to the non-separability of the data model but is instead a result of the neural network failing to learn the optimal classifier.

3. **No overfitting:** Theorem 3.8 provides conditions under which the network achieves zero training loss on points with uncorrupted label signs but nonzero loss on points with corrupted signs. Again the generalization error is bounded and shown to be asymptotically optimal.

To conclude this section we further remark that our proof techniques are quite different from those used in Frei et al. (2022); Xu & Gu (2023) and indeed the other works highlighted in this section. Again we emphasize this is due to the fact we study the hinge loss instead of the logistic loss and discuss the differences arising from this in detail in Section 3. In particular, we set up the problem in such a way that the convergence analysis reduces to counting the number of activations of clean versus corrupt points during various stages of training. Our analysis further provides a detailed description of the dynamics of the network's neurons, thereby allowing us to understand how the network fits both the clean and corrupted data.

## 2 Preliminaries

### 2.1 Data model

We consider a training sample of $2n$ pairs of points and their labels $(\mathbf{x}_i, y_i)_{i=1}^{2n}$ where $(\mathbf{x}_i, y_i) \in \mathbb{R}^d \times \{-1, +1\}$ for all $i \in [2n]$. Furthermore, we identify two disjoint subsets $\mathcal{S}_T \subset [2n] = \{1, \ldots, 2n\}$ and $\mathcal{S}_F \subset [2n]$, $\mathcal{S}_T \cup \mathcal{S}_F = [2n]$, which correspond to the clean and corrupt points in the sample respectively. The categorization of a point as clean or corrupted is determined by its label: for all $i \in [2n]$ we assume $y_i = \beta(i)(-1)^i$ where $\beta(i) = -1$ iff $i \in \mathcal{S}_F$ and $\beta(i) = 1$ otherwise. In addition, we assume $|\mathcal{S}_F \cap [2n]_e| = |\mathcal{S}_F \cap [2n]_o| = k$ and $|\mathcal{S}_T \cap [2n]_e| = |\mathcal{S}_T \cap [2n]_o| = n - k$, where $[2n]_e \subset [2n]$ and $[2n]_o \subset [2n]$ are the even and odd indices, respectively. We remark that this assumption simplifies the exposition of our results but is not integral to our analysis. Each data point is assumed to have the form

$$\mathbf{x}_i = (-1)^i(\sqrt{\gamma}\mathbf{v} + \sqrt{1-\gamma}\beta(i)\mathbf{n}_i). \tag{1}$$

Here $\mathbf{v} \in \mathbb{R}^d$ satisfies $\|\mathbf{v}\| = 1$ and furthermore we refer to $\mathbf{v}$ as the signal vector as the alignment of a clean point with $\mathbf{v}$ determines its sign. Indeed, $\text{sign}(\langle \mathbf{x}_i, \mathbf{v} \rangle) = (-1)^i = y_i$ for $i \in \mathcal{S}_T$ whereas $\text{sign}(\langle \mathbf{x}_i, \mathbf{v} \rangle) = -y_i$ for $i \in \mathcal{S}_F$. Thus we may view the labels of a corrupt point as flipped from their clean state. The vectors $(\mathbf{n}_i)_{i=1}^{2n}$ are mutually independent and identically distributed (i.i.d.) random vectors drawn from the uniform distribution over $\mathbb{S}^{d-1} \cap \text{span}\{\mathbf{v}\}^\perp$, which we denote $U(\mathbb{S}^{d-1} \cap \text{span}\{\mathbf{v}\}^\perp)$. Clearly this distribution is symmetric, mean zero and for any $\mathbf{n} \sim U(\mathbb{S}^{d-1} \cap \text{span}\{\mathbf{v}\}^\perp)$ it holds that $\mathbf{n} \perp \mathbf{v}$ and $\|\mathbf{n}\| = 1$. We refer to these vectors as noise components due to the fact that they are independent of the labels of their respective points. The real, scalar quantity $\gamma \in [0, 1]$ controls the strength of the signal versus the noise and also defines the clean margin. Finally, at test time a clean label $y \sim U(\{-1, 1\})$ is sampled and the corresponding test data point is constructed,

$$\mathbf{x} = y(\sqrt{\gamma}\mathbf{v} + \sqrt{1-\gamma}\mathbf{n}), \tag{2}$$

where again $\mathbf{n} \sim U(\mathbb{S}^{d-1} \cap \text{span}\{\mathbf{v}\}^\perp)$.

The key idea we use to characterize the training dynamics is to reduce the analysis of the trajectory of each neuron to that of counting the number of clean versus corrupt updates to it. This combinatorial approach relies on each point having similar sized signal and noise components. In order to make our analysis as clear as possible, we select a data model which ensures the signal and noise components are consistent in size across all points. We emphasize that these assumptions are not strictly necessary and we believe analogous analyses could be conducted when the signal and noise components are instead appropriately bounded. In addition, and as discussed in more detail in Section 3.2, the orthogonality of the signal and noise components allow us to demonstrate non-benign overfitting even when a perfect classifier exists.

## 2.2 Network architecture, optimization and initialization

We consider a densely connected, single layer feed-forward neural network $f : \mathbb{R}^{2m \times d} \times \mathbb{R}^d \to \mathbb{R}$ with the following forward pass map,

$$f(\mathbf{W}, \mathbf{x}) = \sum_{j=1}^{2m} (-1)^j \phi(\langle \mathbf{w}_j, \mathbf{x} \rangle).$$

Here $\phi := \max\{0, z\}$ denotes the ReLU activation function and $\mathbf{w}_j$ the $j$-th row of the weight matrix $\mathbf{W} \in \mathbb{R}^{2m \times d}$. The network weights are optimized using full batch gradient descent (GD) with step size $\eta > 0$ in order to minimize the hinge loss over a training sample $((\mathbf{x}_i, y_i))_{i=1}^{2n} \subset (\mathbb{R}^d \times \{-1, 1\})^{2n}$ sampled as described in Section 2.1. After $t'$ iterations this optimization process generates a sequence of weight matrices $(\mathbf{W}^{(t)})_{t=0}^{t'}$. For convenience, we overload our notation for the forward pass map of the network and let $f(t, \mathbf{x}) := f(\mathbf{W}^{(t)}, \mathbf{x})$. Furthermore, we denote the hinge loss on the $i$-th point at iteration $t$ as $\ell(t, i) := \max\{0, 1 - y_i f(t, \mathbf{x}_i)\}$. The hinge loss over the entire training sample at iteration $t$ is therefore $L(t) := \sum_{i=1}^{2n} \ell(t, i)$. Let $\mathcal{F}^{(t)} := \{i \in [2n] : \ell(t, \mathbf{x}_i) > 0\}$ and $\mathcal{A}_j^{(t)} := \{i \in [2n] : \langle \mathbf{w}_j^{(t)}, \mathbf{x}_i \rangle > 0\}$ denote the sets of point indices that have nonzero loss and which activate the $j$th neuron at iteration $t$ respectively. With

$$\frac{\partial \ell(t, i)}{\partial w_{jr}} = \begin{cases} 0, & \langle \mathbf{w}_j^{(t)}, \mathbf{x}_i \rangle \leq 0, \\ -(-1)^j y_i x_{ir}, & \langle \mathbf{w}_j^{(t)}, \mathbf{x}_i \rangle > 0 \end{cases}$$

then the GD update rule[1] for the neuron weights at iteration $t \geq 0$ may be written as

$$\mathbf{w}_j^{(t+1)} = \mathbf{w}_j^{(t)} + (-1)^j \eta \sum_{l=1}^{2n} \mathbb{1}(l \in \mathcal{A}_j^{(t)} \cap \mathcal{F}^{(t)}) y_l \mathbf{x}_l. \tag{3}$$

In regard to the initialization of the network parameters, for convenience we assume each neuron's weight vector is drawn mutually i.i.d. uniform from the centered sphere with radius $\lambda_w > 0$. We remark that results analogous to the ones presented hold if the weights are instead initialized mutually i.i.d. as $w_{jc}^{(0)} \sim \mathcal{N}(0, \sigma_w^2)$ for sufficiently small $\sigma_w^2$.

## 2.3 Notation

For indices $i, j \in \mathbb{Z}_{\geq 1}$ we say $i \sim j$ iff $(-1)^i = (-1)^j$. We often refer to a data point or neuron by its index alone, e.g. "point $i$" refers to the $i$-th training point $(\mathbf{x}_i, y_i)$. For two iterations $t_0, t_1$ with $t_1 > t_0$ we define the following.

1. $G_j(t_0, t_1) := \sum_{i \in \mathcal{S}_T} \sum_{\tau=t_0}^{t_1-1} \mathbb{1}(i \in \mathcal{A}_j^{(\tau)} \cap \mathcal{F}^{(\tau)})$ is the number of clean updates applied to the $j$-th neuron between iterations $t_0$ and $t_1$.

2. $B_j(t_0, t_1) := \sum_{i \in \mathcal{S}_F} \sum_{\tau=t_0}^{t_1-1} \mathbb{1}(i \in \mathcal{A}_j^{(\tau)} \cap \mathcal{F}^{(\tau)})$ is the number of corrupt updates applied to the $j$-th neuron between iterations $t_0$ and $t_1$.

3. $G(t_0, t_1) := \sum_{j \in [2m]} G_j(t_0, t_1)$ and $B(t_0, t_1) := \sum_{j \in [2m]} B_j(t_0, t_1)$ are the total number of clean and corrupt updates applied to the entire network between iterations $t_0$ and $t_1$.

---

[1] Although the derivative of ReLU clearly does not exist at zero, we follow the routine procedure of defining an update rule that extends the gradient update to cover this event.

4. $T(t_0, t_1) := G(t_0, t_1) + B(t_0, t_1)$ is the total number of updates from all points applied to the entire network between iterations $t_0$ and $t_1$.

We extend all these definitions to the case $t_0 = t_1$ by letting the empty sum be 0. Finally, we use $C \geq 1$ and $c \leq 1$ to denote generic, positive constants.

## 3 Results

The main contributions of this work are Theorem 3.1, Theorem 3.6 and Theorem 3.8, which characterize how the margin of the clean data drives three different training regimes: namely benign overfitting, non-benign (or harmful) overfitting and no-overfitting respectively. We primarily distinguish between the three aforementioned training outcomes based on conditions on the signal strength $\gamma \in [0, 1]$ which controls the clean margin. Assuming the corrupt points are the minority in the training sample, then heuristically we might expect the following behavior as $\gamma$ varies: if $n\gamma \gg 1$, then the signal dominates the noise during training, corrupted points are never fitted and the network generalizes well. If $n\gamma \ll 1$, then all points are eventually fitted based on their noise component and the network generalizes poorly. As such, we expect to observe benign overfitting when $\gamma$ is small but not too small: in this regime the network learns the signal, thus ensuring it generalizes well, but corrupted points can still be fitted based on their noise component, thereby allowing training to zero loss.

With each theorem we provide here we give a sketch of its proof: full proofs are contained in the Supplementary Materials, which also contain supporting numerical simulations in Appendix F. Throughout this section, and in order to establish a common setting in which to observe a variety of different behaviors, we make the following assumptions on the network and data hyperparameters.

**Assumption 1.** *For a sufficiently large constant $C \geq 1$, failure probability $\delta \in (0, 1/2)$ and noise inner product bound $\rho \in (0, 1)$, let $d \geq C\rho^{-2}\log(n/\delta)$, $k \leq cn$, $\lambda_w \leq c\eta$ and $\eta \leq \xi$, where $\xi$ depends on $n$, $m$, $k$, $\gamma$, and $d$.*

We remark that the condition $d \geq C\rho^{-2}\log(n/\delta)$ ensures the noise components are nearly-orthogonal: in particular, $\max_{i \neq \ell} |\langle \mathbf{n}_i, \mathbf{n}_\ell \rangle| \leq c\rho$ with high probability for some positive constant $c$. This near orthogonality condition on the noise terms is restrictive, but is a common assumption in the related works Frei et al. (2022); Xu & Gu (2023). We note that the value of $\rho$ required for each of our results to hold varies. Likewise, the optimal constants $c$ and $C$ required in each case also vary and we will not concern ourselves with finding the tightest possible constants.

While there are differences the proofs of Theorem 3.1, 3.6 and 3.8 generally fit the following outline.

1. Use concentration to show with high probability the training data is nearly orthogonal and a certain initialization pattern is satisfied.

2. Characterize the activation pattern early in training before any point achieves zero loss.

3. Bound the activations at an iteration just before any training point achieves zero loss.

4. Based on bounds on the activations at a given iteration, derive an iteration-independent upper bound on the number of subsequent updates that can occur before convergence. At convergence all points either have zero loss or activate no neurons.

We emphasize that our proof techniques are significantly different from those used in Frei et al. (2022); Xu & Gu (2023) due to the differences between the hinge and logistic loss. In particular, letting $\sigma(z)$ denote the logistic loss, a key step in the proof of these prior works is showing at any iteration $t \geq 0$ that the ratio $\sigma'(y_i f(t, \mathbf{x}_i))/\sigma'(y_l f(t, \mathbf{x}_l))$ is upper bounded by a constant for all pairs of points $i, l$ in the training sample. For the hinge loss this approach is not feasible: indeed, if at an iteration $t$ some points achieve zero loss while others have not then this ratio is unbounded.

### 3.1 Benign overfitting

The following theorem states conditions in particular on $\gamma$ under which the network simultaneously achieves asymptotically optimal test error and achieves zero loss on both the clean and corrupted data after a finite number of iterations. A detailed proof of this Theorem along with the associated lemmas is provided in Appendix C.

**Theorem 3.1.** *Let Assumption 1 hold and further assume $n \geq C \log(1/\delta)$, $m \geq C \log(n/\delta)$, $\rho \leq c\gamma$ and $C\sqrt{\log(n/\delta)/d} \leq \gamma \leq cn^{-1}$. Then there exists a sufficiently small step-size $\eta$ such that with probability at least $1 - \delta$ over the randomness of the dataset and network initialization the following hold.*

1. *The training process terminates at an iteration $\mathcal{T}_{end} \leq \frac{Cn}{\eta}$.*

2. *For all $i \in [2n]$ then $\ell(\mathcal{T}_{end}, \mathbf{x}_i) = 0$.*

3. *The generalization error satisfies*

$$\mathbb{P}(\mathrm{sgn}(f(\mathcal{T}_{end}, \mathbf{x})) \neq y) \leq \exp\left(-cd\gamma^2\right).$$

*Proof sketch.* Recall the parameter $\rho$ bounds the inner products of the noise components of the training data. Specifically, the conditions on $d$ given in Assumption 1 ensure $\max_{i \neq l} |\langle \mathbf{n}_i, \mathbf{n}_l \rangle| \leq \frac{\rho}{1-\gamma}$ with high probability. We also identify the following sets of neurons for $p \in \{-1, 1\}$,

$$\Gamma_p := \left\{ j \in [2m] \ : \ (-1)^j = p, \ G_j(0,1)(\gamma - \rho) - B_j(0,1)(\gamma + \rho) \geq \frac{2\lambda_w}{\eta} \right\},$$

$$\Theta_p := \{ j \in \Gamma_p \ : \ G_j(0,1)(\gamma + \rho) - B_j(0,1)(\gamma - \rho) \leq 1 - \gamma + \rho \}.$$

These sets are useful in that neurons in $\Gamma_p$ have predictable activation patterns during the early phase of training. Furthermore, if $i$ is the index of a corrupted point which activates a neuron in $\Theta_{y_i}$ at initialization, then this point will continue activating this neuron throughout the early phase of training. Concentration argument shows that $\Gamma_p$ and $\Theta_p$ are sufficiently significant subsets of $[2m]_p$[2] with high probability. In summary, for benign overfitting we say we have a *good initialization* if i) $\max_{i \neq l} |\langle \mathbf{n}_i, \mathbf{n}_l \rangle| \leq \frac{\rho}{1-\gamma}$, ii) for some small constant $\alpha \in (0,1)$ then $|\Gamma_p| \geq (1 - \alpha)m$ for $p \in \{-1, 1\}$, and iii) for each $i \in \mathcal{S}_F$ there exists a $j \in [2m]$ such that $(-1)^j = y_i$ and $i \in \mathcal{A}_j^{(0)}$.

**Lemma 3.2.** *Under the assumptions of Theorem 3.1 and assuming we have a good initialization, suppose at some iteration $t_0$ the loss of every clean point is bounded above by $a \in \mathbb{R}_{\geq 0}$, while the loss of every corrupted point is bounded above by $b \in \mathbb{R}_{\geq 0}$. Then for all $t \geq t_0$ the total number of clean and corrupt updates which occur after $t_0$ are upper bounded as follows,*

$$G(t_0, t) \leq Cn\left(\frac{a + bk\gamma}{\eta}\right), \qquad\qquad B(t_0, t) \leq Ck\left(\frac{b + an\gamma}{\eta}\right).$$

Because these upper bounds are independent of $t$ then we may conclude that training reaches a steady state after a finite number of iterations. In particular, this means every point either has zero loss or activates no neurons. To prove the network achieves zero loss we need only show that every training point activates at least one neuron after the last training update. This property is simple to prove for clean points: indeed, if $i \in \mathcal{S}_T$ then $i$ activates every neuron in $\Gamma_{y_i}$ after the first iteration. An inductive argument then shows $i$ activates a neuron in every subsequent iteration. Showing that every corrupt point activates a neuron at the end of training is not as simple, and requires a more careful consideration of the training dynamics. To this end we say a neuron is a *carrier* of a training point between iterations $t_0$ and $t$ if $i \in \mathcal{A}_j^{(\tau)}$ for all $\tau \in [t_0, t]$. In order to prove the network fits the corrupt data we need to show each corrupt point $(\mathbf{x}_i, y_i)$ has a carrier neuron in $\Theta_{y_i}$ throughout training. If too many clean points activate such a neuron, then it is possible it will eventually cease to carry any corrupt points and if a corrupt point loses all of its carrier neurons then it cannot be fitted. We show this event cannot occur by studying the activation patterns of neurons in $\Gamma := \Gamma_1 \cup \Gamma_{-1}$.

**Lemma 3.3.** *Let the assumptions of Theorem 3.1 hold and suppose we have a good initialization. Let $j \in \Gamma$ and $t > 0$ be an iteration such that no point achieves zero loss at or before this iteration. For a point $i \in \mathcal{S}_T$, then $i \in \mathcal{A}_j^{(t)}$ iff $i \sim j$. For a point $i \in \mathcal{S}_F$ with $i \nsim j$, $i \in \mathcal{A}_j^{(t)}$ iff $i \in \mathcal{A}_j^{(1)}$.*

The next lemma bounds the activations just before any points achieve zero loss.

---

[2] Here we use $[2m]_{+1}$ to refer to the even indices, or those neurons with positive output weights, while $[2m]_{-1}$ the odd indices, or those neurons with negative output weights.

**Lemma 3.4.** *Under the assumptions of Theorem 3.1 and assuming we have a good initialization, there is an iteration $\mathcal{T}_1 \leq \frac{C}{\eta m[1+(\gamma+\rho)(n-k)]}$ before any point achieves zero loss where the following hold for a constant that varies from line to line.*

1. *For all $p \in \{-1, 1\}$, $j \in \Gamma_p$, $i \sim j$, and $i \in \mathcal{S}_T$, then $\langle \mathbf{w}_j^{(\mathcal{T}_1)}, \mathbf{x}_i \rangle \geq cm^{-1}$.*

2. *For all $p \in \{-1, 1\}$, $j \in \Gamma_p$, $i \not\sim j$, and $i \in \mathcal{S}_T$, then $\langle \mathbf{w}_j^{(\mathcal{T}_1)}, \mathbf{x}_i \rangle \leq -cn\gamma m^{-1}$.*

3. *For all $i \in \mathcal{S}_T$, then $\ell(\mathcal{T}_1, \mathbf{x}_i) \leq c$.*

Due to the fact that clean points are the majority and all of them push the network in the same signal direction, then immediately after $\mathcal{T}_1$ the loss of clean points is small and clean points activate all neurons in the relevant $\Gamma_p$ strongly. Furthermore, once the loss of a clean point is small it stays small. In subsequent iterations, if the number of corrupt updates since $\mathcal{T}_1$ is also small, approximately $C\varepsilon n\gamma/(\eta(\gamma + \rho))$, then each clean point will activate on all but an $\varepsilon$ proportion of neurons in the relevant $\Gamma_p$. As the hinge loss switches off the updates from a point once it reaches zero loss, eventually clean points do not participate in every iteration. Furthermore, when they do participate their updates are spread over a large proportion of the neurons. This ensures that most neurons in $\Theta_p$ cannot receive too many clean updates in isolation, thereby ensuring carrier neurons continue to carry corrupted points throughout training.

Lastly, the generalization result follows from the near orthogonality of the noise components of both the training and test data. Indeed, using the same concentration bound, a test point satisfies the same inner product noise condition as the training data with high probability.

**Lemma 3.5.** *Consider a test label $y \in \{-1, 1\}$ and point $\mathbf{x} := y\sqrt{\gamma}\mathbf{v} + \sqrt{1-\gamma}\mathbf{n}$, where $\mathbf{n} \sim \text{Uniform}(\mathcal{S}^{d-1} \cap \text{span}\{\mathbf{v}\}^\perp)$ is mutually i.i.d. from the training sample. Assume the conditions of Theorem 3.1 hold and that we have a good initialization. In addition, suppose that $|\langle \mathbf{n}, \mathbf{n}_l \rangle| < \frac{\rho}{1-\gamma}$ for all $l \in [2n]$, then $yf(\mathcal{T}_{end}, \mathbf{x}) > 0$.*

$\square$

## 3.2 Non-benign overfitting

The next theorem states a harmful overfitting result: for sufficiently small $\gamma$ the network achieves again zero loss on both the clean and corrupt data after a finite number of iterations, but the probability of misclassification is bounded from below by a constant. A detailed proof of this Theorem along with the associated lemmas is provided in Appendix D.

**Theorem 3.6.** *Let Assumption 1 hold and further assume $m \geq C\log(n/\delta)$, $\rho \leq cn^{-1}$, $\eta < 1/(2mn)$ and $\gamma \leq \frac{c}{\sqrt{nd}}$. Then with probability at least $1 - \delta$ over the randomness of the dataset and network initialization the following hold.*

1. *The training process terminates at an iteration $\mathcal{T}_{end} \leq \frac{Cn}{\eta}$.*

2. *For all $i \in [2n]$ then $\ell(\mathcal{T}_{end}, \mathbf{x}_i) = 0$.*

3. *The generalization error satisfies*

$$\mathbb{P}(\text{sgn}(f(\mathcal{T}_{end}, \mathbf{x})) \neq y) \geq \frac{1}{8}.$$

We remark that the above result holds for $n \geq 1$ and any $k$. Indeed, in this regime the noise components dominate the training dynamics and we therefore expect the performance of the network on test points to be close to random. We re-emphasize that, unlike in the data model used by Frei et al. (2022) and Xu & Gu (2023), there does exist a classifier with perfect generalization error for arbitrarily small $\gamma$. The significance of Theorem 3.6 is that under the data model considered GD results in a suboptimal classifier.

*Proof sketch.* Similar to the proof of Theorem 3.1, in the context of non-benign overfitting we say the initialization is "good" if $\max_{i \neq l} |\langle \mathbf{n}_i, \mathbf{n}_l \rangle| \leq \frac{\rho}{1-\gamma}$ and if each point in the training sample

activates a neuron of the same sign. Under the conditions of Theorem 3.6 it can be shown that a good initialization in this context happens with high probability.

**Lemma 3.7.** *In addition to the conditions of Theorem 3.6, suppose we have a good initialization and that for some iteration $t_0$ then $\ell(t_0, \mathbf{x}_i) \leq a$ for all $i \in [2n]$. Then $T(t_0, t) \leq \frac{Cna}{\eta}$.*

As for the benign overfitting case, we need to show that each training point activates a neuron after the last training iteration. Under the assumptions on $\gamma$ it can be shown that the loss of a point decreases during every iteration it participates in, regardless of the status and activations of other points in the training sample. All that remains is to lower bound the generalization error. To this end observe for a test point $(\mathbf{x}, y)$ that

$$y(f(\mathcal{T}_{\text{end}}, \mathbf{x}) - f(\mathcal{T}_{\text{end}}, -\mathbf{x})) = \sum_{j=1}^{2m} y(-1)^j \langle \mathbf{w}_j^{(\mathcal{T}_{\text{end}})}, \mathbf{x} \rangle.$$

If the right-hand-side of this equality is negative we can conclude that either $\mathbf{x}$ or $-\mathbf{x}$ is misclassified. That this event is true with probability lower bounded by a constant in turn follows by appropriately upper bounding the norm of the network weights in the signal subspace, as well as lower bounding the norm of the network weights in the noise subspace. □

## 3.3 No-overfitting

The following theorem illustrates that for $\gamma$ larger than the upper bound required for benign overfitting, then after convergence, which occurs in a finite number of iterations, only the clean points achieve zero loss. By contrast, the corrupt points cease to activate any neurons and are thus zeroed by the network. The network also achieves asymptotically optimal test error. A detailed proof of this theorem along with the associated lemmas is provided in Appendix E.

**Theorem 3.8.** *Let Assumption 1 hold and further assume $m \geq 2$, $n \geq C \log\left(\frac{m}{\delta}\right)$, $\rho \leq c\gamma$ and $cn^{-1} \leq \gamma \leq ck^{-1}$. Then there exists a sufficiently small step-size $\eta$ such that with probability at least $1 - \delta$ over the randomness of the dataset and network initialization we have the following.*

1. *The training process terminates at an iteration $\mathcal{T}_{end} \leq \frac{Cn}{\eta}$.*

2. *For all $i \in \mathcal{S}_T$ then $\ell(\mathcal{T}_{end}, \mathbf{x}_i) = 0$ while $\ell(\mathcal{T}_{end}, \mathbf{x}_i) = 1$ for all $i \in \mathcal{S}_F$.*

3. *The generalization error satisfies*

$$\mathbb{P}(\text{sgn}(f(\mathcal{T}_{end}, \mathbf{x})) \neq y) \leq \exp\left(-cd\gamma^2\right).$$

We remark that the upper bound on $\gamma$ allows us to re-deploy the same proof technique used to prove convergence in the benign overfitting case, thereby ensuring the training process converges within a finite number of iterations. We conjecture this upper bound can be relaxed but leave such an analysis to future work.

*Proof sketch.* In the context of no-overfitting we identify a "good" initialization as one for which $\max_{i \neq l} |\langle \mathbf{n}_i, \mathbf{n}_l \rangle| \leq \frac{\rho}{1-\gamma}$ and $\Gamma = \Gamma_{-1} \cup \Gamma_{+1} = [2m]$. Under the conditions of Theorem 3.8 it can be shown a good initialization in this context occurs with high probability, furthermore the resulting activation pattern early during training is simple to characterize.

**Lemma 3.9.** *Suppose that the conditions of Theorem 3.8 hold and that we have a good initialization. Consider an arbitrary $j \in [2m]$ and iteration $2 \leq t \leq \mathcal{T}_0$ occurring before a point has achieved zero loss. Then $i \in \mathcal{A}_j^{(t)}$ iff $i \sim j$.*

Next we bound the activations of the training points just before $\mathcal{T}_0$, the iteration at which any training points first achieve zero loss. In the following we use $F_1$, $F_2$ and $F_3$ as placeholders for expressions depending on the data and model parameters. Here, for the sake of conveying the ideas in the proof we do not write them in full and refer the reader to Supplementary Material.

**Lemma 3.10.** *Suppose that the conditions of Theorem 3.8 hold and that we have a good initialization, then there is an iteration $\mathcal{T}_1$ before any point achieves zero loss such that*

$$\langle \mathbf{w}_j^{(\mathcal{T}_1)}, \mathbf{x}_i \rangle \leq \frac{F_1}{m} \ \text{if } i \in \mathcal{S}_F, i \sim j,$$

$$\langle \mathbf{w}_j^{(\mathcal{T}_1)}, \mathbf{x}_i \rangle \geq \frac{F_2}{m} \ \text{if } i \in \mathcal{S}_T, i \sim j,$$

$$\langle \mathbf{w}_j^{(\mathcal{T}_1)}, \mathbf{x}_i \rangle \leq -\frac{F_3}{m} \ \text{if } i \nsim j.$$

Next we seek to ensure the activation patterns remain mostly fixed: in particular, we show $i \in \mathcal{A}_j^{(t)}$ if $i \in \mathcal{S}_T$ and $i \sim j$, while $i \notin \mathcal{A}_j^{(t)}$ if $i \nsim j$.

**Lemma 3.11.** *Suppose that the conditions of Theorem 3.8 hold and that we have a good initialization. In addition, for $a, b \in \mathbb{R}$ assume there is a time $t_0$ such that $\ell(t_0, \mathbf{x}_i) \leq a$ for all $i \in \mathcal{S}_T$ and $\phi(\langle \mathbf{w}_j^{(t_0)}, \mathbf{x}_i \rangle) \leq b$ for all $i \in \mathcal{S}_F$ and $i \sim j$. If $i \in \mathcal{S}_T$, $i \sim j$ implies $i \in \mathcal{A}_j^{(\tau)}$ and $i \nsim j$ implies $i \notin \mathcal{A}_j^{(\tau)}$ for all $\tau$ satisfying $t_0 \leq \tau < t$, then*

$$B_j(t_0, t) \leq \frac{Ck}{\eta}\left(b + \frac{a}{m}\right), \qquad \sum_{j \sim s} G_j(t_0, t) \leq \frac{C(a + mb)}{\gamma\eta}.$$

As before, this update bound is finite and iteration-independent, therefore GD converges provided the assumptions on the activation patterns are not violated. Furthermore, if these activation patterns do hold, then every clean point activates a neuron and no corrupt point activates a neuron of the same label sign. Therefore, under the assumption on the activation pattern, at convergence clean points achieve zero loss while corrupt points have non-zero loss, i.e., they activate no neurons. It therefore suffices to prove the condition on the activation pattern, which we show holds as long as

$$\min\left\{\frac{F_3}{m}, \frac{F_2}{m}\right\} \geq Ck(\gamma + \rho)\eta\left(\frac{F_1 + 1 - F_2}{m}\right) \geq \eta(\gamma + \rho)B_j(t_0, t).$$

As $C$ does not depend on the parameters, we can ensure this condition holds by letting $Ck(\gamma + \rho)$ be sufficiently small. With $\rho \leq cn^{-1}$, we show it suffices that $\gamma < ck^{-1}$. Finally, the generalization result follows in a fashion almost identical to that used for Lemma 3.5. $\qquad\square$

### 3.4 Comparison of results

We compare the differing regimes of our results side-by-side with those of Frei et al. (2022); Xu & Gu (2023) in Table 1. We note that comparisons are not like-for-like as Frei et al. (2022) consider smooth, leaky ReLU and logistic loss, Xu & Gu (2023) a generalized family of activation functions, which includes ReLU, and logistic loss, and this paper ReLU and hinge loss. Furthermore, in addition to differences in the noise distribution discussed in Section 2.1, Frei et al. (2022); Xu & Gu (2023) assume a data model where the norm of each data point is approximately proportional to $\sqrt{d}$. We therefore re-scale their results in order to make comparison with this work in which all data points have unit norm.

Taken together these results suggest, at least under the type of data model considered, that benign overfitting occurs for signal strengths proportional to between roughly $1/\sqrt{dn}$ and $1/n$. Furthermore, our results also suggest that above approximately $1/n$ one might expect to see a transition to no-overfitting, while below approximately $1/\sqrt{nd}$ a transition to harmful overfitting. We provide preliminary supporting experiments in the Supplementary Material. We again remark that the latter is non-trivial in our setting as for all $\gamma > 0$ the classifier $h(\mathbf{x}) = \text{sign}(\langle \mathbf{v}, \mathbf{x} \rangle)$ always has perfect accuracy.

---

[3]Frei et al. (2022) show overfitting results for smaller $\gamma$, but assume a data model where benign overfitting is possible only when $1/(\gamma\sqrt{nd})$ is asymptotically zero, implicitly showing non-benign overfitting.

Table 1: across all results $k \leq cn$ while $d \geq Cn^2 \log(n/\delta)$ for (Frei et al., 2022), Xu & Gu (2023) and Theorem 3.1.

|  | Frei et al. (2022) | Xu & Gu (2023) | Theorem 3.1 | Theorem 3.6 | Theorem 3.8 |
|---|---|---|---|---|---|
| $n \geq C \cdot$ | $\log\left(\dfrac{1}{\delta}\right)$ | $\log\left(\dfrac{m}{\delta}\right)$ | $\log\left(\dfrac{1}{\delta}\right)$ | $1$ | $\log\left(\dfrac{m}{\delta}\right)$ |
| $m \geq C \cdot$ | $1$ | $\log\left(\dfrac{n}{\delta}\right)$ | $\log\left(\dfrac{n}{\delta}\right)$ | $\log\left(\dfrac{n}{\delta}\right)$ | $1$ |
| $\gamma \leq c \cdot$ | $\dfrac{1}{n}$ | $\dfrac{1}{n}$ | $\dfrac{1}{n}$ | $\dfrac{1}{\sqrt{nd}}$ | $\dfrac{1}{k}$ |
| $\gamma \geq C \cdot$ | $\dfrac{1}{\sqrt{nd}}$ | $\sqrt{\dfrac{\log(\frac{md}{n\delta})}{nd}}$ | $\sqrt{\dfrac{\log(\frac{n}{\delta})}{d}}$ | $0$ | $\dfrac{1}{n}$ |
| Result | Benign[3] | Benign | Benign | Non-benign | No-overfit |

## 4   Conclusion

Developing a theoretical description of benign overfitting in neural networks is a highly nascent area, with mathematical results available only for very limited data models. Furthermore, the conditions describing the transitions between overfitting versus non-overfitting and benign versus non-benign even in these simplified settings are yet to be fully characterized. The goal of this work was to address this issue as well as explore the impact of using the hinge loss. In particular, and admittedly for a simple data model, we prove three different training outcomes, corresponding to non-benign overfitting, benign overfitting and no-overfitting, based on conditions on the margin of the clean data. Our analysis also differs significantly from prior works due to the fact the ratio of loss between different training points can be unbounded and the implicit bias of using hinge loss versus exponentially tailed loss is poorly understood.

**Limitations and future work:**   the key limitation of this work is the restrictiveness of the data model. In particular, as in prior and related works we use a near-orthogonal noise model and assume a rank one signal, we also place additional conditions on the noise distribution. In addition to generalizing the signal and noise model as well as improving the bounds required for our results to hold, we believe the following themes are important areas for future research: first relaxing the near orthogonal noise condition, second exploring data models beyond those which are linearly separable, third investigating the role and impact of depth.

### Acknowledgments

EG, WS and DN were partially supported by NSF DMS 2011140 and NSF DMS 2108479. EG was also partially supported by NSF DGE 2034835.

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

## Appendix A   Properties of the data and network at initialization

For each of our results to hold we require certain properties on both the network weights and training sample to hold at initialization. Here we bound the probabilities of these events in turn. Later, for each specific setting we combine the relevant conditions using the union bound.

First, and in order to prove convergence, we require the noise components of the training sample to be approximately orthogonal to one another.

**Lemma A.1.** *Let $\rho, \delta \in (0,1)$. Given a sequence $(\mathbf{n}_i)_{i=1}^{2n}$ of mutually i.i.d. random vectors with $\mathbf{n}_i \sim U(\mathbb{S}^{d-1} \cap span(\mathbf{v})^\perp)$ for all $i \in [2n]$, then assuming $d \geq \max\left\{3, 3\rho^{-2} \ln\left(\frac{2n^2}{\delta}\right)\right\}$*

$$\mathbb{P}\left(\bigcap_{i,l \in [2n], i \neq \ell} \{|\langle \mathbf{n}_i, \mathbf{n}_\ell\rangle| \leq \rho\}\right) \geq 1 - \delta.$$

*Proof.* Consider two pairs of mutually i.i.d. random vectors $\mathbf{n}, \mathbf{n}' \sim U(\mathbb{S}^{d-1} \cap \mathrm{span}(\mathbf{v})^\perp)$ and $\mathbf{u}, \mathbf{u}' \sim U(\mathbb{S}^{d-2})$, observe

$$\langle \mathbf{n}, \mathbf{n}'\rangle \overset{d}{=} \langle \mathbf{u}, \mathbf{u}'\rangle.$$

Due to the fact that $\mathbf{u}$ and $\mathbf{u}'$ are independent as well as the rotational invariance of $U(\mathbb{S}^{d-2})$ it follows that

$$\langle \mathbf{u}, \mathbf{u}'\rangle \overset{d}{=} \langle \mathbf{u}, \mathbf{e}_1\rangle,$$

where $\mathbf{e}_1 := [1, 0..0]^T$. Let $\mathrm{Cap}(\mathbf{e}_1, \rho) := \{\mathbf{z} \in \mathbb{S}^{d-2} : \langle \mathbf{e}_1, \mathbf{z}\rangle \geq \rho\}$ denote the *spherical cap* of $\mathbb{S}^{d-2}$ centered on $\mathbf{e}_1$. As $d \geq 3$ then from Ball (1997)[Lemma 2.2] it follows that

$$\mathbb{P}(|\langle \mathbf{n}, \mathbf{n}'\rangle| \geq \rho) = \mathbb{P}(\mathbf{u} \in \mathrm{Cap}(\mathbf{e}_1, \rho)) \leq \exp\left(-\frac{(d-1)\rho^2}{2}\right) \leq \exp\left(-\frac{d\rho^2}{3}\right).$$

Applying the union bound

$$\mathbb{P}\left(\bigcap_{i,\ell \in [2n], i \neq \ell} \{|\langle \mathbf{n}_i, \mathbf{n}_\ell\rangle| \leq \rho\}\right) = 1 - \mathbb{P}\left(\bigcup_{i,\ell \in [2n], i \neq \ell} \{|\langle \mathbf{n}_i, \mathbf{n}_\ell\rangle| \geq \rho\}\right)$$

$$\geq 1 - 2n^2 \mathbb{P}\left(|\langle \mathbf{n}_i, \mathbf{n}_\ell\rangle| \geq \rho\right)$$

$$\geq 1 - 2n^2 \exp\left(-\frac{d\rho^2}{3}\right).$$

Setting $\delta \geq 2n^2 \exp\left(-\frac{d\rho^2}{3}\right)$ and rearranging we arrive at the result claimed. □

In addition to requiring the approximate orthogonality property on the training data, our approach also requires a large proportion of the neurons at initialization to satisfy particular conditions in regard to the number of clean versus corrupt activations. To this end, we introduce the following terms where $p \in \{-1, 1\}$.

- Let $\Gamma_p := \{j : (-1)^j = p, \ G_j(0,1)(\gamma - \rho) - B_j(0,1)(\gamma + \rho) \geq \frac{2\lambda_w}{\eta}\}$ denote the set of neurons with output weight $(-1)^p$ which have more clean points activating them than corrupt ones at initialization. We will show that these sets of neurons have a *predictable* behavior early during training before any clean points achieve zero loss. We further let $\Gamma = \Gamma_1 \cup \Gamma_{-1}$.

- Let $\Theta_p := \{j \sim \Gamma_p : G_j(0,1)(\gamma + \rho) - B_j(0,1)(\gamma - \rho) < 1 - \gamma + \rho\} \subset \Gamma_p$. For our benign overfitting result we will show that neurons in this subset are able to *carry* corrupt points throughout training, eventually, at least in the overfitting setting, enabling them to achieve zero loss. We further let $\Theta = \Theta_1 \cup \Theta_{-1}$.

First we show $\Gamma$ accounts for a significant proportion of neurons. To this end we first provide the following result.

**Lemma A.2.** *Define* $\mu := \frac{2k}{n+k}$ *and assume* $\kappa \in (0,1)$ *satisfies* $\kappa > \mu$. *Given an arbitrary neuron* $\mathbf{w}_j \sim U(\mathbb{S}^{d-1})$, *we say that a collection of training points is* $(\varepsilon, \kappa)$-*good iff both* $T_j(0,1) \geq 1$ *and* $B_j(0,1) < \kappa T_j(0,1)$ *with probability at least* $1 - \epsilon$ *over the randomness of the neuron. There exist positive constants* $C, c$ *such that if* $\delta := \exp(-cn(\kappa - \mu^2))$ *and* $n \geq C$ *then with probability at least* $1 - \frac{\delta}{\epsilon}$ *the training sample is* $(\varepsilon, \kappa)$-*good.*

*Proof.* First we establish certain pieces of notation specific to what follows: we say a point $\mathbf{x}$ is positive iff $\langle \mathbf{x}, \mathbf{v} \rangle > 0$ and is negative iff $\langle \mathbf{x}, \mathbf{v} \rangle < 0$. We use $\mathcal{S}^+$ and $\mathcal{S}^-$ to denote these sets of points respectively. Note by construction, see (1), clean and corrupt points of the same sign are mutually i.i.d. As here we only ever consider one neuron and the activations of the training sample on this neuron at initialization, we also drop both the subscript $j$ as well as the argument parentheses on the counting functions. We also use $\pm$ superscripts to denote the subsets corresponding to activations from positive and negative points respectively: as examples $T$ is used as shorthand for the total number of activations, $B^+$ is the number corrupt positive activations and $G^-$ is the number of clean negative activations.

First by the symmetry of the distribution of $\mathbf{w}$, $\mathbb{P}(\langle \mathbf{w}, \mathbf{v} \rangle > 0) = \mathbb{P}(\langle \mathbf{w}, \mathbf{v} \rangle < 0) = \frac{1}{2}$. As a result

$$
\mathbb{P}((B < \kappa T) \cap (T > 0)) = \frac{1}{2} \mathbb{P}((B < \kappa T) \cap (T > 0) \mid \langle \mathbf{w}, \mathbf{v} \rangle > 0)
$$
$$
+ \frac{1}{2} \mathbb{P}((B < \kappa T) \cap (T > 0) \mid \langle \mathbf{w}, \mathbf{v} \rangle < 0).
$$

As the analysis and results derived under either condition will prove identical under reversal of the signs involved, without loss of generality we let $\langle \mathbf{w}, \mathbf{v} \rangle > 0$. Using the union bound

$$
\mathbb{P}((B < \kappa T) \cap (T > 0) \mid \langle \mathbf{w}, \mathbf{v} \rangle > 0) \geq 1 - \mathbb{P}(T = 0 \mid \langle \mathbf{w}, \mathbf{v} \rangle > 0) - \mathbb{P}(B \geq \kappa T \mid \langle \mathbf{w}, \mathbf{v} \rangle > 0),
$$

therefore it suffices to upper bound the two probabilities on the right-hand-side.

Observe if $\langle \mathbf{w}, \mathbf{v} \rangle > 0$ then for $\mathbf{x} \in \mathcal{S}^+$ we have $\mathbb{P}(\langle \mathbf{x}, \mathbf{w} \rangle) > 1/2$ and for $\mathbf{x} \in \mathcal{S}^-$ we have $\mathbb{P}(\langle \mathbf{x}, \mathbf{w} \rangle) < 1/2$. By the mutual independence of the preactivations $(\langle \mathbf{x}, \mathbf{w}_j \rangle)_{i=1}^{2n}$ then $\mathbb{P}(T = 0 \mid \langle \mathbf{w}, \mathbf{v} \rangle > 0) \leq (1/2)^n$. Consider now a slightly different data model, in which a training sample consists of $n - k$ clean positive points and $2k$ corrupt positive points. Abusing notation, we let $\zeta$ denote the event that we are instead drawing our training sample in this manner and also that $\langle \mathbf{w}, \mathbf{v} \rangle > 0$. In this setting $T^+ = T$ and furthermore the event $B < \kappa T$ is equivalent to $B^+ < \kappa T^+$. Again, as the preactivations are mutually independent the number of positive activations can be lower bounded using a binomial distribution with probability $1/2$. Applying a Chernoff bound it follows that

$$
\mathbb{P}\left(T^+ < \frac{n+k}{4} \,\middle|\, \zeta\right) \leq \exp\left(-\frac{n+k}{16}\right).
$$

Furthermore, observe sampling positive points which activate $\mathbf{w}_j$ is equivalent to uniformly sampling without replacement $T^+$ points from $\mathcal{S}^+$. Let $Z_\ell = 1$ iff the $\ell$-th element sampled from $\mathcal{S}^+$ is corrupt and is 0 otherwise. Using a variant of Hoeffding's bound for sampling without replacement (see for example Proposition 1.2 of Bardenet & Maillard (2015))

$$
\mathbb{P}\left(B^+ \geq \kappa T^+ \mid \zeta\right) = \mathbb{P}\left(\frac{1}{T^+} \sum_{\ell=1}^{T^+} Z_\ell - \mu \geq \kappa - \mu\right) \leq \exp\left(-2T^+(\kappa - \mu)^2\right).
$$

Therefore

$$
\mathbb{P}(B \geq \kappa T \mid \langle \mathbf{w}, \mathbf{v} \rangle > 0) \leq \mathbb{P}\left(B^+ \geq \kappa T^+ \mid \zeta\right)
$$
$$
\leq \mathbb{P}\left(B^+ \geq \kappa T^+ \,\middle|\, T^+ \geq \frac{n+k}{4}, \zeta\right) + \mathbb{P}\left(T^+ < \frac{n+k}{4} \,\middle|\, \zeta\right)
$$
$$
\leq 2 \exp\left(-\frac{(n+k)(\kappa - \mu)^2}{16}\right).
$$

Combining these results it follows that

$$
\mathbb{P}((B < \kappa T) \cap (T > 0) \mid \langle \mathbf{w}, \mathbf{v} \rangle > 0) \geq 1 - \mathbb{P}(T = 0 \mid \langle \mathbf{w}, \mathbf{v} \rangle > 0) - \mathbb{P}(B \geq \kappa T \mid \langle \mathbf{w}, \mathbf{v} \rangle > 0)
$$
$$
\geq 1 - 3 \exp\left(-\frac{(n+k)(\kappa - \mu)^2}{16}\right).
$$

Therefore, there exist constants $C, c > 0$ such that if $n \geq C$ then

$$\mathbb{P}((B < \kappa T) \cap (T > 0) \mid \langle \mathbf{w}, \mathbf{v} \rangle > 0) \geq 1 - \delta.$$

Note if instead we condition on the event $\langle \mathbf{x}, \mathbf{w} \rangle < 0$ then swapping the roles of the negative and positive points in the argument above gives the same outcome. As a result

$$\mathbb{P}((B < \kappa T) \cap (T > 0)) \geq 1 - \delta.$$

For convenience let $X := (\mathbf{x}_i)_{i=1}^{2n}$ denote the training sample and $X_{\epsilon, \kappa}^c := \{X : \mathbb{P}_w((B \geq \kappa T) \cup (T = 0)) > \epsilon\}$ the set of training samples which are *not* $(\epsilon, \kappa)$-good. Note here that the subscript $w$ indicates randomness over the neuron alone, furthermore by construction

$$\mathbb{P}\left((B \geq \kappa T) \cup (T = 0) \mid X \in X_{\epsilon, \kappa}^c\right) \geq \epsilon.$$

Furthermore, as

$$\delta \geq \mathbb{P}\left((B \geq \kappa T) \cup (T = 0)\right) \geq \mathbb{P}\left((B \geq \kappa T) \cup (T = 0) \mid X \in X_{\epsilon, \kappa}^c\right) \mathbb{P}(X \in \chi_{\epsilon, \kappa}^c),$$

then it follows that $\mathbb{P}\left(X \in X_{\epsilon, \kappa}^c\right) \leq \frac{\delta}{\epsilon}$. As a result we conclude that the probability of drawing a $(\epsilon, \kappa)$-good training sample is at least $1 - \frac{\delta}{\epsilon}$. $\qquad\square$

Based on Lemma A.2, the following lemma bounds the probability that the cardinality of $\Gamma_p$ is large. We note that the result presented here on non-overfitting requires $|\Gamma_p| = m$ while the result on benign overfitting that $|\Gamma_p| \geq (1 - \alpha)m$ for some small constant $\alpha \in (0, 1)$.

**Lemma A.3.** *Suppose* $n \geq 15k$, $\lambda_w \leq \eta \frac{\gamma - \rho}{4\gamma}$ *and* $\gamma \geq 4\rho$. *Then there exist positive constants* $C, c > 0$ *such that if* $n \geq C$ *the following are true.*

1. $\mathbb{P}\left(|\Gamma_p| = m\right) \geq 1 - m \exp(-cn)$.

2. *With* $\alpha \in (0, 1)$ *then*

$$\mathbb{P}\left(|\Gamma_p| \geq (1 - \alpha)m\right) \geq 1 - \exp(-cn).$$

*Proof.* Again as here we only ever consider the activations at initialization, we write $T_j(0, 1)$ as $T_j$, $G_j(0, 1)$ as $G_j$ and $B_j(0, 1)$ as $B_j$. Let $p \in \{-1, 1\}$ and consider an arbitrary neuron $j$ such that $(-1)^j = p$, by definition if

$$G_j(\gamma - \rho) - B_j(\gamma + \rho) \geq \frac{2\lambda_w}{\eta}$$

we may conclude $j \in \Gamma_p$. Rearranging this expression, equivalently $j \in \Gamma_p$ if

$$B_j + \frac{\lambda_w}{\eta} \leq \frac{\gamma - \rho}{2\gamma} T_j.$$

As $\frac{\lambda_w}{\eta} \leq \frac{\gamma - \rho}{4\gamma}$, then membership to $\Gamma_p$ is guaranteed as long as $T_j \geq 1$ and $B_j \leq \frac{\gamma - \rho}{4\gamma} T_j$. Note by the assumptions of the lemma $\mu := \frac{2k}{n+k} \leq \frac{1}{8}$ and $\frac{\gamma - \rho}{4\gamma} \geq \frac{3}{16}$. Conditioning on the event we draw a $(\varepsilon, \frac{3}{16})$-good training sample then the probability that $j \notin \Gamma_p$ is at most $\epsilon$ by Lemma A.2. Furthermore, with the training sample fixed the activations of each neuron are mutually independent. Let $X = (\mathbf{x}_i)_{i=1}^{2n}$ denote the draw of the training sample and $X_{\epsilon, \kappa} = \{X : \mathbb{P}_w((B \geq \kappa T) \cup (T = 0)) \leq \epsilon\}$ the set of $(\varepsilon, \kappa)$-good training samples. In what follows we assume $\kappa = 3/16$ and let $\epsilon = \exp(-cn)$ where $c < 16^{-2}$ is a sufficiently small positive constant, then by Lemma A.2 there exist constants $C, c$ such that if $n \geq C$

$$\mathbb{P}\left(X \in X_{\epsilon, \kappa}\right) \geq 1 - \frac{\delta}{\epsilon} \geq 1 - \exp\left(-cn\right).$$

The first result follows by applying the union bound,

$$\begin{aligned}
\mathbb{P}\left(|\Gamma_p| = m\right) &\geq \mathbb{P}\left(|\Gamma_p| = m \mid X \in X_{\epsilon, \kappa}\right) \mathbb{P}\left(X \in X_{\epsilon, \kappa}\right) \\
&\geq (1 - m \exp(-cn))(1 - \exp(-cn)) \\
&\geq 1 - m \exp(-cn).
\end{aligned}$$

For the second result, let $\alpha' \in (0,1)$ denote the smallest scalar satisfying both $\alpha < \alpha'$ and $\alpha' m \in [m]$. Observe

$$\mathbb{P}\left(|\Gamma_p| < (1-\alpha)m \mid X \in X_{\epsilon,\kappa}\right) = \mathbb{P}\left(\exists \mathcal{J} \subset [2m]_p, |\mathcal{J}| = \alpha' m : j \notin \Gamma_p \ \forall j \in \mathcal{J} \mid X \in X_{\epsilon,\kappa}\right)$$

$$\leq \binom{m}{\alpha' m} \epsilon^{\alpha' m}$$

$$\leq \left(\frac{\epsilon e}{\alpha'}\right)^{\alpha' m}.$$

As $\alpha$ is a constant, there exists positive constants $C, c$ such that if $n \geq C$ then

$$\frac{\epsilon e}{\alpha} = \exp(-cn + 1 + \log(1/\alpha')) \leq \exp(-cn).$$

Therefore

$$\mathbb{P}\left(|\Gamma_p| \geq (1-\alpha)m\right) \geq \mathbb{P}\left(|\Gamma_p| \geq (1-\alpha)m \mid X \in X_{\epsilon,\kappa}\right) \mathbb{P}\left(X \in X_{\epsilon,\kappa}\right)$$

$$\geq \left(1 - \exp\left(-cmn\alpha'\right)\right)\left(1 - \exp\left(-cn\right)\right)$$

$$\geq 1 - \exp(-cn)$$

as claimed. $\qquad \square$

We now turn our attention to establishing the conditions required at initialization on the corrupt points for the result on benign overfitting. To this end, in the following two lemmas we introduce the notion of an $\epsilon$-*fine* training sample and lower bound the probability of drawing one. Then, by conditioning on drawing such a training sample, we lower bound the cardinality of a set of neurons, which we denote $\Lambda$, which satisfy a property related to $\Theta$.

**Lemma A.4.** *Assume* $\gamma \leq \frac{4}{5n}$ *and* $\gamma \geq 5\rho$. *We say a training sample is* $\epsilon$-*fine if for a random neuron* $\mathbf{w}_j$ *the inequality* $G_j < \frac{4}{10}T_j + \frac{5}{8}n$ *holds with probability at least* $1 - \epsilon$. *There exist constants* $C, c > 0$ *such that if* $n \geq C$ *then with probability at least* $1 - \epsilon^{-1}\exp\left(-cn\right)$ *the training sample is* $\epsilon$-*fine.*

*Proof.* This lemma is analogous to Lemma A.2 and to this end we reuse much of the same notation. In particular, recall a point $\mathbf{x}$ is positive iff $\langle \mathbf{x}, \mathbf{v} \rangle > 0$ and is negative iff $\langle \mathbf{x}, \mathbf{v} \rangle < 0$. Note all points with the same sign are mutually i.i.d. by construction. We use $\mathcal{S}^+$ and $\mathcal{S}^-$ to denote these sets of positive and negative points respectively. As here we only consider a single random neuron $\mathbf{w}_j$ and the activations at initialization of the training sample on it, we also drop both the subscript $j$ as well as the argument parentheses on the counting functions. We also use $\pm$ superscripts to denote the subsets corresponding to activations from clean and corrupt points. As indicative examples of our notation going forward, we denote $\mathbf{w}_j$ as $\mathbf{w}$, $T$ is used as shorthand for the total number of activations while $B^+$ and $G^-$ are the number of corrupt positive and clean negative activations respectively.

By the symmetry of the distribution of $\mathbf{w}$, $\mathbb{P}(\langle \mathbf{w}, \mathbf{v} \rangle > 0) = \mathbb{P}(\langle \mathbf{w}, \mathbf{v} \rangle < 0) = \frac{1}{2}$. As a result

$$\mathbb{P}\left(G < \frac{4}{10}T + \frac{5}{8}n\right) = \frac{1}{2}\mathbb{P}\left(G < \frac{4}{10}T + \frac{5}{8}n \mid \langle \mathbf{w}, \mathbf{v} \rangle > 0\right)$$

$$+ \frac{1}{2}\mathbb{P}\left(G < \frac{4}{10}T + \frac{5}{8}n \mid \langle \mathbf{w}, \mathbf{v} \rangle < 0\right).$$

As the analysis and results derived under either condition will prove identical under reversal of the signs involved, without loss of generality we let $\langle \mathbf{w}, \mathbf{v} \rangle > 0$. Consider this problem for a slightly different data model, in which a training sample consists of $2(n-k)$ clean, positive points and $k$ negative points. Abusing notation, we let $\zeta$ denote the event that we are instead drawing our training sample in this manner and also that $\langle \mathbf{w}, \mathbf{v} \rangle > 0$. Clearly

$$\mathbb{P}\left(G \geq \frac{4}{10}T + \frac{5}{8}n \mid \langle \mathbf{w}, \mathbf{v} \rangle > 0\right) \leq \mathbb{P}\left(G^+ \geq \frac{4}{10}T^+ + \frac{5}{8}n \mid \zeta\right).$$

In this setting, only positive points activate $\mathbf{w}$, therefore all points which activate $\mathbf{w}$ are identically distributed. As a result, sampling positive points which activate $\mathbf{w}$ is equivalent to uniformly sampling without replacement $T^+$ points from $\mathcal{S}^+$. Let $Z_\ell = 1$ if the $\ell$-th element sampled from $\mathcal{S}^+$ is clean

and is 0 otherwise. Define $\mu = \frac{2(n-k)}{2n-k}$, then again using a variant of Hoeffding's bound for sampling without replacement (Bardenet & Maillard, 2015)[Proposition 1.2] and as long as $\frac{4}{10} + \frac{5n}{8T^+} > \mu$, we have

$$
\begin{aligned}
\mathbb{P}\left(G^+ \geq \frac{4}{10}T^+ + \frac{5}{8}n \mid \zeta\right) &= \mathbb{P}\left(\frac{1}{T^+}G^+ \geq \frac{4}{10} + \frac{5n}{8T^+} \mid \zeta\right) \\
&= \mathbb{P}\left(\frac{1}{T^+}\sum_{l=1}^{T^+} Z_l - \mu \geq \frac{4}{10} + \frac{5n}{8T^+} - \mu \mid \zeta\right) \\
&\leq \exp\left(-2T^+\left(\frac{4}{10} + \frac{5n}{8T^+} - \mu\right)^2\right).
\end{aligned}
\tag{4}
$$

We proceed to lower and upper bound $T^+$, to this end we first lower and upper bound the probability that a positive point activates $\mathbf{w}$ conditioned on the event $\langle \mathbf{w}, \mathbf{v}\rangle > 0$. Let $\mathbf{x} = \sqrt{\gamma}\mathbf{v} + \sqrt{1-\gamma}\mathbf{n}$ be a *fixed* positive point, then by the symmetry of the distribution of $\mathbf{w}$

$$
\begin{aligned}
\mathbb{P}\left(\langle \mathbf{w}, \mathbf{x}\rangle > 0 | \langle \mathbf{w}, \mathbf{v}\rangle > 0\right) &= \mathbb{P}\left(\sqrt{\gamma}\langle \mathbf{w}, \mathbf{v}\rangle + \sqrt{1-\gamma}\langle \mathbf{w}, \mathbf{n}\rangle > 0 | \langle \mathbf{w}, \mathbf{v}\rangle < 0\right) \\
&\geq \mathbb{P}\left(\langle \mathbf{w}, \mathbf{n}\rangle > 0\right) \\
&= \frac{1}{2}.
\end{aligned}
$$

For convenience, let $E_1$ denote the event $\langle \mathbf{w}, \mathbf{x}\rangle > 0$, $E_2$ the event $\langle \mathbf{w}, \mathbf{v}\rangle > 0$ and $E_3$ the event $|\langle \mathbf{w}, \mathbf{v}\rangle| \leq \varphi$ for some arbitrary $\varphi \in (0,1)$. For the upper bound observe

$$
\begin{aligned}
\mathbb{P}\left(\langle \mathbf{w}, \mathbf{x}\rangle > 0 \mid \langle \mathbf{w}, \mathbf{v}\rangle > 0\right) &= \mathbb{P}\left(E_1 \mid E_2\right) \\
&= \mathbb{P}\left(E_1 \mid E_2, E_3\right)\mathbb{P}\left(E_3 \mid E_2\right) + \mathbb{P}\left(E_1 \mid E_2, E_3^c\right)\mathbb{P}\left(E_3^c \mid E_2\right) \\
&\leq \mathbb{P}\left(E_1 \mid E_2, E_3\right) + \mathbb{P}\left(E_3^c \mid E_2\right).
\end{aligned}
$$

Let $\text{Cap}(\mathbf{n}, \varphi) := \{\mathbf{z} \in \mathbb{S}^{d-1} : \langle \mathbf{n}, \mathbf{z}\rangle \geq \varphi\}$ denote the *spherical cap* of $\mathbb{S}^{d-1}$ centered on $\mathbf{n}$. As $d \geq 3$ and $\mathbf{w} \sim U(\mathbb{S}^{d-1})$ then from Ball (1997)[Lemma 2.2] it follows that

$$
\mathbb{P}(|\langle \mathbf{w}, \mathbf{u}\rangle| \geq \varphi) = \mathbb{P}(\mathbf{w} \in \text{Cap}(\mathbf{u}, \varphi)) \leq \exp\left(-\frac{d\rho^2}{3}\right).
$$

Furthermore

$$
\begin{aligned}
\mathbb{P}\left(E_1 \mid E_2, E_3\right) &= \mathbb{P}(\sqrt{\gamma}\langle \mathbf{w}, \mathbf{v}\rangle + \sqrt{1-\gamma}\langle \mathbf{w}, \mathbf{n}\rangle > 0 \mid \langle \mathbf{w}, \mathbf{v}\rangle > 0, |\langle \mathbf{w}, \mathbf{v}\rangle| \leq \varphi) \\
&\leq \mathbb{P}\left(\langle \mathbf{w}, \mathbf{n}\rangle \geq -\sqrt{\frac{\gamma}{1-\gamma}}\varphi\right) \\
&= \frac{1}{2} + \frac{1}{2}\mathbb{P}\left(|\langle \mathbf{w}, \mathbf{n}\rangle| \leq \sqrt{\frac{\gamma}{1-\gamma}}\varphi\right) \\
&\leq 1 - \frac{1}{2}\exp\left(-\frac{d\gamma\varphi^2}{3(1-\gamma)}\right).
\end{aligned}
$$

Therefore, and noting under the assumptions of the lemma that $\frac{\gamma}{1-\gamma} \leq \frac{4}{n}$,

$$
\begin{aligned}
\mathbb{P}\left(\langle \mathbf{w}, \mathbf{x}\rangle > 0 | \langle \mathbf{w}, \mathbf{v}\rangle > 0\right) &\leq 1 + \exp\left(-\frac{d\varphi^2}{3}\right) - \frac{1}{2}\exp\left(-\frac{d\gamma\varphi^2}{3(1-\gamma)}\right) \\
&\leq 1 + \exp\left(-\frac{d\varphi^2}{3}\right) - \frac{1}{2}\exp\left(-\frac{4d\varphi^2}{3n}\right).
\end{aligned}
$$

Set $\varphi^2 = \frac{3\sqrt{n}}{d}$, then

$$
\mathbb{P}\left(\langle \mathbf{w}, \mathbf{x}\rangle > 0 | \langle \mathbf{w}, \mathbf{v}\rangle > 0\right) \leq 1 + \exp\left(-\sqrt{n}\right) - \frac{1}{2}\exp\left(-\frac{4}{\sqrt{n}}\right).
$$

Letting $\omega \in (0, 1/2)$ be an arbitrary constant, then as long $n \geq \max\{\ln^2\left(\frac{1}{\omega}\right), 16\ln^{-2}\left(\frac{1}{1-\omega}\right)\}$

$$
\mathbb{P}\left(\langle \mathbf{w}, \mathbf{x}\rangle > 0 | \langle \mathbf{w}, \mathbf{v}\rangle > 0\right) \leq \frac{1}{2} + \frac{\omega}{2}.
$$

In order to take advantage of concentration the upper bound on $T^+$ must be greater than $\mathbb{E}[T^+]$. On the other hand, if the upper bound is too large then the condition $\frac{4}{10} + \frac{5n}{8T_j^+} > \mu$ will be compromised. As $\mu < 1$ if

$$T^+ = \sum_{i \in \mathcal{S}^+} \mathbb{1}(i \in \mathcal{A}^{(0)}) \leq \frac{100}{96} n$$

then this condition is not compromised. Setting $\omega = \frac{1}{48}$, if $n \geq 16 \ln^{-2}\left(\frac{48}{47}\right) =: C$ then

$$\mathbb{E}[T^+] = (2n - k)\mathbb{P}(i \in \mathcal{A}^{(0)}) \leq \frac{98}{96} n$$

and therefore, applying a Chernoff bound,

$$\mathbb{P}\left(T^+ \leq \frac{99}{96}n \mid \zeta\right) \geq 1 - \exp\left(-\frac{n}{19900}\right).$$

Under the same conditions, to lower bound $T^+$ we again apply a Chernoff, which gives

$$\mathbb{P}\left(T^+ \geq \frac{2(n-k)}{4} \,\Big|\, \zeta\right) \geq 1 - \exp\left(-\frac{2(n-k)}{16}\right).$$

Therefore, there exists a small positive constant $c$ and a large constant positive constant $C$ such that if $n \geq C$ then

$$\mathbb{P}\left(\frac{2(n-k)}{4} \leq T^+ \leq \frac{99}{96}n \,\Big|\, \zeta\right) \geq 1 - \exp\left(-cn\right).$$

In combination with the bound in (4), from this result it follows that there also exists a sufficiently small constant $c > 0$ such that

$$\mathbb{P}\left(G_j^+ \geq \frac{4}{10} + \frac{5}{8}n \;\Big|\; \zeta, \frac{2(n-k)}{4} \leq T^+ \leq \frac{99}{96}n\right) \leq \exp(-cn).$$

As a result, there exist positive constants $C, c$ such that if $n \geq C$ then

$$\mathbb{P}\left(G \geq \frac{4}{10}T + \frac{5}{8}n \mid \langle \mathbf{w}, \mathbf{v}\rangle < 0\right)$$

$$\leq \mathbb{P}\left(G^+ \geq \frac{4}{10}T^+ + \frac{5}{8}n \;\Big|\; \zeta\right)$$

$$\leq \mathbb{P}\left(G^+ \geq \frac{4}{10}T^+ + \frac{5}{8}n \;\Big|\; \zeta, \frac{2(n-k)}{4} \leq T^+ \leq \frac{99}{96}n\right) \mathbb{P}\left(\frac{2(n-k)}{4} \leq T^+ \leq \frac{99}{96}n \;\Big|\; \zeta\right)$$

$$+ \mathbb{P}\left(G^+ \geq \frac{4}{10}T^+ + \frac{5}{8}n \;\Big|\; \zeta, \frac{2(n-k)}{4} > T^+ \text{ or } T^+ > \frac{99}{96}n\right) \mathbb{P}\left(\frac{2(n-k)}{4} > T^+ \text{ or } T^+ > \frac{99}{96}n \;\Big|\; \zeta\right)$$

$$\leq \mathbb{P}\left(G^+ \geq \frac{4}{10}T^+ + \frac{5}{8}n \;\Big|\; \zeta, \frac{2(n-k)}{4} \leq T^+ \leq \frac{99}{96}n\right) \mathbb{P}\left(\frac{2(n-k)}{4} \leq T^+ \leq \frac{99}{96}n \;\Big|\; \zeta\right)$$

$$+ \mathbb{P}\left(\frac{2(n-k)}{4} \geq T^+ \text{ or } T^+ \geq \frac{99}{96}n \;\Big|\; \zeta\right)$$

$$\leq \exp(-cn).$$

Note if instead $\langle \mathbf{x}, \mathbf{v}\rangle < 0$, then swapping the roles of the negative and positive points in the argument outlined above gives the same result. Therefore, under the assumptions of the lemma,

$$\mathbb{P}\left(G < \frac{4}{10}T + \frac{5}{8}n\right) \geq 1 - \exp\left(-cn\right).$$

Let $X = (\mathbf{x}_i)_{i=1}^{2n}$ denote the training sample and $X_\epsilon^c = \{X : \mathbb{P}_w\left(G \geq \frac{4}{10}T + \frac{5}{8}n\right) > \epsilon\}$ the set of training samples which are *not* $\epsilon$-fine. Note the subscript $w$ above indicates randomness over the neuron $\mathbf{w}$ alone. Clearly by construction

$$\mathbb{P}\left(G \geq \frac{4}{10}T + \frac{5}{8}n \mid X \in X_\epsilon^c\right) \geq \epsilon.$$

Furthermore, as

$$\exp\left(-cn\right) \geq \mathbb{P}\left(G \geq \frac{4}{10}T + \frac{5}{8}n\right) \geq \mathbb{P}\left(G \geq \frac{4}{10}T + \frac{5}{8}n \mid X \in X_\epsilon^c\right)\mathbb{P}(X \in X_\epsilon^c),$$

then it follows that $\mathbb{P}\left(X \in X_\epsilon^c\right) \leq \epsilon^{-1}\exp\left(-cn\right)$. As a result we conclude that there exist positive constants $C, c$ such that if $n \geq C$ then the probability of drawing an $\epsilon$-fine training sample is at least $1 - \epsilon^{-1}\exp\left(-cn\right)$. $\qquad\square$

**Lemma A.5.** *Assume $\gamma \leq \frac{4}{5n}$, $\gamma \geq 5\rho$ and let $\Lambda := \{j \in [2m] : G_j < \frac{\gamma-\rho}{2\gamma}T_j + \frac{1}{2\gamma}\}$. There exists a positive constants $C$ such that for any $\delta \in (0,1)$ if $n \geq C\log(1/\delta)$ then*

$$\mathbb{P}(|\Lambda| > 1.75m) \geq 1 - \delta.$$

*Proof.* To bound the size of $\Lambda$ with high probability we follow a similar approach used to bound the size of $\Gamma_p$ with high probability. As $\gamma \leq \frac{4}{5n}$ and $\rho \leq \frac{\gamma}{5}$, then observe

$$\frac{\gamma-\rho}{2\gamma}T_j + \frac{1}{2\gamma} \geq \frac{4}{10}T_j + \frac{5}{8}n.$$

Therefore, $j \in \Lambda$ if $G_j < \frac{4}{10}T_j + \frac{5}{8}n$. Conditioned on the event that the training sample is $\epsilon$-fine for $\epsilon := \exp(-cn)$ for some sufficiently small constant $c$, then with the data fixed the preactivations of the data on each neuron are mutually independent and identically distributed by construction. As such, in this setting the events $(G_j < \frac{4}{10}T_j + \frac{5}{8}n)_{j=1}^{2m}$ are also mutually independent. Let $X = (\mathbf{x}_i)_{i=1}^{2n}$ denote the training sample and $X_\epsilon$ the set of $\epsilon$-fine training samples, then

$$\mathbb{P}\left(|\Lambda| \leq 1.75m \mid X \in X_\epsilon\right) = \mathbb{P}\left(\exists \mathcal{J} \subset [2m]_p, |\mathcal{J}| = 0.25m : j \notin \Lambda\ \forall j \in \mathcal{J} \mid X \in X_\epsilon\right)$$

$$\leq \binom{m}{0.25m}\epsilon^{0.25m}$$

$$\leq (4\epsilon e)^{0.25m}.$$

It follows that there exists a sufficiently small positive constant $c$ such that

$$4\epsilon e = \exp(-cn + 1 + \log(4)) \leq \exp(-cn).$$

Therefore, using Lemma A.4 there exists a sufficiently small positive constant $c$ such that

$$\begin{aligned}
\mathbb{P}(|\Lambda| > 1.75m) &\geq \mathbb{P}(|\Lambda| > 1.75m \mid X \in X_\epsilon)\mathbb{P}(X \in X_\epsilon) \\
&\geq (1 - \exp(-c0.25mn))(1 - \exp(-cn)) \\
&\geq (1 - \exp(-cn))
\end{aligned}$$

from which the result claimed follows. $\qquad\square$

**Lemma A.6.** *Assume $\gamma \leq \frac{4}{5n}$, $\gamma \geq 5\rho$ and $|\Gamma_p| > 0.99m$ for $p \in \{-1, 1\}$. There exists a positive constant $C$ such that for any $\delta \in (0,1)$, if $n \geq C\log(\frac{1}{\delta})$ and $m \geq C\log\left(\frac{k}{\delta}\right)$, then with probability at least $1 - \delta$ for all $i \in S_F$ there exists a $j \in \Theta_{y_i}$ for which $i \in \mathcal{A}_j^{(0)}$.*

*Proof.* For convenience here we use $T_j$, $G_j$ and $B_j$ for $T_j(0,1)$, $G_j(0,1)$ and $B_j(0,1)$ respectively. For a neuron to be in $\Theta_p$ it must satisfy the following condition,

$$G_j(\gamma+\rho) - B_j(\gamma-\rho) < 1 - \gamma + \rho.$$

Adding and subtracting $T_j(\gamma - \rho)$ to the left-hand-side gives

$$G_j2\gamma - T_j(\gamma-\rho) < 1 - \gamma + \rho.$$

Rearranging this inequality it follows that the conditions $j \in \Gamma_p$ and $G_j < \frac{\gamma-\rho}{2\gamma}T_j + \frac{1}{2\gamma}$ are sufficient to conclude $j \in \Theta_p$. Let $\Lambda := \{j \in [2m] : G_j < \frac{\gamma-\rho}{2\gamma}T_j + \frac{1}{2\gamma}\}$. Therefore, in order to prove the desired result it suffices to lower bound the probability that for each corrupt point $(\mathbf{x}_i, y_i)$, the intersection between the set of neurons which $\mathbf{x}_i$ activates, the set of neurons $\Gamma_{y_i}$ and the set of neurons $\Lambda$ is nonempty.

By a Chernoff bound there exists a small constant $c > 0$ such that with probability at least $1 - \exp(-cm)$ a fixed training point is activated by at least $1/3$ of the neurons of each sign. Therefore, using the union bound, every corrupt training point is activated by at least $m/3$ of the neurons with matching sign with probability at least $1 - 2k\exp(-cm)$. Conditioning on this event, then under the assumption $|\Gamma_p| > 0.99m$ for $p \in \{-1, 1\}$, each corrupt point $(\mathbf{x}_i, y_i)$ activates at least $\frac{97}{300}m$ neurons in $\Gamma_{y_i}$. Therefore, if for instance, $|\Lambda| > 1.75m$, we can conclude for each $(\mathbf{x}_i, y_i)$ with $i \in \mathcal{S}_F$ that there exists a $j \in \Theta_{y_i}$ such that $i \in \mathcal{A}_j^{(0)}$. Therefore, under the conditions of the lemma, using the union bound and Lemmas A.5 and A.3, we can upper bound the failure probability of this as

$$2k\exp(-cm) + \exp(-cn).$$

Therefore, for $\delta \in (0, 1)$ there exists a positive constant $C$ such that if $n \geq C\log(\frac{1}{\delta})$ and $m \geq C\log\left(\frac{k}{\delta}\right)$ then the probability that for all $i \in \mathcal{S}_F$ there exists a $j \in \Theta_{y_i}$ such that $i \in \mathcal{A}_j^{(0)}$ is at least $1 - \delta$. □

The final lemma we provide here states, under mild conditions on the network width, that with high probability every point in the training sample activates a neuron whose output weight matches its label in sign. We use this to prove the result on non-benign overfitting, detailed in Section D.

**Lemma A.7.** *Let $\delta \in (0, 1)$, if $m \geq \log_2(\frac{2n}{\delta})$ then the probability that for all $i \in [2n]$ there exists a $j \in [2m]$ such that $(-1)^j = y_i$ and $i \in \mathcal{A}_j^{(0)}$ is at least $1 - \delta$.*

*Proof.* Observe by the rotational symmetry of the weight distribution that for any $j \in [2m]$

$$\mathbb{P}(\langle \mathbf{w}_j, \mathbf{x}_i \rangle < 0) = \mathbb{P}(\langle \mathbf{w}_j, \mathbf{e}_1 \rangle < 0) = 1/2.$$

By construction, for each element in the training sample $(\mathbf{x}_i, y_i)_{i=1}^{2n}$ there are $m$ neurons whose output weight has the same sign. As the preactivations of $\mathbf{x}_i$ with each neuron are mutually independent from one another, then using the union bound it follows that

$$
\begin{aligned}
\mathbb{P}\left(\bigcap_{i=1}^{2n}\{\exists j \in [2m] : (-1)^j = y_i, \ i \in \mathcal{A}_j^{(0)}\}\right) &= 1 - \mathbb{P}\left(\bigcup_{i=1}^{2n}\{\nexists j \in [2m] : (-1)^j = y_i, \ i \in \mathcal{A}_j^{(0)}\}\right) \\
&\geq 1 - 2n\left(\mathbb{P}(\langle \mathbf{w}_j, \mathbf{x}_i \rangle < 0)\right)^m \\
&= 1 - 2n2^{-m}.
\end{aligned}
$$

Setting $\delta \geq 2n2^{-m}$ and rearranging we arrive at the stated result. □

## Appendix B  Supporting Lemmas

### B.1  Bounds on activations and preactivations

For any pair of iterations $t, t_0$ satisfying $t > t_0$, unrolling the GD update rule (3) gives

$$\mathbf{w}_j^{(t)} = \mathbf{w}_j^{(t_0)} + (-1)^j \eta \sum_{\ell=1}^{2n} T_{\ell j}(t_0, t) y_\ell \mathbf{x}_\ell.$$

Using (1) and the fact that $\mathbf{n}_i \perp \mathbf{v}$ for any $i \in [2n]$, then

$$
\begin{aligned}
\langle \mathbf{w}_j^{(t)}, \mathbf{x}_i \rangle &= \langle \mathbf{w}_j^{(t_0)}, \mathbf{x}_i \rangle + (-1)^j \eta \sum_{\ell=1}^{2n} T_{\ell j}(t_0, t) y_\ell \langle \mathbf{x}_\ell, \mathbf{x}_i \rangle \\
&= \langle \mathbf{w}_j^{(t_0)}, \mathbf{x}_i \rangle + (-1)^{j+i} \eta \sum_{\ell=1}^{2n} T_{\ell j}(t_0, t)(-1)^{\ell+i}\beta(\ell)\langle \mathbf{x}_\ell, \mathbf{x}_i \rangle \qquad (5) \\
&= \langle \mathbf{w}_j^{(t_0)}, \mathbf{x}_i \rangle + (-1)^{j+i}\beta(i)\eta \sum_{\ell=1}^{2n} T_{\ell j}(t_0, t)\lambda_{i\ell},
\end{aligned}
$$

where we define $\lambda_{i\ell} := (-1)^{\ell+i}\beta(i)\beta(\ell)\langle \mathbf{x}_\ell, \mathbf{x}_i \rangle$. Towards the goal of bounding the activation of a neuron with a data point we provide the following results.

**Lemma B.1.** *Assume* $|\langle \mathbf{n}_i, \mathbf{n}_\ell \rangle| \leq \frac{\rho}{1-\gamma}$ *for all* $i, \ell \in [2n]$ *such that* $i \neq \ell$.

1. *If* $i = \ell$ *then* $\lambda_{i\ell} = 1$.

2. *If* $i \neq \ell$, $i \in \mathcal{S}_T$, *and* $\ell \in \mathcal{S}_F$, *then* $-(\gamma + \rho) \leq \lambda_{i\ell} \leq -(\gamma - \rho)$.

3. *If* $i \neq \ell$, $i \in \mathcal{S}_F$, *and* $\ell \in \mathcal{S}_T$, *then* $-(\gamma + \rho) \leq \lambda_{i\ell} \leq -(\gamma - \rho)$.

4. *If* $i \neq \ell$ *and* $i, \ell \in \mathcal{S}_T$, *then* $\gamma - \rho \leq \lambda_{i\ell} \leq \gamma + \rho$.

5. *If* $i \neq \ell$ *and* $i, \ell \in \mathcal{S}_F$, *then* $\gamma - \rho \leq \lambda_{i\ell} \leq \gamma + \rho$.

*Proof.* Observe by the data model that

$$\langle \mathbf{x}_i, \mathbf{x}_\ell \rangle = (-1)^{\ell+i} \left( \gamma + (1 - \gamma)\beta(i)\beta(\ell)\langle \mathbf{n}_i, \mathbf{n}_\ell \rangle \right).$$

Therefore

$$\lambda_{i\ell} = \beta(i)\beta(\ell)\gamma + (1 - \gamma)\langle \mathbf{n}_i, \mathbf{n}_\ell \rangle$$

from which the results claimed follow. $\qquad\square$

**Lemma B.2.** *Assume* $|\langle \mathbf{n}_i, \mathbf{n}_\ell \rangle| \leq \frac{\rho}{1-\gamma}$ *for all* $i, \ell \in [2n]$ *such that* $i \neq \ell$. *Then for any* $j \in [2m]$ *the following are true.*

1. *If* $i \in \mathcal{S}_T$, $i \sim j$ *then*

$$\langle \mathbf{w}_j^{(t)}, \mathbf{x}_i \rangle \geq \langle \mathbf{w}_j^{(t_0)}, \mathbf{x}_i \rangle + \eta \left( T_{ij}(t_0, t) + G_j^{(i)}(t_0, t)(\gamma - \rho) - B_j^{(i)}(t_0, t)(\gamma + \rho) \right)$$

$$\langle \mathbf{w}_j^{(t)}, \mathbf{x}_i \rangle \leq \langle \mathbf{w}_j^{(t_0)}, \mathbf{x}_i \rangle + \eta \left( T_{ij}(t_0, t) + G_j^{(i)}(t_0, t)(\gamma + \rho) - B_j^{(i)}(t_0, t)(\gamma - \rho) \right).$$

2. *If* $i \in \mathcal{S}_T$, $i \not\sim j$ *then*

$$\langle \mathbf{w}_j^{(t)}, \mathbf{x}_i \rangle \geq \langle \mathbf{w}_j^{(t_0)}, \mathbf{x}_i \rangle - \eta \left( T_{ij}(t_0, t) + G_j^{(i)}(t_0, t)(\gamma + \rho) - B_j^{(i)}(t_0, t)(\gamma - \rho) \right)$$

$$\langle \mathbf{w}_j^{(t)}, \mathbf{x}_i \rangle \leq \langle \mathbf{w}_j^{(t_0)}, \mathbf{x}_i \rangle - \eta \left( T_{ij}(t_0, t) + G_j^{(i)}(t_0, t)(\gamma - \rho) - B_j^{(i)}(t_0, t)(\gamma + \rho) \right).$$

3. *If* $i \in \mathcal{S}_F$, $i \sim j$ *then*

$$\langle \mathbf{w}_j^{(t)}, \mathbf{x}_i \rangle \geq \langle \mathbf{w}_j^{(t_0)}, \mathbf{x}_i \rangle - \eta \left( T_{ij}(t_0, t) - G_j^{(i)}(t_0, t)(\gamma - \rho) + B_j^{(i)}(t_0, t)(\gamma + \rho) \right)$$

$$\langle \mathbf{w}_j^{(t)}, \mathbf{x}_i \rangle \leq \langle \mathbf{w}_j^{(t_0)}, \mathbf{x}_i \rangle - \eta \left( T_{ij}(t_0, t) - G_j^{(i)}(t_0, t)(\gamma + \rho) + B_j^{(i)}(t_0, t)(\gamma - \rho) \right).$$

4. *If* $i \in \mathcal{S}_F$, $i \not\sim j$ *then*

$$\langle \mathbf{w}_j^{(t)}, \mathbf{x}_i \rangle \geq \langle \mathbf{w}_j^{(t_0)}, \mathbf{x}_i \rangle + \eta \left( T_{ij}(t_0, t) - G_j^{(i)}(t_0, t)(\gamma + \rho) + B_j^{(i)}(t_0, t)(\gamma - \rho) \right)$$

$$\langle \mathbf{w}_j^{(t)}, \mathbf{x}_i \rangle \leq \langle \mathbf{w}_j^{(t_0)}, \mathbf{x}_i \rangle + \eta \left( T_{ij}(t_0, t) - G_j^{(i)}(t_0, t)(\gamma - \rho) + B_j^{(i)}(t_0, t)(\gamma + \rho) \right).$$

*Proof.* Considering (5) we can further separate the summation term as follows,

$$\langle \mathbf{w}_j^{(t)}, \mathbf{x}_i \rangle = \langle \mathbf{w}_j^{(t_0)}, \mathbf{x}_i \rangle + (-1)^{j+i}\beta(i)\eta \left( T_{ij}(t_0, t) + \sum_{\substack{\ell \in \mathcal{S}_T \\ \ell \neq i}} T_{\ell j}(t_0, t)\lambda_{il} + \sum_{\substack{\ell \in \mathcal{S}_F \\ \ell \neq i}} T_{\ell j}(t_0, t)\lambda_{i\ell} \right).$$

Note, with $i \in \mathcal{S}_T$ and $i \sim j$, or $i \in \mathcal{S}_F$ and $i \not\sim j$, then $(-1)^{j+i}\beta(i) = 1$. On the other hand, with $i \in \mathcal{S}_T$ and $i \not\sim j$, or $i \in \mathcal{S}_F$ and $i \sim j$, then $(-1)^{j+i}\beta(i) = -1$. Substituting the relevant bounds on $\lambda_{i\ell}$ provided in Lemma B.1, and observing by definition that $G_j^{(i)}(t_0, t) = \sum_{\ell \in \mathcal{S}_T, \ell \neq i} T_{\ell j}(t_0, t)$ and $B_j^{(i)}(t_0, t) = \sum_{\ell \in \mathcal{S}_T, \ell \neq i} T_{\ell j}(t_0, t)$, one arrives at the results claimed. $\qquad\square$

We will often make use of the following similar but more pessimistic bounds on the activations. Recall that $\phi$ is the ReLU function: $\phi(a) = \max\{a, 0\}$.

**Lemma B.3.** *For any $j \in [2m]$ and iterations $t_0, t$ with $t_0 \leq t$ the following hold:*

1. *If $i \in \mathcal{S}_T, i \sim j$ then*

$$\phi(\langle \mathbf{w}_j^{(t)}, \mathbf{x}_i \rangle) \geq \phi(\langle \mathbf{w}_j^{(t_0)}, \mathbf{x}_i \rangle) + \eta T_{ij}(t_0, t) - \eta(\gamma + \rho)B_j(t_0, t) - \eta\phi(\rho - \gamma)G_j^{(i)}(t_0, t)$$

$$\phi(\langle \mathbf{w}_j^{(t)}, \mathbf{x}_i \rangle) \leq \phi(\langle \mathbf{w}_j^{(t_0)}, \mathbf{x}_i \rangle) + \eta T_{ij}(t_0, t) + \eta(\gamma + \rho)G_j^{(i)}(t_0, t) + \eta\phi(\rho - \gamma)B_j(t_0, t).$$

2. *If $i \in \mathcal{S}_T, i \not\sim j$ then*

$$\phi(\langle \mathbf{w}_j^{(t)}, \mathbf{x}_i \rangle) \geq \phi(\langle \mathbf{w}_j^{(t_0)}, \mathbf{x}_i \rangle) - \eta T_{ij}(t_0, t) - \eta(\gamma + \rho)G_j^{(i)}(t_0, t) - \eta\phi(\rho - \gamma)B_j(t_0, t)$$

$$\phi(\langle \mathbf{w}_j^{(t)}, \mathbf{x}_i \rangle) \leq \phi(\langle \mathbf{w}_j^{(t_0)}, \mathbf{x}_i \rangle) - \eta T_{ij}(t_0, t) + \eta(\gamma + \rho)B_j(t_0, t) + \eta\phi(\rho - \gamma)G_j^{(i)}(t_0, t) + \eta.$$

3. *If $i \in \mathcal{S}_F, i \sim j$ then*

$$\phi(\langle \mathbf{w}_j^{(t)}, \mathbf{x}_i \rangle) \geq \phi(\langle \mathbf{w}_j^{(t_0)}, \mathbf{x}_i \rangle) - \eta T_{ij}(t_0, t) - \eta(\gamma + \rho)B_j^{(i)}(t_0, t) - \eta\phi(\rho - \gamma)G_j(t_0, t)$$

$$\phi(\langle \mathbf{w}_j^{(t)}, \mathbf{x}_i \rangle) \leq \phi(\langle \mathbf{w}_j^{(t_0)}, \mathbf{x}_i \rangle) - \eta T_{ij}(t_0, t) + \eta(\gamma + \rho)G_j(t_0, t) + \eta\phi(\rho - \gamma)B_j^{(i)}(t_0, t) + \eta.$$

4. *If $i \in \mathcal{S}_F, i \not\sim j$ then*

$$\phi(\langle \mathbf{w}_j^{(t)}, \mathbf{x}_i \rangle) \geq \phi(\langle \mathbf{w}_j^{(t_0)}, \mathbf{x}_i \rangle) + \eta T_{ij}(t_0, t) - \eta(\gamma + \rho)G_j(t_0, t) - \eta\phi(\rho - \gamma)B_j^{(i)}(t_0, t)$$

$$\phi(\langle \mathbf{w}_j^{(t)}, \mathbf{x}_i \rangle) \leq \phi(\langle \mathbf{w}_j^{(t_0)}, \mathbf{x}_i \rangle) + \eta T_{ij}(t_0, t) + \eta(\gamma + \rho)B_j^{(i)}(t_0, t) + \eta\phi(\rho - \gamma)G_j(t_0, t).$$

*The $\eta$ term in the upper bound for cases 2 and 3 is only necessary if $T_{ij}(t_0, t) > 0$.*

We remark that we will often use this result in a setting where $\rho \leq \gamma$. In these cases, the terms that involve $\phi(\rho - \gamma)$ are zero and will be dropped.

*Proof.* For each of these results, we make use of Lemma B.2, $a \leq \phi(a)$ for all $a \in \mathbb{R}$, and

$$0 \leq G_j^{(i)}(t_0, t_1) \leq G_j(t_0, t_1)$$

$$0 \leq B_j^{(i)}(t_0, t_1) \leq B_j(t_0, t_1)$$

for all $i, j, t_0, t_1$. We will only prove the inequalities for $i \in \mathcal{S}_T$ here, as the inequalities for $i \in \mathcal{S}_F$ are analogous.

For the first inequality in Statement 1 we claim it suffices to show

$$\phi(\langle \mathbf{w}_j^{(\tau+1)}, \mathbf{x}_i \rangle) \geq \phi(\langle \mathbf{w}_j^{(\tau)}, \mathbf{x}_i \rangle) + \eta T_{ij}(\tau, \tau+1) - \eta(\gamma+\rho)B_j(\tau, \tau+1) - \eta\phi(\rho-\gamma)G_j^{(i)}(\tau, \tau+1) \tag{6}$$

Indeed, if (6) is true then the result claimed follows as

$$\phi(\langle \mathbf{w}_j^{(t)}, \mathbf{x}_i \rangle) - \phi(\langle \mathbf{w}_j^{(t_0)}, \mathbf{x}_i \rangle) = \sum_{\tau=t_0}^{t-1} \phi(\langle \mathbf{w}_j^{(\tau+1)}, \mathbf{x}_i \rangle) - \phi(\langle \mathbf{w}_j^{(\tau)}, \mathbf{x}_i \rangle)$$

$$\geq \sum_{\tau=t_0}^{t-1} \Bigg( \eta T_{ij}(\tau, \tau+1) - \eta(\gamma+\rho)B_j(\tau, \tau+1)$$

$$- \eta\phi(\rho - \gamma)G_j^{(i)}(\tau, \tau+1) \Bigg)$$

$$= \eta T_{ij}(t_0, t) - \eta(\gamma+\rho)B_j(t_0, t) - \eta\phi(\rho-\gamma)G_j^{(i)}(t_0, t).$$

In order to prove (6) we bound

$$\phi(\langle \mathbf{w}_j^{(\tau+1)}, \mathbf{x}_i \rangle) \geq \langle \mathbf{w}_j^{(\tau+1)}, \mathbf{x}_i \rangle$$

$$\geq \langle \mathbf{w}_j^{(\tau)}, \mathbf{x}_i \rangle + \eta T_{ij}(\tau, \tau+1) - \eta(\gamma+\rho)B_j(\tau, \tau+1)$$

$$- \phi(\rho - \gamma)G_j^{(i)}(\tau, \tau+1).$$

This follows from Statement 1 in Lemma B.2. From here, we consider two cases: first, if $\langle \mathbf{w}_j^{(\tau)}, \mathbf{x}_i \rangle \geq 0$ then $\langle \mathbf{w}_j^{(\tau)}, \mathbf{x}_i \rangle = \phi(\langle \mathbf{w}_j^{(\tau)}, \mathbf{x}_i \rangle)$ and so (6) clearly holds. Alternatively, if $\langle \mathbf{w}_j^{(\tau)}, \mathbf{x}_i \rangle < 0$ then $T_{ij}(\tau, \tau+1) = 0$, $\phi(\langle \mathbf{w}_j^{(\tau)}, \mathbf{x}_i \rangle) = 0$ and as a result the right-hand-side of (6) is non-positive while the left is non-negative. As such (6) holds trivially.

For the second equality in Statement 1 we bound

$$\langle \mathbf{w}_j^{(t)}, \mathbf{x}_i \rangle \leq \langle \mathbf{w}_j^{(t_0)}, \mathbf{x}_i \rangle + \eta \left( T_{ij}(t_0, t) + G_j^{(i)}(t_0, t)(\gamma + \rho) - B_j^{(i)}(t_0, t)(\gamma - \rho) \right)$$

$$\leq \phi(\langle \mathbf{w}_j^{(t_0)}, \mathbf{x}_i \rangle) + \eta T_{ij}(t_0, t) + \eta(\gamma + \rho) G_j^{(i)}(t_0, t) + \eta \phi(\rho - \gamma) B_j(t_0, t).$$

Since the right-hand side is non-negative, this inequality is true even if we replace the left-hand side by $\phi(\langle \mathbf{w}_j^{(t)}, \mathbf{x}_i \rangle)$.

We now proceed to Statement 2. For the first inequality, notice that if $\phi(\langle \mathbf{w}_j^{(t_0)}, \mathbf{x}_i \rangle) = 0$ then the right-hand side is non-positive and therefore the inequality trivially holds. Otherwise, it must be the case that $\phi(\langle \mathbf{w}_j^{(t_0)}, \mathbf{x}_i \rangle) = \langle \mathbf{w}_j^{(t_0)}, \mathbf{x}_i \rangle$. Using Statement 2 from Lemma B.2, we obtain the bound

$$\phi(\langle \mathbf{w}_j^{(t)}, \mathbf{x}_i \rangle) \geq \langle \mathbf{w}_j^{(t)}, \mathbf{x}_i \rangle$$

$$\geq \langle \mathbf{w}_j^{(t_0)}, \mathbf{x}_i \rangle - \eta \left( T_{ij}(t_0, t) + G_j^{(i)}(t_0, t)(\gamma + \rho) - B_j^{(i)}(t_0, t)(\gamma - \rho) \right)$$

$$\geq \phi(\langle \mathbf{w}_j^{(t)}, \mathbf{x}_i \rangle) - \eta T_{ij}(t_0, t) - \eta(\gamma + \rho) G_j^{(i)}(t_0, t) - \eta \phi(\rho - \gamma) B_j(t_0, t).$$

We now turn to the second inequality in Statement 2. The corresponding statement from Lemma B.2 yields

$$\langle \mathbf{w}_j^{(t)}, \mathbf{x}_i \rangle \leq \langle \mathbf{w}_j^{(t_0)}, \mathbf{x}_i \rangle - \eta T_{ij}(t_0, t) + \eta(\gamma + \rho) B_j(t_0, t) + \eta \phi(\rho - \gamma) G_j^{(i)}(t_0, t)$$

$$\leq \phi(\langle \mathbf{w}_j^{(t_0)}, \mathbf{x}_i \rangle) - \eta T_{ij}(t_0, t) + \eta(\gamma + \rho) B_j(t_0, t) + \eta \phi(\rho - \gamma) G_j^{(i)}(t_0, t) \qquad (7)$$

$$\leq \phi(\langle \mathbf{w}_j^{(t_0)}, \mathbf{x}_i \rangle) - \eta T_{ij}(t_0, t) + \eta(\gamma + \rho) B_j(t_0, t) + \eta \phi(\rho - \gamma) G_j^{(i)}(t_0, t) + \eta,$$

we remark that the reason for the addition of $\eta$ to the right-hand-side will soon become apparent. The desired inequality holds as long as the right-hand-side is non-negative, we therefore proceed by induction to prove

$$\phi(\langle \mathbf{w}_j^{(t_0)}, \mathbf{x}_i \rangle) - \eta T_{ij}(t_0, \tau) + \eta(\gamma + \rho) B_j(t_0, \tau) + \eta \phi(\rho - \gamma) G_j^{(i)}(t_0, t) + \eta \geq 0$$

for $\tau \geq t_0$. The base case $\tau = t_0$ is trivial, assume then that the induction hypothesis holds for some $\tau \geq t_0$. For iteration $\tau + 1$ there are two cases to consider: first, if $\langle \mathbf{w}_j^{(\tau)}, \mathbf{x}_i \rangle < 0$ then $T_{ij}(t_0, \tau+1) = T_{ij}(t_0, \tau)$. In addition, as $B_j(t_0, \tau) \leq B_j(t_0, \tau+1)$ and $G_j^{(i)}(t_0, \tau) \leq G_j^{(i)}(t_0, \tau+1)$ then

$$0 \leq \phi(\langle \mathbf{w}_j^{(t_0)}, \mathbf{x}_i \rangle) - \eta T_{ij}(t_0, \tau) + \eta(\gamma + \rho) B_j(t_0, \tau) + \eta \phi(\rho - \gamma) G_j^{(i)}(t_0, \tau) + \eta$$

$$\leq \phi(\langle \mathbf{w}_j^{(t_0)}, \mathbf{x}_i \rangle) - \eta T_{ij}(t_0, \tau+1) + \eta(\gamma + \rho) B_j(t_0, \tau+1) + \eta \phi(\rho - \gamma) G_j^{(i)}(t_0, \tau+1) + \eta$$

by the induction hypothesis. Alternatively, if instead $\langle \mathbf{w}_j^{(\tau)}, \mathbf{x}_i \rangle \geq 0$ one may use the second inequality from (7) to conclude that

$$0 \leq \langle \mathbf{w}_j^{(\tau)}, \mathbf{x}_i \rangle \leq \phi(\langle \mathbf{w}_j^{(t_0)}, \mathbf{x}_i \rangle) - \eta T_{ij}(t_0, \tau) + \eta(\gamma + \rho) B_j(t_0, \tau) + \eta \phi(\rho - \gamma) G_j^{(i)}(t_0, \tau).$$

In addition, as $T_{ij}(t_0, \tau + 1) \leq T_{ij}(t_0, \tau) + 1$ it follows that

$$0 \leq \phi(\langle \mathbf{w}_j^{(t_0)}, \mathbf{x}_i \rangle) - \eta T_{ij}(t_0, \tau) + \eta(\gamma + \rho) B_j(t_0, \tau) + \eta \phi(\rho - \gamma) G_j^{(i)}(t_0, \tau)$$

$$\leq \phi(\langle \mathbf{w}_j^{(t_0)}, \mathbf{x}_i \rangle) - \eta T_{ij}(t_0, \tau+1) + \eta(\gamma + \rho) B_j(t_0, \tau+1) + \eta \phi(\rho - \gamma) G_j^{(i)}(t_0, \tau+1) + \eta$$

which completes the induction.

Lastly, we consider the final remark in the statement of the lemma: if $T_{ij}(t_0, t) = 0$ then the right hand side of the second line in (7) is non-negative trivially, so we do not need the additional $\eta$ term. $\qquad \square$

## B.2 Convergence of training

We say that GD terminates if it reaches a finite iteration in which a zero update is applied to the network parameters. The following lemmas are used to show that GD terminates by in turn upper bounding the number of clean and corrupt updates. The first lemma facilitates the bounding of the hinge loss of clean and corrupt points.

**Lemma B.4.** *For any iterations $t, t_0$ satisfying $t \geq t_0$,*

1. *if $i \in \mathcal{S}_T$ then*

$$y_i f(t, \mathbf{x}_i) \geq y_i f(t_0, \mathbf{x}_i) + \eta(T_i(t_0, t) - (\gamma + \rho)B(t_0, t) - \phi(\rho - \gamma)G^{(i)}(t_0, t) - m),$$

2. *if $i \in \mathcal{S}_F$ then*

$$y_i f(t, \mathbf{x}_i) \geq y_i f(t_0, \mathbf{x}_i) + \eta(T_i(t_0, t) - (\gamma + \rho)G(t_0, t) - \phi(\rho - \gamma)B^{(i)}(t_0, t) - m).$$

*Proof.* Both statements follow from the bounds provided in Lemma B.3. For Statement 1

$$
\begin{aligned}
y_i f(t, \mathbf{x}_i) &= \sum_{j \sim i} \phi(\langle \mathbf{w}_j^{(t)}, \mathbf{x}_i \rangle) - \sum_{j \not\sim i} \phi(\langle \mathbf{w}_j^{(t)}, \mathbf{x}_i \rangle) \\
&\geq \sum_{j \sim i} \left( \phi(\langle \mathbf{w}_j^{(t_0)}, \mathbf{x}_i \rangle) + \eta T_{ij}(t_0, t) - \eta(\gamma + \rho)B_j(t_0, t) - \eta \phi(\rho - \gamma)G_j^{(i)}(t_0, t) \right) \\
&\quad - \sum_{j \not\sim i} \left( \phi(\langle \mathbf{w}_j^{(t_0)}, \mathbf{x}_i \rangle) - \eta T_{ij}(t_0, t) + \eta(\gamma + \rho)B_j(t_0, t) + \eta \phi(\rho - \gamma)G_j^{(i)}(t_0, t) + \eta \right) \\
&= y_i f(t_0, \mathbf{x}_i) + \eta(T_i(t_0, t) - (\gamma + \rho)B(t_0, t) - \phi(\rho - \gamma)G^{(i)}(t_0, t) - m).
\end{aligned}
$$

For Statement 2

$$
\begin{aligned}
y_i f(t, \mathbf{x}_i) &= \sum_{j \not\sim i} \phi(\langle \mathbf{w}_j^{(t)}, \mathbf{x}_i \rangle) - \sum_{j \sim i} \phi(\langle \mathbf{w}_j^{(t)}, \mathbf{x}_i \rangle) \\
&\geq \sum_{j \not\sim i} \left( \phi(\langle \mathbf{w}_j^{(t_0)}, \mathbf{x}_i \rangle) + \eta T_{ij}(t_0, t) - \eta(\gamma + \rho)G_j(t_0, t) - \eta \phi(\rho - \gamma)B_j^{(i)}(t_0, t) \right) \\
&\quad - \sum_{j \not\sim i} \left( \phi(\langle \mathbf{w}_j^{(t_0)}, \mathbf{x}_i \rangle) - \eta T_{ij}(t_0, t) + \eta(\gamma + \rho)G_j(t_0, t) + \eta \phi(\rho - \gamma)B_j^{(i)}(t_0, t) + \eta \right) \\
&\geq y_i f(t_0, \mathbf{x}_i) + \eta(T_i(t_0, t) - (\gamma + \rho)G(t_0, t) - \phi(\rho - \gamma)B^{(i)}(t_0, t) - m). \qquad \square
\end{aligned}
$$

The following lemma bounds the number of updates of corrupt and clean points in an interval of iterations in terms of their hinge loss at the beginning of the interval as well as the number of clean and corrupt updates.

**Lemma B.5.** *Let $t \geq t_0$. For $i \in \mathcal{S}_T$,*

$$T_i(t_0, t) \leq \frac{\ell(t_0, \mathbf{x}_i)}{\eta} + (\gamma + \rho)B(t_0, t) + \phi(\rho - \gamma)G^{(i)}(t_0, t) + 3m.$$

*For $i \in \mathcal{S}_F$,*

$$T_i(t_0, t) \leq \frac{\ell(t_0, \mathbf{x}_i)}{\eta} + (\gamma + \rho)G(t_0, t) + \phi(\rho - \gamma)B^{(i)}(t_0, t) + 3m.$$

*Proof.* We will show this for $i \in \mathcal{S}_T$; the $i \in \mathcal{S}_F$ case is analogous but with the roles of corrupt and clean points reversed. We proceed by induction on $t$ and assume $\ell(t_0, \mathbf{x}_i) \leq a$. If $t = t_0$ this holds trivially because the left-hand side is zero and the right-hand side is positive. Otherwise, assume the inequality holds at iteration $t$. By Lemma B.4 and our assumption on $\ell(t_0, \mathbf{x}_i)$,

$$
\begin{aligned}
y_i f(t, \mathbf{x}_i) &\geq y_i f(t_0, \mathbf{x}_i) + \eta(T_i(t_0, t) - (\gamma + \rho)B(t_0, t) - \phi(\rho - \gamma)G^{(i)}(t_0, t) - m) \\
&\geq (1 - a) + \eta(T_i(t_0, t) - (\gamma + \rho)B(t_0, t) - \phi(\rho - \gamma)G^{(i)}(t_0, t) - m).
\end{aligned}
$$

We consider two cases:

1. If $\eta(T_i(t_0, t) - (\gamma + \rho)B(t_0, t) - \phi(\rho - \gamma)G^{(i)}(t_0, t) - m) \geq a$ then we see that $\ell(t, \mathbf{x}_i) = 0$. Therefore,

$$T_i(t_0, t+1) = T_i(t_0, t)$$
$$\leq \frac{a}{\eta} + (\gamma + \rho)B(t_0, t) + 3m$$
$$\leq \frac{a}{\eta} + (\gamma + \rho)B(t_0, t+1) + 3m.$$

2. Otherwise, $T_i(t_0, t) \leq \frac{a}{\eta} + (\gamma + \rho)B(t_0, t) + \phi(\rho - \gamma)G^{(i)}(t_0, t) + m$. Since there are only $2m$ neurons, we bound

$$T_i(t_0, t+1) = T_i(t_0, t) + 2m$$
$$\leq \left( \frac{a}{\eta} + (\gamma + \rho)B(t_0, t) + \phi(\rho - \gamma)G^{(i)}(t_0, t) + m \right) + 2m$$
$$\leq \frac{a}{\eta} + (\gamma + \rho)B(t_0, t+1) + \phi(\rho - \gamma)G^{(i)}(t_0, t) + 3m.$$

$\square$

## Appendix C    Benign overfitting

**Assumption 2.** *With $\delta, \rho \in (0, 1)$ and $C$ a generic, positive constant, then we assume the following conditions on the data and model hyperparameters.*

1. $n \geq C \log(\frac{3}{\delta})$,

2. $m \geq C \log(\frac{6k}{\delta})$,

3. $d \geq \max \left\{ 3, 3\rho^{-2} \ln \left( \frac{9n^2}{\delta} \right) \right\}$.

4. $k < \frac{n}{100}$,

5. $5\sqrt{\frac{\ln(9n^2/\delta)}{d}} \leq \gamma \leq \frac{4}{5n}$,

6. $\lambda_w < \eta$.

In addition to the assumptions detailed in Assumption 2, in our analysis we use three further conditions.

**Assumption 3.** *Let $\rho \in (0, 1)$ satisfy $\gamma \geq 5\rho$. In addition to the assumptions detailed in Assumption 2, assume that the following conditions hold.*

1. $|\Gamma_p| > 0.99m$ *for* $p \in \{-1, 1\}$.

2. *For all $i \in \mathcal{S}_F$ there is $j \in \Gamma_{y_i}$ such that $i \in \mathcal{A}_j^{(0)}$.*

3. *For all $i, l \in [2n]$, $i \neq l$ then $|\langle \mathbf{n}_i, \mathbf{n}_l \rangle| \leq \frac{\rho}{1-\gamma}$.*

We remark that under these conditions then for sufficiently large $n$ the inequalities $\rho \leq \min \left\{ \frac{n-3k}{n+k}\gamma, \frac{1}{6(n-k)} \right\}$ and $\gamma + \rho < \min \left\{ \sqrt{\frac{1}{4(n-k)k}}, \frac{1}{n-k}, \frac{1}{99k}, \frac{1}{100} \right\}$ are satisfied. As shown in the following lemma, these three additional conditions hold with high probability over the randomness of the initialization and training sample.

**Lemma C.1.** *There exists a positive constant $C$ such that for any $\delta \in (0, 1)$ if $n \geq C \log(\frac{3}{\delta})$ then the extra conditions of Assumption 3 hold with probability at least $1 - \delta$.*

*Proof.* Using Lemma A.3, under the Assumption 3 for sufficiently large $n$ there exists a positive constant $c$ such that the probability the first condition does not hold is at most $\exp(-cn)$. Alternatively,

setting $\delta \geq 3 \exp(-cn)$ and rearranging, as long as $n \geq C \log\left(\frac{3}{\delta}\right)$ then the probability the first condition does not hold is at most $\frac{\delta}{3}$. Conditioned on the first event, using Lemma A.6 then if $n \geq C \log(\frac{3}{\delta})$ and $m \geq C \log\left(\frac{6k}{\delta}\right)$ then the probability condition two does not hold is also at most $\frac{\delta}{3}$. Therefore the probability that the first two events hold is at least $(1 - \frac{\delta}{3})^2 \geq 1 - \frac{2\delta}{3}$. For the third condition, noting $\frac{\rho}{1-\gamma} > \rho$ and $\rho \leq \gamma/5$, then by Lemma A.1 the probability the third condition does not hold is also at most $\frac{\delta}{3}$. Therefore, we conclude that all three properties hold with probability at least $1 - \delta$. $\qquad\square$

## C.1 Proof of Lemma 3.2

The following lemma characterizes an iteration independent upper bound on the number of clean and corrupt updates. This result will prove significant for proving the termination of GD.

**Lemma C.2** (Lemma 3.2). *Assume Assumption 3 holds. Suppose further that at some epoch $t_0$ the loss of every clean point is bounded above by $a \in \mathbb{R}_{\geq 0}$, while the loss of every corrupted point is bounded above by $b \in \mathbb{R}_{\geq 0}$. Then the total number of updates which occurs after this epoch is upper bounded as follows,*

$$G(t_0, t) \leq \frac{2(n-k)}{1 - 4k(n-k)(\gamma + \rho)^2} \left( \frac{a}{\eta} + 3m + 2k(\gamma + \rho)\left( \frac{b}{\eta} + 3m \right) \right),$$

$$B(t_0, t) \leq \frac{2k}{1 - 4k(n-k)(\gamma + \rho)^2} \left( \frac{b}{\eta} + 3m + 2(n-k)(\gamma + \rho)\left( \frac{a}{\eta} + 3m \right) \right)$$

*for all $t \geq t_0$.*

*Proof.* From Lemma B.5, $\rho \leq \gamma$, and the assumption on $a$ and $b$,

$$G(t_0, t) = \sum_{i \in \mathcal{S}_T} T_i(t_0, t) \leq 2(n-k)\left( \frac{a}{\eta} + (\gamma + \rho)B(t_0, t) + 3m \right),$$

$$B(t_0, t) = \sum_{i \in \mathcal{S}_F} T_i(t_0, t) \leq 2k\left( \frac{b}{\eta} + (\gamma + \rho)G(t_0, t) + 3m \right).$$

Substituting these bounds into each other, and as $\gamma + \rho < (4(n-k)k)^{-1/2}$ under Assumption 3, we arrive at the iteration independent bound on the number of updates as claimed in the statement of the theorem. $\qquad\square$

## C.2 Early training and proof of Lemma 3.3

**Lemma C.3.** *Under Assumption 3, then for $i \in \mathcal{S}_T$ and $j \in \Gamma$ it follows that $\langle \mathbf{w}_j^{(1)}, \mathbf{x}_i \rangle > 0$ iff $i \sim j$. For $i \in \mathcal{S}_F$, $j \in \Theta_p$, and $i \nsim j$ it follows that $\langle \mathbf{w}_j^{(1)}, \mathbf{x}_i \rangle > 0$ if $i \in \mathcal{A}_j^{(0)}$.*

*Proof.* Suppose $j \in \Gamma$, $i \sim j$, $i \in \mathcal{S}_T$. Recall from definition of $\Gamma_p$ that $G_j^{(i)}(0, 1)(\gamma - \rho) - B_j^{(i)}(0, 1)(\gamma + \rho) \geq \frac{2\lambda_w}{\eta}$. Using Lemma B.2

$$
\begin{aligned}
\langle \mathbf{w}_j^{(1)}, \mathbf{x}_i \rangle &\geq \langle \mathbf{w}_j^{(0)}, \mathbf{x}_i \rangle + \eta\left( T_{ij}(0,1) + G_j^{(i)}(0,1)(\gamma - \rho) - B_j^{(i)}(0,1)(\gamma + \rho) \right) \\
&> \langle \mathbf{w}_j^{(0)}, \mathbf{x}_i \rangle + \eta\left( G_j(0,1)(\gamma - \rho) - B_j(0,1)(\gamma + \rho) \right) \\
&\geq \langle \mathbf{w}_j^{(0)}, \mathbf{x}_i \rangle + 2\lambda_w \\
&> \lambda_w.
\end{aligned}
$$

On the other hand, if $i \nsim j$ then again from Lemma B.2

$$
\begin{aligned}
\langle \mathbf{w}_j^{(1)}, \mathbf{x}_i \rangle &\leq \langle \mathbf{w}_j^{(0)}, \mathbf{x}_i \rangle - \eta\left( T_{ij}(0,1) + G_j^{(i)}(0,1)(\gamma - \rho) - B_j^{(i)}(0,1)(\gamma + \rho) \right) \\
&\leq \langle \mathbf{w}_j^{(0)}, \mathbf{x}_i \rangle - \eta\left( G_j(0,1)(\gamma - \rho) - B_j(0,1)(\gamma + \rho) \right) \\
&\leq -\lambda_w.
\end{aligned}
$$

Now consider $i \in \mathcal{S}_F$, $i \nsim j$, and $i \in \mathcal{A}_j^{(0)}$. By Lemma B.2 and the definition of $\Theta$,

$$\langle \mathbf{w}_j^{(1)}, \mathbf{x}_i \rangle \geq \langle \mathbf{w}_j^{(0)}, \mathbf{x}_i \rangle + \eta \left( T_{ij}(0,1) + B_j^{(i)}(0,1)(\gamma - \rho) - G_j^{(i)}(0,1)(\gamma + \rho) \right)$$
$$> \eta(1 - \gamma + \rho) + \eta \left( B_j(0,1)(\gamma - \rho) - G_j(0,1)(\gamma + \rho) \right)$$
$$\geq 0. \qquad \square$$

**Lemma C.4** (Lemma 3.3). *Suppose Assumption 3 holds. Let $j \in \Gamma_p$. Let $0 < t < \mathcal{T}_0$. A point $i \in \mathcal{A}_j^{(t)}$ if one of the following conditions hold:*

1. $i \in \mathcal{S}_T$ and $i \sim j$

2. $i \in \mathcal{S}_F$, $i \nsim j$, and $i \in \mathcal{A}_j^{(1)}$.

*Furthermore, if one of the following conditions hold, then $i \notin \mathcal{A}_j^{(t)}$:*

1. $i \in \mathcal{S}_T$ and $i \nsim j$

2. $i \in \mathcal{S}_F$, $i \nsim j$, and $i \notin \mathcal{A}_j^{(1)}$.

*Proof.* We proceed by induction. For $t = 1$, the $i \in \mathcal{S}_T$ case was shown in Lemma C.3 and the $i \in \mathcal{S}_F$, $i \nsim j$ case is clear. Now, suppose the lemma holds for iteration $t$ and consider iteration $t + 1$. First let $i \in \mathcal{S}_F$, $i \nsim j$. If $i \in \mathcal{A}_j^{(1)}$ then

$$\langle \mathbf{w}_j^{(t+1)}, \mathbf{x}_i \rangle \geq \langle \mathbf{w}_j^{(t)}, \mathbf{x}_i \rangle + \eta \left( T_{ij}(t, t+1) - G_j^{(i)}(t, t+1)(\gamma + \rho) + B_j^{(i)}(t, t+1)(\gamma - \rho) \right)$$
$$> \eta \left( 1 - (n - k)(\gamma + \rho) \right)$$
$$\geq 0.$$

Here the first line is Lemma B.2, the second line comes from the inductive hypothesis, and the third line comes from $(\gamma + \rho) < \frac{1}{n-k}$ (Assumption 3). If $i \notin \mathcal{A}_j^{(1)}$ then

$$\langle \mathbf{w}_j^{(t+1)}, \mathbf{x}_i \rangle \leq \langle \mathbf{w}_j^{(t)}, \mathbf{x}_i \rangle + \eta \left( T_{ij}(t, t+1) - G_j^{(i)}(t, t+1)(\gamma - \rho) + B_j^{(i)}(t, t+1)(\gamma + \rho) \right)$$
$$< \eta \left( -(n - k)(\gamma - \rho) + 2k(\gamma + \rho) \right)$$
$$= -\eta \left( (n - 3k)\gamma - (n + k)\rho \right)$$
$$\leq 0.$$

Again, the first line is Lemma B.2, the second line uses the inductive hypothesis, and the fourth line uses $\rho \leq \frac{n-3k}{n+k}\gamma$ (Assumption 3).

Now, let $i \in \mathcal{S}_T$. We again use, in order, Lemma B.2, the inductive hypothesis, and $\rho \leq \frac{n-3k}{n+k}\gamma$. If $i \sim j$ then

$$\langle \mathbf{w}_j^{(t+1)}, \mathbf{x}_i \rangle \geq \langle \mathbf{w}_j^{(t)}, \mathbf{x}_i \rangle + \eta \left( T_{ij}(t, t+1) + G_j^{(i)}(t, t+1)(\gamma - \rho) - B_j^{(i)}(t, t+1)(\gamma + \rho) \right)$$
$$> \eta(1 + \rho - \gamma) + \eta \left( (n - k)(\gamma - \rho) - 2k(\gamma + \rho) \right)$$
$$> 0.$$

If $i \nsim j$ then

$$\langle \mathbf{w}_j^{(t+1)}, \mathbf{x}_i \rangle \leq \langle \mathbf{w}_j^{(t)}, \mathbf{x}_i \rangle - \eta \left( T_{ij}(t, t+1) + G_j^{(i)}(t, t+1)(\gamma - \rho) - B_j^{(i)}(t, t+1)(\gamma + \rho) \right)$$
$$< -\eta \left( (n - k)(\gamma - \rho) - 2k(\gamma + \rho) \right)$$
$$= -\eta \left( (n - 3k)\gamma - (n + k)\rho \right)$$
$$\leq 0. \qquad \square$$

**Lemma C.5.** *Suppose Assumption 3 holds. For all $t_0 \leq t_1 < \mathcal{T}_0$,*

$$G_j(t_0, t_1) \leq (n - k)(t_1 - t_0 + 2) + \frac{1}{\gamma - \rho}$$

*Proof.* First we claim that for all $i \in \mathcal{S}_T$, $j \nsim i$, and $t < \mathcal{T}_0$,

$$\langle \mathbf{w}_j^{(t)}, \mathbf{x}_i \rangle \le \lambda_w + 2\eta k(\gamma + \rho).$$

We prove the claim by induction. The base case $t = 0$ follows because $\langle \mathbf{w}_j^{(0)}, \mathbf{x}_i \rangle \le \lambda_w$. Now suppose it is true at iteration $t$. If $\langle \mathbf{w}_j^{(t)}, \mathbf{x}_i \rangle > 0$ then by Lemma B.2,

$$
\begin{aligned}
\langle \mathbf{w}_j^{(t+1)}, \mathbf{x}_i \rangle &\le \langle \mathbf{w}_j^{(t)}, \mathbf{x}_i \rangle - \eta T_{ij}(t, t+1) + \eta(\gamma + \rho) B_j(t, t+1) \\
&\le \langle \mathbf{w}_j^{(t)}, \mathbf{x}_i \rangle - \eta(1 - 2k(\gamma + \rho)) \\
&\le \langle \mathbf{w}_j^{(t)}, \mathbf{x}_i \rangle
\end{aligned}
$$

using $\gamma + \rho < \frac{1}{2k}$. From this, the claim follows. Otherwise,

$$
\begin{aligned}
\langle \mathbf{w}_j^{(t+1)}, \mathbf{x}_i \rangle &\le \langle \mathbf{w}_j^{(t)}, \mathbf{x}_i \rangle - \eta T_{ij}(t, t+1) + \eta(\gamma + \rho) B_j(t, t+1) \\
&\le 2\eta k(\gamma + \rho).
\end{aligned}
$$

We now turn to the statement of the lemma, again proceeding by induction. The base case $t_1 = t_0$ is clear. Otherwise, we consider two cases:

1. If $G_j(t_0, t_1) > \frac{1 + 2k(t_1 - t_0 + 1)(\gamma + \rho)}{\gamma - \rho}$ then for all $i \in \mathcal{S}_T$ and $j \nsim i$, by Lemma B.2,

$$
\begin{aligned}
\langle \mathbf{w}_j^{(t_1)}, \mathbf{x}_i \rangle &\le \langle \mathbf{w}_j^{(t_0)}, \mathbf{x}_i \rangle - \eta \left( T_{ij}(t_0, t_1) + G_j^{(i)}(t_0, t_1)(\gamma - \rho) - B_j^{(i)}(t_0, t_1)(\gamma + \rho) \right) \\
&\le \langle \mathbf{w}_j^{(t_0)}, \mathbf{x}_i \rangle - \eta \left( G_j(t_0, t_1)(\gamma - \rho) - B_j(t_0, t_1)(\gamma + \rho) \right) \\
&< (\lambda_w + 2\eta k(\gamma + \rho)) + 2\eta k(t_1 - t_0)(\gamma + \rho) - \eta G_j(t_0, t_1)(\gamma - \rho) \\
&\le 0
\end{aligned}
$$

by the claim and $\lambda_w < \eta$ (Assumption 3). Therefore, $G_j(t_0, t_1 + 1) \le G_j(t_0, t_1) + (n - k)$.

2. If $G_j(t_0, t_1) \le \frac{1 + 2k(t_1 - t_0 + 1)(\gamma + \rho)}{\gamma - \rho}$, then

$$
\begin{aligned}
G_j(t_0, t_1 + 1) &\le \frac{1 + 2k(t_1 - t_0 + 1)(\gamma + \rho)}{\gamma - \rho} + 2(n - k) \\
&\le \frac{1}{\gamma - \rho} + \frac{2k(t_1 - t_0 + 1)(n - k)}{2k} + 2(n - k) \\
&\le \frac{1}{\gamma - \rho} + (t_1 - t_0 + 3)(n - k). \qquad \square
\end{aligned}
$$

**Lemma C.6.** *Suppose Assumption 3 holds. For all $t < \mathcal{T}_0$, $i \in \mathcal{S}_F$, and $i \sim j$,*

$$\langle \mathbf{w}_j^{(t)}, \mathbf{x}_i \rangle \le (\lambda_w + 2\eta k(\gamma - \rho)) + 2\eta(\gamma + \rho)(n - k) + \frac{\eta(\gamma + \rho)}{\gamma - \rho}$$

*Proof.* Consider $t < \mathcal{T}_0$. We consider three cases

1. If $t = 0$ then

$$\langle \mathbf{w}_j^{(0)}, \mathbf{x}_i \rangle \le \lambda_w.$$

2. If $\langle \mathbf{w}_j^{(t-1)}, \mathbf{x}_i \rangle \le 0$ then by Lemma B.2

$$
\begin{aligned}
\langle \mathbf{w}_j^{(t)}, \mathbf{x}_i \rangle &\le \langle \mathbf{w}_j^{(t-1)}, \mathbf{x}_i \rangle \\
&\quad - \eta \left( T_{ij}(t-1, t) - G_j^{(i)}(t-1, t)(\gamma + \rho) + B_j^{(i)}(t-1, t)(\gamma - \rho) \right) \\
&\le 2\eta k(\gamma - \rho).
\end{aligned}
$$

3. If $\langle \mathbf{w}_j^{(t-1)}, \mathbf{x}_i \rangle > 0$ then let $t' < t$ be the smallest iteration such that $\langle \mathbf{w}_j^{(\tau)}, \mathbf{x}_i \rangle > 0$ for all $t' \leq \tau < t$. By Lemma B.2, Lemma C.5, and the previous two cases above,

$$\langle \mathbf{w}_j^{(t)}, \mathbf{x}_i \rangle \leq \langle \mathbf{w}_j^{(t')}, \mathbf{x}_i \rangle - \eta \left( T_{ij}(t', t) - G_j^{(i)}(t', t)(\gamma + \rho) + B_j^{(i)}(t', t)(\gamma - \rho) \right)$$

$$\leq (\lambda_w + 2\eta k(\gamma - \rho))$$

$$- \eta \left( [1 - (\gamma + \rho)(n - k)](t - t') - 2(\gamma + \rho)(n - k) - \frac{\gamma + \rho}{\gamma - \rho} \right).$$

By $\gamma + \rho < \frac{1}{n-k}$ (Assumption 3) we conclude

$$\langle \mathbf{w}_j^{(t)}, \mathbf{x}_i \rangle \leq (\lambda_w + 2\eta k(\gamma - \rho)) + 2\eta(\gamma + \rho)(n - k) + \frac{\eta(\gamma + \rho)}{\gamma - \rho}. \qquad \square$$

### C.3 Proof of Lemma 3.4

**Lemma C.7** (Lemma 3.4). *Suppose Assumption 3 holds. There is an iteration $\mathcal{T}_1 < \mathcal{T}_0$ during training and expressions $C_1$, $C_2$, and $C_3$ where the following hold:*

*1. For all $p \in \{-1, 1\}$, $j \in \Gamma_p$, $i \sim j$, and $i \in \mathcal{S}_T$,*

$$\langle \mathbf{w}_j^{(\mathcal{T}_1)}, \mathbf{x}_i \rangle \geq \frac{3}{4m}.$$

*2. For all $p \in \{-1, 1\}$, $j \in \Gamma_p$, $i \not\sim j$, and $i \in \mathcal{S}_T$,*

$$\langle \mathbf{w}_j^{(\mathcal{T}_1)}, \mathbf{x}_i \rangle \leq -\frac{3n\gamma}{4m}.$$

*3. For all $i \in \mathcal{S}_T$,*

$$\ell(\mathcal{T}_1, \mathbf{x}_i) \leq \frac{1}{3}.$$

*Furthermore,*

$$\mathcal{T}_1 = \frac{1}{1.03\eta m(1 + (\gamma + \rho)(n - k))} + O(1)$$

*Proof.* Fix $i \in \mathcal{S}_T$. At every iteration $1 \leq t < \mathcal{T}_0$, we bound

$$\ell(t, \mathbf{x}_i) - \ell(t + 1, \mathbf{x}_i) = y_i[f(t + 1, \mathbf{x}_i) - f(t, \mathbf{x}_i)]$$

$$= \sum_{j=1}^{2m} (-1)^{i+j}[\phi(\langle \mathbf{w}_j^{(t+1)}, \mathbf{x}_i \rangle) - \phi(\langle \mathbf{w}_j^{(t)}, \mathbf{x}_i \rangle)]$$

$$= \sum_{j \notin \Gamma_{(-1)^{i+1}}} (-1)^{i+j}[\phi(\langle \mathbf{w}_j^{(t+1)}, \mathbf{x}_i \rangle) - \phi(\langle \mathbf{w}_j^{(t)}, \mathbf{x}_i \rangle)]$$

$$\leq \eta \sum_{j \notin \Gamma_{(-1)^{i+1}}} T_{ij}(t, t + 1) + (\gamma + \rho)G_j^{(i)}(t, t + 1)$$

$$= \eta \left( \sum_{j \in \Gamma_{(-1)^i}} T_{ij}(t, t + 1) + (\gamma + \rho)G_j^{(i)}(t, t + 1) \right)$$

$$+ \eta \left( \sum_{j \notin \Gamma} T_{ij}(t, t + 1) + (\gamma + \rho)G_j^{(i)}(t, t + 1) \right)$$

$$\leq \eta 0.99m[1 + (\gamma + \rho)(n - k)] + 0.02\eta m[1 + 2(\gamma + \rho)(n - k)],$$

where we use in order: $t < \mathcal{T}_0$, the definition of $f(t, \mathbf{x})$, Lemma C.4, Lemma B.3, $\Gamma_{-1} \cap \Gamma_1 = \emptyset$, and Lemma C.4 again. We also use $|\Gamma_p| \geq 0.99m$ (Assumption 3). We further simplify this bound to conclude

$$\ell(t + 1, \mathbf{x}_i) - \ell(t, \mathbf{x}_i) \leq 1.03\eta m[1 + (\gamma + \rho)(n - k)]$$

Additionally, we bound

$$\ell(1, \mathbf{x}_i) = 1 - y_i f(1, \mathbf{x}_i)$$

$$= 1 - \sum_{j=1}^{2m} (-1)^{i+j} \phi(\langle \mathbf{w}_j^{(1)}, \mathbf{x}_i \rangle)$$

$$\geq 1 - \sum_{i \sim j} \phi(\langle \mathbf{w}_j^{(1)}, \mathbf{x}_i \rangle)$$

$$\geq 1 - \sum_{i \sim j} [\phi(\langle \mathbf{w}_j^{(0)}, \mathbf{x}_i \rangle) + \eta T_{ij}(0, 1) + \eta(\gamma + \rho) G_j^{(i)}(0, 1)]$$

$$\geq 1 - m[\lambda_w + \eta + 2\eta(\gamma + \rho)(n - k)].$$

Therefore as long as

$$t < \frac{1 - m[\lambda_w + \eta + 2\eta(\gamma + \rho)(n - k)]}{1.03\eta m[1 + (\gamma + \rho)(n - k)]} + 1 = \frac{1}{1.03\eta m[1 + (\gamma + \rho)(n - k)]} + O(1),$$

then

$$\ell(t, \mathbf{x}_i) = \ell(1, \mathbf{x}_i) + \sum_{t'=1}^{t} \ell(t' + 1, \mathbf{x}_i) - \ell(t', \mathbf{x}_i)$$

$$\geq 1 - m[\lambda_w + \eta + 2\eta(\gamma + \rho)(n - k)] - 1.03(t' - 1)\eta m[1 + (\gamma + \rho)(n - k)]$$

$$> 0.$$

Notice that this does not depend on $i$ or $p$. Therefore we can let $\mathcal{T}_1$ be the largest integer satisfying this bound for $t$ and bound $\ell(\mathcal{T}_1, \mathbf{x}_\ell) > 0$ for all $l \in \mathcal{S}_T$. To verify that $\mathcal{T}_1 < \mathcal{T}_0$, consider $i \in \mathcal{S}_F$:

$$y_i f(t, \mathbf{x}_i) = \sum_{j=1}^{2m} -(-1)^{i+j} \phi(\langle \mathbf{w}_j^{(t)}, \mathbf{x}_i \rangle)$$

$$= \sum_{i \nsim j} [(\phi(\langle \mathbf{w}_j^{(t)}, \mathbf{x}_i \rangle) - \phi(\langle \mathbf{w}_j^{(0)}, \mathbf{x}_i \rangle)) + \phi(\langle \mathbf{w}_j^{(0)}, \mathbf{x}_i \rangle)]$$

$$\leq \sum_{i \nsim j} \left( T_{ij}(0, t) + (\gamma + \rho) B_j^{(i)}(0, t) + \lambda_w \right)$$

$$\leq \eta m t [1 + 2(\gamma + \rho)k] + m\lambda_w$$

This is less than 1 for all $t < \mathcal{T}_1$ since $k \leq \frac{n}{3}$ (Assumption 3).

Now, fix $i \in \mathcal{S}_T$ again. For $i \sim j$, we then can use Lemma B.2 and Lemma C.4:

$$\langle \mathbf{w}_j^{(\mathcal{T}_1)}, \mathbf{x}_i \rangle \geq \langle \mathbf{w}_j^{(1)}, \mathbf{x}_i \rangle + \eta \left( T_{ij}(1, \mathcal{T}_1) + (\gamma - \rho) G_j^{(i)}(1, \mathcal{T}_1) - (\gamma + \rho) B_j^{(i)}(1, \mathcal{T}_1) \right)$$

$$\geq 0 + \eta(\mathcal{T}_1 - 2)(1 + (n - k - 1)(\gamma - \rho) - 2k(\gamma + \rho))$$

$$= \frac{1 + (n - k - 1)(\gamma - \rho) - 2k(\gamma + \rho)}{1.03m[1 + (\gamma + \rho)(n - k)]} + O(\eta)$$

$$\geq \frac{1}{m} - \frac{2\rho(n - k) - 2k(\gamma + \rho) - (\gamma + \rho) - 2k(\gamma + \rho)}{m(1 + (\gamma + \rho)(n - k))} + O(\eta)$$

$$\geq \frac{1}{m} - \frac{1/6 + 4/99 + 1/100}{m} + O(\eta)$$

$$\geq \frac{3}{4m}.$$

using $\rho \le \frac{1}{6(n-k)}$, $\eta$ is sufficiently small, and $\gamma + \rho < \min\left\{\frac{1}{99k}, \frac{1}{100}\right\}$ (Assumption 3). Now assume $i \not\sim j$. Using Lemma B.2 and Lemma C.4 we can bound

$$
\begin{aligned}
\langle \mathbf{w}_j^{(\mathcal{T}_1)}, \mathbf{x}_i \rangle &\le \langle \mathbf{w}_j^{(1)}, \mathbf{x}_i \rangle - \eta\left(T_{ij}(1, \mathcal{T}_1) + (\gamma - \rho)G_j^{(i)}(1, \mathcal{T}_1) - (\gamma + \rho)B_j^{(i)}(1, \mathcal{T}_1)\right) \\
&\le 0 - \eta(\mathcal{T}_1 - 2)((n-k)(\gamma - \rho) - 2k(\gamma + \rho)) \\
&= -\frac{(n-k)(\gamma - \rho) - 2k(\gamma + \rho)}{1.03m[1 + (\gamma + \rho)(n-k)]} + O(\eta) \\
&\le -\frac{(n - 3k)\gamma - (n + k)\rho}{1.03m} + O(\eta) \\
&\le -\frac{0.97n\gamma - \frac{1.01}{5}n}{1.03m} + O(\eta) \\
&\le -\frac{3n\gamma}{4m}.
\end{aligned}
$$

using $\eta$ is sufficiently small and $k \le \frac{n}{100}$ and $\rho \le \frac{\gamma}{5}$ (Assumption 3). Likewise, for $1 \le t < \mathcal{T}_1$,

$$
\begin{aligned}
\ell(t, \mathbf{x}_i) - \ell(t+1, \mathbf{x}_i) &= y_i[f(t+1, \mathbf{x}_i) - f(t, \mathbf{x}_i)] \\
&= \sum_{j=1}^{2m} (-1)^{i+j}[\phi(\langle \mathbf{w}_j^{(t+1)}, \mathbf{x}_i \rangle) - \phi(\langle \mathbf{w}_j^{(t)}, \mathbf{x}_i \rangle)] \\
&= \sum_{j \notin \Gamma_{(-1)^{i+1}}} (-1)^{i+j}[\phi(\langle \mathbf{w}_j^{(t+1)}, \mathbf{x}_i \rangle) - \phi(\langle \mathbf{w}_j^{(t)}, \mathbf{x}_i \rangle)] \\
&= \eta\left(\sum_{j \in \Gamma_{(-1)^i}} [\langle \mathbf{w}_j^{(t+1)}, \mathbf{x}_i \rangle - \langle \mathbf{w}_j^{(t)}, \mathbf{x}_i \rangle]\right) \\
&\quad + \eta\left(\sum_{j \notin \Gamma} (-1)^{i+j}[\phi(\langle \mathbf{w}_j^{(t+1)}, \mathbf{x}_i \rangle) - \phi(\langle \mathbf{w}_j^{(t)}, \mathbf{x}_i \rangle)]\right) \\
&\ge \eta\left(\sum_{j \in \Gamma_{(-1)^i}} T_{ij}(t_0, t) + G_j^{(i)}(t_0, t)(\gamma - \rho) - B_j^{(i)}(t_0, t)(\gamma + \rho)\right) \\
&\quad + \eta\left(\sum_{j \notin \Gamma} T_{ij}(t, t+1) - (\gamma + \rho)B_j(t, t+1)\right) \\
&\ge 0.99\eta m[1 + (\gamma - \rho)(n - k - 1) - 2(\gamma + \rho)k] - 0.04\eta m(\gamma + \rho)k \\
&\ge 0.99\eta m[1 + (\gamma - \rho)(n - k - 1)] - 2.02\eta m(\gamma + \rho)k
\end{aligned}
$$

In the first six lines we use: $t < \mathcal{T}_0$, the definition of $f(t, \mathbf{x})$, Lemma C.4, Lemma B.2 and Lemma B.3, Lemma C.4 again, and $|\Gamma_p| \ge 0.99m$ (Assumption 3), respectively.

We also bound

$$
\begin{aligned}
\ell(1, \mathbf{x}_i) &= 1 - y_i f(1, \mathbf{x}_i) \\
&= 1 - \sum_{j=1}^{2m} (-1)^{i+j} \phi(\langle \mathbf{w}_j^{(1)}, \mathbf{x}_i \rangle) \\
&\le 1 + \sum_{i \not\sim j} \phi(\langle \mathbf{w}_j^{(1)}, \mathbf{x}_i \rangle) \\
&\le 1 + \sum_{i \not\sim j} [\phi(\langle \mathbf{w}_j^{(0)}, \mathbf{x}_i \rangle) - \eta T_{ij}(0, 1) + \eta(\gamma + \rho)B_j(0, 1) + \eta] \\
&\le 1 + m[\lambda_w + \eta + 2\eta(\gamma + \rho)k].
\end{aligned}
$$

Combining these two bounds we see that

$$\ell(\mathcal{T}_1, \mathbf{x}_i) \leq \ell(1, \mathbf{x}_i) - \eta(\mathcal{T}_1 - 1)(0.99m(1 + (\gamma - \rho)(n - k - 1)) - 2.02m(\gamma + \rho)k)$$

$$\leq 1 - \frac{0.99m[1 + (\gamma - \rho)(n - k - 1)] - 2.02m(\gamma + \rho)k}{1.03m(1 + (\gamma + \rho)(n - k))} + O(\eta)$$

$$\leq \frac{0.99((\gamma + \rho) + 2\rho(n - k - 1)) + 0.08 + \frac{4.04}{99}}{1.03m(1 + (\gamma + \rho)(n - k))} + O(\eta)$$

$$\leq \frac{0.99(\frac{1}{100} + \frac{1}{6} + 0.08 + \frac{4.04}{99})}{1.03} + O(\eta)$$

$$\leq \frac{1}{3}.$$

at this iteration, as desired. Here we use $(\gamma + \rho) \leq \min\left\{\frac{1}{99k}, \frac{1}{100}, \frac{1}{n-k}\right\}$, $\eta$ is sufficiently small, and $\rho \leq \frac{1}{6(n-k)}$ (Assumption 3) $\qquad\square$

### C.4 Late training

**Lemma C.8.** *Suppose Assumption 3 holds. Fix $\varepsilon > 0$. We will say a neuron is aligned (at iteration $t$) if*

$$(-1)^j \operatorname{sgn}\langle \mathbf{w}_j^{(t)}, \mathbf{x}_i \rangle = y_i$$

*for all $i \in \mathcal{S}_T$. For $t \geq \mathcal{T}_1$, if*

$$B(\mathcal{T}_1, t) \leq \frac{5\varepsilon n}{8\eta}$$

*than at least $(1 - \varepsilon)m$ neurons in each $\Gamma_p$ will be aligned.*

*Proof.* Let $p \in \{-1, 1\}$ and $t$ be such that $\varepsilon m$ different neurons in $\Gamma_p$ are unaligned at iteration $t$. For any neuron index $j \in \Gamma_p$ and $i \in \mathcal{S}_T$, we can use Lemma B.2 to bound

$$(-1)^{i+j}\langle \mathbf{w}_j^{(t)}, \mathbf{x}_i \rangle \geq (-1)^{i+j}\langle \mathbf{w}_j^{(\mathcal{T}_1)}, \mathbf{x}_i \rangle + \eta T_{ij}(\mathcal{T}_1, t)$$
$$+ \eta(\gamma - \rho)G_j^{(i)}(\mathcal{T}_1, t) - \eta(\gamma + \rho)B_j^{(i)}(\mathcal{T}_1, t)$$
$$= \min\left\{\frac{3}{4m}, \frac{3n\gamma}{4m}\right\} - \eta(\gamma + \rho)B_j(\mathcal{T}_1, t)$$
$$\geq \frac{3n\gamma}{4m} - \eta(\gamma + \rho)B_j(\mathcal{T}_1, t).$$

Since $n\gamma < 1$ (Assumption 3), we see that $\min\{\frac{3}{4m}, \frac{3n\gamma}{4m}\} = \frac{3n\gamma}{4m}$. If the lower bound above is positive then $(-1)^j \operatorname{sgn}\langle \mathbf{w}_j^{(t)}, \mathbf{x}_i \rangle = y_i$. Therefore, if a neuron $j$ is unaligned then $B_j(\mathcal{T}_1, t) \geq \frac{3n\gamma}{4\eta m(\gamma + \rho)} \geq \frac{5n}{8\eta m}$ (using $\rho \leq \frac{\gamma}{5}$ from Assumption 3). If there are $\varepsilon m$ unaligned neurons, then

$$B(\mathcal{T}_1, t) = \sum_{j=1}^{2m} B_j(\mathcal{T}_1, t) \geq \frac{5\varepsilon n}{8\eta}. \qquad\square$$

Denote the first iteration after $\mathcal{T}_1$ where more than $\varepsilon m$ neurons in one of the $\Gamma_p$ are unaligned as $\mathcal{T}_\varepsilon$. If no such iteration exists, let $\mathcal{T}_\varepsilon = \infty$. We will eventually show that indeed $\mathcal{T}_\varepsilon = \infty$, by showing that the training process reaches zero loss before such an iteration can happen.

**Lemma C.9.** *Assume Assumption 3 holds and also $\gamma + \rho < \frac{0.99(1-\varepsilon)}{4k}$. There is an iteration $\mathcal{T}_2 \geq \mathcal{T}_1$ so that for all iterations $t$ satisfying $\mathcal{T}_2 \leq t < \mathcal{T}_\varepsilon$ and all $i \in \mathcal{S}_T$,*

$$\ell(t, \mathbf{x}_i) \leq 4\eta(\gamma + \rho)km.$$

*Furthermore, we can choose $\mathcal{T}_2$ so that*

$$\mathcal{T}_2 - \mathcal{T}_1 \leq \frac{1}{3\eta m(0.99(1 - \varepsilon) - 4k(\gamma + \rho))} + 1.$$

*Proof.* Fix $i \in \mathcal{S}_T$ and $t < \mathcal{T}_\varepsilon - 1$. Suppose $\ell(t, \mathbf{x}_i) > 0$. Using Lemma B.3 and $t < \mathcal{T}_\varepsilon$,

$$
\begin{aligned}
y_i f(t+1, \mathbf{x}_i) - y_i f(t, \mathbf{x}_i) &= \sum_{j=1}^{2m} (-1)^{i+j} (\phi(\langle \mathbf{w}_j^{(t+1)}, \mathbf{x}_i \rangle) - \phi(\langle \mathbf{w}_j^{(t)}, \mathbf{x}_i \rangle)) \\
&\geq \sum_{j=1}^{2m} \eta(T_{ij}(t, t+1) - (\gamma + \rho) B_j(t, t+1)) \\
&\geq \eta 0.99(1 - \varepsilon)m - 4\eta mk(\gamma + \rho).
\end{aligned}
$$

Therefore,

$$
\ell(t+1, \mathbf{x}_i) \leq \min\{\ell(t, \mathbf{x}_i) - \eta m(0.99(1-\varepsilon) - 4k), 0\}.
$$

By Lemma C.7, the loss of each clean point at $\mathcal{T}_1$ is at most $\frac{1}{3}$, so each clean point reaches zero loss in at most

$$
\left\lceil \frac{1}{3\eta m(0.99(1-\varepsilon) - 4k(\gamma + \rho))} \right\rceil
$$

iterations.

Now suppose $\ell(t, \mathbf{x}_i) = 0$. We similarly argue

$$
\begin{aligned}
y_i f(t+1, \mathbf{x}_i) - y_i f(t, \mathbf{x}_i) &= \sum_{j=1}^{2m} (-1)^{i+j} (\phi(\langle \mathbf{w}_j^{(t+1)}, \mathbf{x}_i \rangle) - \phi(\langle \mathbf{w}_j^{(t)}, \mathbf{x}_i \rangle)) \\
&\geq \sum_{j=1}^{2m} \eta(T_{ij}(t, t+1) - B_j(t, t+1)) \\
&\geq -4\eta mk.
\end{aligned}
$$

This implies $\ell(t+1, \mathbf{x}_i) \leq 4\eta mk$. By induction, we see that if $t < \mathcal{T}_\varepsilon$ and $\ell(t, \mathbf{x}_i) \leq 4\eta mk$, then $\ell(t+1, \mathbf{x}_i) \leq 4\eta mk$. $\qquad\square$

**Lemma C.10.** *Assume Assumption 3 holds and $\gamma + \rho < \frac{0.99(1-\varepsilon)}{4k}$. For all $t_1, t_2$ satisfying $\mathcal{T}_2 \leq t_1 \leq t_2 < \mathcal{T}_\varepsilon$ and $i \in \mathcal{S}_T$,*

$$
S_i(t_1, t_2) \leq \frac{4(t_2 - t_1)(\gamma + \rho)k + 4k + 3}{0.99(1-\varepsilon)}.
$$

*Proof.* Recall Lemma B.5 (restated in this setting, using Lemma C.9):

$$
T_i(t_1, t_2) \leq \frac{4\eta mk}{\eta} + (\gamma + \rho) B(t_1, t_2) + 3m.
$$

Using $t < \mathcal{T}_\varepsilon$ we bound

$$
\begin{aligned}
0.99(1-\varepsilon)m S_i(t_1, t_2) &\leq T_i(t_1, t_2) \\
&\leq \frac{4\eta mk}{\eta} + (\gamma + \rho) B(t_1, t_2) + 3m \\
&\leq 4mk + 4(t_2 - t_1)(\gamma + \rho)mk + 3m.
\end{aligned}
$$

From this, the desired inequality follows. $\qquad\square$

**Lemma C.11.** *Assume Assumption 3 holds and $\gamma + \rho \leq \min\left\{\frac{0.99(1-\varepsilon)}{4k}, \sqrt{\frac{0.99(1-\varepsilon)}{8(n-k)k}}\right\}$. Let $i \in \mathcal{S}_F$. Suppose there is $j \not\sim i$ such that*

$$
\langle \mathbf{w}_j^{(\mathcal{T}_2)}, \mathbf{x}_i \rangle > \eta \frac{2(n-k)(\gamma + \rho)(4k + 3)}{0.99(1-\varepsilon)}.
$$

*Then for all $t$ satisfying $\mathcal{T}_2 \leq t < \mathcal{T}_\varepsilon$, $\langle \mathbf{w}_{j'}^{(t)}, \mathbf{x}_i \rangle > 0$ for some neuron $j'$ depending on $t$.*

*Proof.* Let $\tau_0 \geq \mathcal{T}_2$ be the first iteration after $\mathcal{T}_2$ where $\ell(t, \mathbf{x}_i) = 0$. We will show by induction that for all $t$ satisfying $\mathcal{T}_2 \leq t \leq \tau_0$ that $\langle \mathbf{w}_j^{(t)}, \mathbf{x}_i \rangle > 0$. The case $t = \mathcal{T}_2$ follows immediately by the assumption of the lemma. Otherwise, assume $\langle \mathbf{w}_j^{(t')}, \mathbf{x}_i \rangle > 0$ for all $\mathcal{T}_2 \leq t' < t$. By Lemma B.2 and Lemma C.10,

$$
\begin{aligned}
\langle \mathbf{w}_j^{(t+1)}, \mathbf{x}_i \rangle &\geq \langle \mathbf{w}_j^{(\mathcal{T}_2)}, \mathbf{x}_i \rangle \\
&\quad + \eta \left( T_{ij}(\mathcal{T}_2, t+1) - G_j^{(i)}(\mathcal{T}_2, t+1)(\gamma + \rho) + B_j^{(i)}(\mathcal{T}_2, t+1)(\gamma - \rho) \right) \\
&\geq \langle \mathbf{w}_j^{(\mathcal{T}_2)}, \mathbf{x}_i \rangle + \eta(t+1 - \mathcal{T}_2) - \eta(\gamma + \rho) \sum_{i \in \mathcal{S}_T} S_i(\mathcal{T}_2, t+1) \\
&\geq \langle \mathbf{w}_j^{(\mathcal{T}_2)}, \mathbf{x}_i \rangle + \eta(t+1 - \mathcal{T}_2) \\
&\quad - \eta 2(n-k)(\gamma + \rho) \left( \frac{4(t+1-\mathcal{T}_2)(\gamma + \rho)k + 4k + 3}{0.99(1-\varepsilon)} \right) \\
&= \eta(t+1-\mathcal{T}_2) \left( 1 - \frac{8(n-k)k(\gamma + \rho)^2}{0.99(1-\varepsilon)} \right) + \langle \mathbf{w}_j^{(\mathcal{T}_2)}, \mathbf{x}_i \rangle \\
&\quad - \eta \frac{2(n-k)(\gamma + \rho)(4k + 3)}{0.99(1-\varepsilon)} \\
&> 0.
\end{aligned}
$$

We now continue the induction past $\tau_0$. If $\ell(t, \mathbf{x}_i) = 0$, then point $i$ clearly activates some neuron. Let $\tau_1$ be the first iteration after $\tau_0$ where $\ell(t, \mathbf{x}_i) > 0$. By Lemma B.3

$$
\begin{aligned}
y_i f(\tau_1, \mathbf{x}_i) - y_i f(\tau_1 - 1, \mathbf{x}_i) &= \sum_{j'=1}^{2m} -(-1)^{i+j} (\phi(\langle \mathbf{w}_j^{(\tau_1)}, \mathbf{x}_i \rangle) - \phi(\langle \mathbf{w}_j^{(\tau_1 - 1)}, \mathbf{x}_i \rangle)) \\
&\geq \sum_{j=1}^{2m} \eta(T_{ij}(t, \tau_1) - G_j(t, \tau_1 - 1)) \\
&\geq -4\eta m(n-k).
\end{aligned}
$$

This means
$$
\sum_{j' \not\sim i} (-1)^j \phi(\langle \mathbf{w}_{j'}^{(\tau_1)}, \mathbf{x}_i \rangle) \geq y_i f(\tau_1, \mathbf{x}_i) \geq 1 - 4\eta m(n-k)
$$

and there is some $j'$ satisfying

$$
\phi(\langle \mathbf{w}_{j'}^{(\tau_1)}, \mathbf{x}_i \rangle) \geq \frac{1}{m} - 4\eta(n-k) > \eta \frac{2(n-k)(\gamma + \rho)(4k+3)}{0.99(1-\varepsilon)},
$$

assuming $\eta$ is sufficiently small (Assumption 3). We can run the original induction argument with $\tau_1$ replacing $\mathcal{T}_2$ and $\tau_2 = \min\{t \geq \tau_2 : \ell(t, \mathbf{x}_i) = 0\}$ replacing $\tau_0$ to verify the conclusion for $\tau_1 \leq t \leq \tau_2$. By switching back and forth between these two arguments, we can show that point $i$ activates some neuron for all $\mathcal{T}_2 \leq t < \mathcal{T}_\varepsilon$. $\qquad \square$

**Lemma C.12.** *If Assumption 3 holds, the training process reaches loss.*

*Proof.* In this proof, let $\varepsilon = \frac{1}{5}$. The conditions of Lemma C.9, Lemma C.10, and Lemma C.11 hold because $\gamma + \rho \leq \min\left\{ \frac{0.99/5}{k}, \sqrt{\frac{0.99/5}{2k(n-k)}} \right\}$.

By Lemma C.2, there is a finite bound on the number of updates, independent of the number of iterations spent training. If we carry out the training procedure for infinitely many iterations, there must be some iteration where we make no updates. Since the training procedure is deterministic, we will not make any updates after this point, and we will have converged. It remains to show that this convergence results in zero training loss. The only way for a point to not update any neurons is for that point's loss to be zero or for that point to activate no neurons.

Lemma C.8 and Lemma C.11 say, under certain conditions, that every clean point and every corrupted point activates some neuron for each iteration $t \leq \mathcal{T}_\varepsilon$. We need only to verify that these conditions hold and that $B(\mathcal{T}_1, t)$ remains below the limitation set in Lemma C.8.

We apply Lemma C.2 starting at $t_0 = \mathcal{T}_1$. By Lemma C.7, $\ell(\mathcal{T}_1, \mathbf{x}_i) \leq \frac{1}{3}$ for all $i \in \mathcal{S}_T$. Using Lemma C.6, for $i \in \mathcal{S}_F$,

$$\ell(\mathcal{T}_1, \mathbf{x}_i) \leq 1 - y_i \sum_{j=1}^{2m} (-1)^j \phi(\langle \mathbf{w}_j^{(\mathcal{T}_1)}, \mathbf{x}_i \rangle)$$

$$\leq 1 + \sum_{j \sim i}^{2m} \phi(\langle \mathbf{w}_j^{(\mathcal{T}_1)}, \mathbf{x}_i \rangle)$$

$$\leq 1 + O(\eta).$$

With these bounds, Lemma C.2 shows that for all $t \geq \mathcal{T}_1$,

$$B(\mathcal{T}_1, t) \leq \frac{2k}{1 - 4k(n-k)(\gamma+\rho)^2} \left( \frac{1}{\eta} + 2(n-k)(\gamma+\rho)\frac{1}{3\eta} \right) + O(1)$$

$$\leq \frac{2k(1 + (2/3)(n-k)(\gamma+\rho))}{\eta(1 - 4k(n-k)(\gamma+\rho)^2)} + O(1)$$

$$\leq \frac{2k(5/3)}{\eta(1 - 4/99)} + O(1)$$

$$\leq \frac{n}{10\eta}$$

using $\gamma + \rho \leq \min\{\frac{1}{n-k}, \frac{1}{99k}\}$, $\eta$ sufficiently small, and $k \leq \frac{n}{100}$ (Assumption 3). By Lemma C.8, $\mathcal{T}_\varepsilon = \infty$ if $\frac{n}{10\eta} < \frac{5\varepsilon n}{8\eta}$, which is clearly true for $\varepsilon = \frac{1}{5}$.

We now show that every training point $i$ activates at least one neuron each iteration. By Lemma C.9, this is true if $i \in \mathcal{S}_T$. By Lemma C.11, this is true for $i \in \mathcal{S}_F$ if there is a neuron $j \not\sim i$ such that $\langle \mathbf{w}_j^{(\mathcal{T}_2)}, \mathbf{x}_i \rangle > \eta \frac{2(n-k)(\gamma+\rho)(4k+3)}{0.99(1-\varepsilon)}$. Fix $i \in \mathcal{S}_F$.

First, assume that $\ell(t, \mathbf{x}_i) > 0$ for all $t < \mathcal{T}_2$. By Assumption 3 and Lemma C.3, we know there is $j \in \Gamma_{y_i}$ such that $i \in \mathcal{A}_j^{(1)}$. Using Lemma B.2 and Lemma C.4 we can bound

$$\langle \mathbf{w}_j^{(\mathcal{T}_1)}, \mathbf{x}_i \rangle \geq \langle \mathbf{w}_j^{(1)}, \mathbf{x}_i \rangle + \eta T_{ij}(1, \mathcal{T}_1) - \eta(\gamma+\rho)G_j^{(i)}(1, \mathcal{T}_1) + \eta(\gamma-\rho)B_j^{(i)}(1, \mathcal{T}_1)$$

$$\geq \eta(1 - (\gamma+\rho)(n-k))(\mathcal{T}_1 - 1)$$

and

$$\langle \mathbf{w}_j^{(\mathcal{T}_2)}, \mathbf{x}_i \rangle \geq \langle \mathbf{w}_j^{(\mathcal{T}_1)}, \mathbf{x}_i \rangle + \eta T_{ij}(\mathcal{T}_1, \mathcal{T}_2) - \eta(\gamma+\rho)G_j^{(i)}(\mathcal{T}_1, \mathcal{T}_2) + \eta(\gamma-\rho)B_j^{(i)}(\mathcal{T}_1, \mathcal{T}_2).$$

By induction on $t$ we see that

$$G_j(\mathcal{T}_1, t) \leq \max_{\mathcal{T}_1 \leq t' < t} \left( (n-k)(t+1-t') + \frac{(\gamma+\rho)B_j(\mathcal{T}_1, t')}{\gamma-\rho} \right)$$

for $t > \mathcal{T}_1$. The base case $t = \mathcal{T}_1 + 1$ is clear. Suppose the inequality holds for $t$. Either $G_j(\mathcal{T}_1, t)$ increases by at most $n - k$ or there is some $i' \in \mathcal{S}_T \cap \mathcal{A}_j^{(t+1)}$ with $i' \not\sim j$. By Lemma B.2 and Lemma C.7,

$$0 < \langle \mathbf{w}_j^{(t)}, \mathbf{x}_{i'} \rangle \leq \langle \mathbf{w}_j^{(\mathcal{T}_1)}, \mathbf{x}_{i'} \rangle - \eta(\gamma-\rho)G_j(\mathcal{T}_1, t+1) + \eta(\gamma+\rho)B_j(\mathcal{T}_1, t)$$

$$\leq -\eta(\gamma-\rho)G_j(\mathcal{T}_1, t) + \eta(\gamma+\rho)B_j(\mathcal{T}_1, t)$$

from which the inequality follows. Since $\frac{(\gamma+\rho)B_j(\mathcal{T}_1, t')}{\gamma-\rho} \leq 3k(t' - \mathcal{T}_1) < (n-k)(t' - \mathcal{T}_1)$ (using $\rho < \frac{\gamma}{5}$ and $k \leq \frac{n}{100}$ from Assumption 3), this maximum occurs at $\tau = \mathcal{T}_1$. This yields

$$\langle \mathbf{w}_j^{(\mathcal{T}_2)}, \mathbf{x}_i \rangle \geq \eta(1 - (\gamma+\rho)(n-k))(\mathcal{T}_2 + 1 - 1)\eta(\gamma-\rho)B_j^{(i)}(\mathcal{T}_1, \mathcal{T}_2)$$

$$\geq \eta(1 - (\gamma+\rho)(n-k))\mathcal{T}_2$$

We want to show this bound is larger than a quantity that is $O(\eta)$. This happens when both of the following hold:

$$O(1) \leq (1 - (\gamma + \rho)(n - k))\mathcal{T}_2,$$

which holds when $\eta$ is sufficiently small.

Now suppose $\ell(\tau, \mathbf{x}_i) = 0$ for some iteration $\tau \leq \mathcal{T}_2$. By Lemma C.7, $\mathcal{T}_1 < \tau$. In this case, we see that

$$
\begin{aligned}
y_i f(\mathcal{T}_2, \mathbf{x}_i) &= y_i f(\mathcal{T}_2, \mathbf{x}_i) + \sum_{j=1}^{2m} -(-1)^{i+j}[\phi(\langle \mathbf{w}_j^{(\mathcal{T}_1)}, \mathbf{x}_i \rangle) - \phi(\langle \mathbf{w}_j^{(\tau)}, \mathbf{x}_i \rangle)] \\
&\geq 1 - \eta(\gamma + \rho)G(\tau, \mathcal{T}_2) \\
&\geq 1 - (\gamma + \rho)\frac{2(n - k)}{1 - 4k(n - k)(\gamma + \rho)^2}\left[\frac{1}{3} + 2k(\gamma + \rho)\right] + O(\eta) \\
&\geq 1 - \frac{2}{1 - 4/99}\left(\frac{1}{3} + \frac{2}{99}\right) + O(\eta) \geq \frac{1}{4}
\end{aligned}
$$

where in the third line we use Lemma C.2 and the fourth line we use $\gamma + \rho \leq \min\{\frac{1}{n-k}, \frac{1}{99k}\}$ and $\eta$ sufficiently small (Assumption 3). Sine this is positive, there is some neuron $j$ with $i \not\sim j$ such that $\phi(\langle \mathbf{w}_j^{(\mathcal{T}_2)}, \mathbf{x}_i \rangle)$ is at least $\frac{1}{m}$ this bound. This is an $\Omega(1)$ lower bound. Since the required condition is $\langle \mathbf{w}_j^{(\mathcal{T}_2)}, \mathbf{x}_i \rangle > O(\eta)$, this can be achieved by taking $\eta$ sufficiently small. $\qquad\square$

## C.5  Proof of Lemma 3.5

**Lemma C.13** (Lemma 3.5). *Assume Assumption 3 holds. Let $y \in \{-1, 1\}$ chosen uniformly and $\mathbf{x} := y\sqrt{\gamma}\mathbf{v} + \sqrt{1 - \gamma}\mathbf{n}$, where $\mathbf{n} \sim Uniform(\mathcal{S}^{d-1} \cap \text{span}\{\mathbf{v}\}^\perp)$. Suppose that $|\langle \mathbf{n}, \mathbf{n}_\ell \rangle| < \frac{\rho}{1-\gamma}$ for all $l \in [2n]$, then $yf(\mathcal{T}_{end}, \mathbf{x}) > 0$.*

*Proof.* Following the same steps as in (5) for any $j \in [2m]$

$$
\begin{aligned}
\langle \mathbf{w}_j^{(\mathcal{T}_{end})}, \mathbf{x} \rangle &= \langle \mathbf{w}_j^{(1)}, \mathbf{x}_i \rangle + (-1)^j \eta \sum_{\ell=1}^{2n} T_{\ell j}(1, \mathcal{T}_{end})y_\ell \langle \mathbf{x}_\ell, \mathbf{x}_i \rangle \\
&= \langle \mathbf{w}_j^{(t_0)}, \mathbf{x}_i \rangle + (-1)^j y\eta \sum_{\ell=1}^{2n} T_{\ell j}(t_0, t)(-1)^\ell y\beta(\ell)\langle \mathbf{x}_\ell, \mathbf{x} \rangle \\
&= \langle \mathbf{w}_j^{(t_0)}, \mathbf{x}_i \rangle + (-1)^j y\eta \sum_{\ell=1}^{2n} T_{\ell j}(t_0, t)\lambda'_{i\ell},
\end{aligned}
$$

where $\lambda'_\ell := (-1)^l y\beta(\ell)\langle \mathbf{x}_\ell, \mathbf{x} \rangle = \beta(\ell)\gamma + (1 - \gamma)\langle \mathbf{n}_\ell, \mathbf{n} \rangle$. Then as in Lemma B.1

$$
\begin{aligned}
\gamma - \rho \leq &\lambda'_\ell \leq \gamma + \rho, \text{ if } i \in \mathcal{S}_T, \\
-(\gamma + \rho) \leq &\lambda'_\ell \leq -(\gamma - \rho), \text{ if } i \in \mathcal{S}_F,.
\end{aligned}
$$

Recall, from Lemma C.4 for any $j \in \Gamma_p$ then $G_j(1, \mathcal{T}_{end}) \geq G_j(1, \mathcal{T}_1) = \mathcal{T}_1(n - k)$. As a consequence, for $j \in \Gamma_p$ we have

$$
\begin{aligned}
\langle \mathbf{w}_j^{(\mathcal{T}_{end})}, \mathbf{x} \rangle &\geq \langle \mathbf{w}_j^{(1)}, \mathbf{x}_i \rangle + \eta G_j(1, \mathcal{T}_{end})(\gamma - \rho) - \eta B_j(1, \mathcal{T}_{end})(\gamma + \rho) \\
&\geq O(\eta) + \mathcal{T}_1(n - k)(\gamma - \rho) - \eta B_j(1, \mathcal{T}_{end})(\gamma + \rho).
\end{aligned}
$$

For $j$ such that $(-1)^j = y$ then

$$
\begin{aligned}
\phi(\langle \mathbf{w}_j^{(\mathcal{T}_{end})}, \mathbf{x} \rangle) &\leq \phi(\langle \mathbf{w}_j^{(1)}, \mathbf{x}_i \rangle - \eta G_j(1, \mathcal{T}_{end})(\gamma - \rho) + \eta B_j(1, \mathcal{T}_{end})(\gamma + \rho)) \\
&\leq O(\eta) + \eta B_j(1, \mathcal{T}_{end})(\gamma + \rho).
\end{aligned}
$$

As a result

$$yf(\mathcal{T}_{\text{end}}, \mathbf{x}) = \sum_{j \in [2m]} \phi(\langle \mathbf{w}_j^{(\mathcal{T}_{\text{end}})}, \mathbf{x} \rangle)$$

$$\geq \sum_{j \in \Gamma_p} \langle \mathbf{w}_j^{(\mathcal{T}_{\text{end}})}, \mathbf{x} \rangle - \sum_{j \,:\, (-1)^j \neq y} \phi(\langle \mathbf{w}_j^{(\mathcal{T}_{\text{end}})}, \mathbf{x} \rangle)$$

$$\geq \eta \sum_{j \in \Gamma_p} \mathcal{T}_1(n-k)(\gamma - \rho) - \eta \sum_{j \in [2n]} B_j(1, \mathcal{T}_{\text{end}})(\gamma + \rho) + O(\eta)$$

$$\geq 0.99 \eta m \mathcal{T}_1 (n-k) - \eta B(1, \mathcal{T}_{\text{end}})(\gamma + \rho)) + O(\eta)$$

using $|\Gamma_p| \geq 0.99$ (Assumption 3). From Lemma C.7

$$\mathcal{T}_1 = \frac{1}{1.03 \eta m [1 + (\gamma + \rho)(n-k)]} + O(1),$$

furthermore, combining the assumptions $100k < n$, $\eta$ sufficiently small, and $\gamma + \rho < \frac{1}{n-k}$ with Lemma C.2 we see

$$B(1, \mathcal{T}_{\text{end}}) \leq \frac{2k}{1 - 4k(n-k)(\gamma + \rho)^2} \left[ \frac{1}{\eta} + 2(n-k)(\gamma + \rho) \left( \frac{1}{\eta} \right) \right] + O(n)$$

$$\leq \frac{n}{10\eta}.$$

Here we also use that $\ell(0, \mathbf{x}_i) \leq 1 + m\lambda_w = 1 + O(\eta)$ for all $i$.

Combining these inequalities it follows that

$$yf(\mathcal{T}_{\text{end}}, \mathbf{x}) \geq \frac{0.99}{1.03} \cdot \frac{n-k}{2}(\gamma - \rho) - \frac{n}{10}(\gamma + \rho) + O(\eta)$$

$$\geq \frac{n}{3} - \frac{3n}{25} + O(\eta) > 0,$$

again using $100k < n$, $\eta$ sufficiently small, and $\gamma + \rho < \frac{1}{n-k}$. $\qquad \square$

## C.6 Proof of Theorem 3.1

**Theorem C.14** (Theorem 3.1). *Let Assumption 2 hold with $\rho = \gamma/5$. There exists a sufficiently small step-size $\eta$ such that with probability at least $1 - \delta$ over the randomness of the dataset and network initialization the following hold.*

1. *There exists a positive constant $C$ such that the training process terminates at an iteration $\mathcal{T}_{end} \leq \frac{Cn}{\eta}$.*

2. *For all $i \in [2n]$ then $\ell(\mathcal{T}_{end}, \mathbf{x}_i) = 0$.*

3. *There exists a positive constant $c$ such that the generalization error satisfies*
$$\mathbb{P}(\text{sgn}(f(\mathcal{T}_{end}, \mathbf{x})) \neq y) \leq \exp\left(-cd\gamma^2\right).$$

*Proof.* Under Assumption 3 Statement 1 and 2 follow from Lemma C.12, Note the bound in Statement 1 comes from Lemma C.2 applied from iteration 0 to iteration $\mathcal{T}_{\text{end}}$, using $4k(n-k)(\gamma + \rho)^2 < 4/99$ and $\ell(0, \mathbf{x}_i) = 1 + O(\eta)$ for all $i \in [2n]$. With regards to Statement 3, from Lemma C.13 if $|\langle \mathbf{n}, \mathbf{n}_\ell \rangle| < \frac{\rho}{1-\gamma}$ for all $l \in [2n]$ it follows that $\text{sgn}(f(\mathcal{T}_{\text{end}}, \mathbf{x})) = y$. Therefore, as $\frac{\rho}{1-\rho} > \rho$ and analogous to Lemma A.1, under Assumption 3 there exists a positive constant $c$ such that

$$\mathbb{P}(\text{sgn}(f(\mathcal{T}_{\text{end}}, \mathbf{x})) \neq y) \leq \mathbb{P}\left( \bigcup_{l=1}^{2n} \left( |\langle \mathbf{n}, \mathbf{n}_\ell \rangle| \geq \frac{\rho}{1-\gamma} \right) \right)$$

$$\leq 2n \mathbb{P}(\mathbf{n} \in \text{Cap}(\mathbf{e}_1, \gamma/5))$$

$$\leq 2n \exp\left( -\frac{d\gamma^2}{15} \right)$$

$$\leq \exp\left(-cd\gamma^2\right).$$

Finally, under Assumption 2 then Assumption 3 holds with probability at least $1 - \delta$ by Lemma C.1. $\qquad \square$

# Appendix D  Non-benign overfitting

**Assumption 4.** *With $\delta, \rho \in (0,1)$ we assume the following conditions on the data and model hyperparameters.*

    *1. $n \geq 1$,*

    *2. $m \geq \log_2(\frac{4n}{\delta})$,*

    *3. $d \geq \max\left\{3, 3\rho^{-2} \ln\left(\frac{4n^2}{\delta}\right)\right\}$,*

    *4. $k \geq 0$,*

    *5. $\gamma \leq \frac{1}{6\sqrt{dn}}$,*

    *6. $\eta < \frac{1}{2mn}$,*

    *7. $\lambda_w < \eta$.*

In our analysis we require two additional assumptions on the training sample and activations at initialization.

**Assumption 5.** *Let $\rho \in (0,1)$ satisfy $\rho \leq \min\{\frac{1-\gamma}{4n}, \frac{1}{2n-1} - \gamma\}$ and in addition to the conditions detailed in Assumption 5, assume the following two conditions hold.*

    *1. For all $i \in [2n]$ there exists a $j \in [2m]$ such that $(-1)^j = y_i$ and $i \in \mathcal{A}_j^{(0)}$.*

    *2. For all $i, l \in [2n]$, $i \neq l$ $|\langle \mathbf{n}_i, \mathbf{n}_l \rangle| \leq \frac{\rho}{1-\gamma}$.*

Note under these assumptions that $\gamma < \frac{1}{24n^2}$, this implies $\frac{1}{2n-1} > \gamma$ and $\rho \leq \min\{\frac{1-\gamma}{4n}, \frac{1}{2n-1} - \gamma\}$ if $\rho \leq \frac{1}{5n}$. As demonstrated in the following Lemma, these additional two conditions hold with high probability over the randomness of the initialization and training set.

**Lemma D.1.** *The additional conditions of Assumption 5 hold with probability at least $1 - \delta$.*

*Proof.* Using Lemma A.7, then as long as $m \geq \log_2(\frac{4n}{\delta})$ the probability the first condition does not hold is at most $\delta/2$. Using Lemma A.1, and observing $\frac{\rho}{1-\gamma} > \rho$, then as long as

$$d \geq \max\left\{3, 3\rho^{-2} \ln\left(\frac{4n^2}{\delta}\right)\right\}$$

the probability that the second condition does not hold is also at most $\delta/2$. Using the union bound we conclude that both properties hold with probability at least $\delta$. $\qquad\square$

## D.1  Proof of Lemma 3.7

**Lemma D.2** (Lemma 3.7). *In addition to Assumption 5, assume also that $\ell(t_0, \mathbf{x}_i) \leq a$ for all $i \in [2n]$. Then*

$$T(t_0, t) \leq \frac{2n}{1 - (2n-1)(\gamma + \rho)}\left(\frac{a}{\eta} + 3m\right).$$

*Proof.* From Lemma B.5, $\phi(\rho - \gamma) \leq \rho + \gamma$, and the assumption on $a$,

$$T_i(t_0, t) \leq \frac{a}{\eta} + 3m + (\gamma + \rho)T^{(i)}(t_0, t).$$

If we sum over $i$, we get

$$T(t_0, t) \leq \frac{2na}{\eta} + 6mn + (2n-1)(\gamma + \rho)T(t_0, t),$$

from which the result follows. $\qquad\square$

## D.2 Supporting lemmas

**Lemma D.3.** *If Assumption 5 holds, then the training process converges to zero loss.*

*Proof.* By Lemma D.2, there is an upper bound on the number of updates independent of iteration. This can only happen if there is some iteration after which we make no updates. In turn, this can only happen if every point is either at zero loss or activates no neurons. We prove by induction that every point activates a neuron each iteration $t = 0$. Consider an arbitrary point $i \in [2n]$, at $t = 0$ the induction hypothesis is true by Statement 1 of Assumption 5. Suppose the induction hypothesis is true at iteration $t$, we consider the following two cases separately in order to show the induction hypothesis also holds at iteration $t + 1$.

1. If $\ell(t, \mathbf{x}_i) > 0$, then by assumption we can choose a $j \in [2m]$ such that $\phi(\langle \mathbf{w}_j^{(t)}, \mathbf{x}_i \rangle) > 0$. We bound

$$
\begin{aligned}
\phi(\langle \mathbf{w}_j^{(t+1)}, \mathbf{x}_i \rangle) &\geq \phi(\langle \mathbf{w}_j^{(t)}, \mathbf{x}_i \rangle) + \eta - \eta(\gamma + \rho)T_j^{(i)}(t, t+1) \\
&\geq \phi(\langle \mathbf{w}_j^{(t)}, \mathbf{x}_i \rangle) + \eta[1 - (\gamma + \rho)(2n - 1)] \\
&> 0,
\end{aligned}
$$

which follows as $\gamma + \rho < \frac{1}{2n-1}$ (Assumption 5).

2. If $\ell(t, \mathbf{x}_i) = 0$, then

$$
\begin{aligned}
y_i f(t, \mathbf{x}_i) = y_i \sum_{j=1}^{2m} (-1)^j \phi(\langle \mathbf{w}_j^{(t)}, \mathbf{x}_i \rangle) \\
\leq \sum_{j : (-1)^j = y_i} \phi(\langle \mathbf{w}_j^{(t)}, \mathbf{x}_i \rangle)
\end{aligned}
$$

is bounded below by 1. This means that there is some $j$ such that $\phi(\langle \mathbf{w}_j^{(t)}, \mathbf{x}_i \rangle) \geq \frac{1}{m}$. We bound

$$
\begin{aligned}
\phi(\langle \mathbf{w}_j^{(t+1)}, \mathbf{x}_i \rangle) &\geq \phi(\langle \mathbf{w}_j^{(t)}, \mathbf{x}_i \rangle) - \eta(\gamma + \rho)T_j^{(i)}(t, t+1) \\
&\geq \frac{1}{m} - \eta(\gamma + \rho)(2n - 1) \\
&> 0
\end{aligned}
$$

as $\eta < \frac{1}{2mn}$ (Assumption 5). $\qquad\square$

**Lemma D.4.** *Suppose that at epoch $\tau$ every point is at zero loss. Then*

$$
\eta T(0, \tau) \geq n + O(\eta).
$$

*Proof.* If $\ell(\tau, \mathbf{x}_i) = 0$ for all $i \in [2n]$, then $y_i f(\tau, \mathbf{x}_i) \geq 1$ for all $i \in [2n]$. We bound

$$
\begin{aligned}
y_i f(\tau, \mathbf{x}_i) &\leq \sum_{j : (-1)^j = y_i} (\phi(\langle \mathbf{w}_j^{(\tau)}, \mathbf{x}_i \rangle) - \phi(\langle \mathbf{w}_j^{(0)}, \mathbf{x}_i \rangle)) + O(\eta) \\
&\leq \sum_{j : (-1)^j = y_i} \eta[T_{ij}(0, \tau) + (\gamma + \rho)T_j^{(i)}(0, \tau)] + O(\eta) \\
&\leq \eta T_i(0, \tau) + \eta(\gamma + \rho)T^{(i)}(0, \tau) + O(\eta).
\end{aligned}
$$

Summing over $i \in [2n]$ we see that

$$
2n \leq \eta[1 + (2n - 1)(\gamma + \rho)]T(0, \tau) \leq 2\eta T(0, \tau) + O(\eta),
$$

from which the result claimed follows. $\qquad\square$

**Lemma D.5.** *Let $y \sim U(\{-1, 1\})$ and consider a clean test point $\mathbf{x} = y(\sqrt{\gamma}\mathbf{v} + \sqrt{1 - \gamma}\mathbf{n})$, where $\mathbf{n} \sim U(\mathbb{S}^d \cap \mathrm{span}(\mathbf{v})^\perp)$. If Assumption 5 holds, then*

$$
\mathbb{P}(y f(\mathcal{T}_{end}, \mathbf{x}) < 0) \geq \frac{1}{8}.
$$

*Proof.* Observe by symmetry of the distributions of both $y$ and $\mathbf{n}$ that $-\mathbf{x}$ is identically distributed to $\mathbf{x}$ and furthermore that the labels of $\mathbf{x}$ and $-\mathbf{x}$ are opposite. As a result, if $y(f(\mathcal{T}_{\text{end}}, \mathbf{x}) - f(\mathcal{T}_{\text{end}}, -\mathbf{x})) < 0$ then at least one of $y(f(\mathcal{T}_{\text{end}}, \mathbf{x}) < 0$ or $-y(f(\mathcal{T}_{\text{end}}, -\mathbf{x}) < 0$, in turn implying at least one of them is misclassified. By construction, $\langle \mathbf{w}_j^{(t)}, \mathbf{x} \rangle > 0$ iff $\langle \mathbf{w}_j^{(t)}, -\mathbf{x} \rangle < 0$, therefore

$$y(f(\mathcal{T}_{\text{end}}, \mathbf{x}) - f(\mathcal{T}_{\text{end}}, -\mathbf{x})) = y \sum_{j=1}^{2m} (-1)^j \left( \phi(\langle \mathbf{w}_j^{(\mathcal{T}_{\text{end}})}, \mathbf{x} \rangle) - \phi(\langle \mathbf{w}_j^{(\mathcal{T}_{\text{end}})}, -\mathbf{x}_i \rangle) \right)$$

$$= \sum_{j=1}^{2m} y(-1)^j \langle \mathbf{w}_j^{(\mathcal{T}_{\text{end}})}, \mathbf{x} \rangle.$$

Unwinding the GD update to a neuron we have

$$\mathbf{w}_j^{(\mathcal{T}_{\text{end}})} = \mathbf{w}_j^{(0)} + \eta \sum_{i=1}^{2n} T_{ij}(0, \mathcal{T}_{\text{end}})(-1)^{i+j} \mathbf{x}_i.$$

Furthermore, as

$$\langle \mathbf{x}_i, \mathbf{x} \rangle = y(-1)^i (\gamma + (1-\gamma)\beta(i)\langle \mathbf{n}_i, \mathbf{n} \rangle))$$

then

$$\sum_{j=1}^{2m} y(-1)^j \langle \mathbf{w}_j^{(\mathcal{T}_{\text{end}})}, \mathbf{x} \rangle = \sum_{j=1}^{2m} y(-1)^j \langle \mathbf{w}_j^{(0)}, \mathbf{x} \rangle + \eta \sum_{j=1}^{2m} \sum_{i=1}^{2n} y(-1)^j T_{ij}(0, \mathcal{T}_{\text{end}})(-1)^{i+j} \langle \mathbf{x}_i, \mathbf{x} \rangle$$

$$\leq 2m\lambda_w + \eta \sum_{i=1}^{2n} T_i(0, \mathcal{T}_{\text{end}})(\gamma + (1-\gamma)\beta(i)\langle \mathbf{n}_i, \mathbf{n} \rangle))$$

$$= 2m\lambda_w + \eta \left( T(0, \mathcal{T}_{\text{end}})\gamma + \left\langle \mathbf{n}, (1-\gamma) \sum_{i=1}^{2n} T_i(0, \mathcal{T}_{\text{end}})\beta(i)\mathbf{n}_i \right\rangle \right)$$

$$\stackrel{d}{=} 2m\lambda_w + \eta \left( T(0, \mathcal{T}_{\text{end}})\gamma - \|\mathbf{z}\| (1-\gamma) \langle \mathbf{n}, \mathbf{u} \rangle \right),$$

where the final equality follows from symmetry of the noise distribution, $\mathbf{z} := \sum_{i=1}^{2n} T_i(0, \mathcal{T}_{\text{end}})\mathbf{n}_i$ and $\mathbf{u} = \frac{\mathbf{z}}{\|\mathbf{z}\|}$. Observe

$$\|\mathbf{z}\|^2 = \sum_{i,l=1}^{2n} T_i(0, \mathcal{T}_{\text{end}})T_\ell(0, \mathcal{T}_{\text{end}})\langle \mathbf{n}_i, \mathbf{n}_\ell \rangle$$

$$\geq \sum_i^{2n} T_i^2(0, \mathcal{T}_{\text{end}}) - \frac{\rho}{1-\gamma} \sum_{i \neq \ell} T_i(0, \mathcal{T}_{\text{end}})T_\ell(0, \mathcal{T}_{\text{end}})$$

$$= \frac{1-\gamma+\rho}{1-\gamma} \sum_i^{2n} T_i^2(0, \mathcal{T}_{\text{end}}) - \frac{\rho}{1-\gamma} T^2(0, \mathcal{T}_{\text{end}})$$

$$\geq \left( \frac{1}{2n} - \frac{\rho}{1-\gamma} \right) T^2(0, \mathcal{T}_{\text{end}}).$$

where the final inequality follows from Jensen's inequality. By assumption $4n\rho < 1 - \gamma$, and $10m\lambda_w \leq n\gamma \leq \frac{\sqrt{n}}{6\sqrt{d}}$, furthermore trivially $(1-\gamma) > 0.8$. Conditioning on the event $\langle \mathbf{n}, \mathbf{u} \rangle > 0$, which holds with probability $1/2$, these inequalities in combination with Lemma D.4 give

$$\sum_{j=1}^{2m} y(-1)^j \langle \mathbf{w}_j^{(\mathcal{T}_{\text{end}})}, \mathbf{x} \rangle \leq 2m\lambda_w + n\gamma - \frac{\sqrt{n}}{2}(1-\gamma) \langle \mathbf{n}, \mathbf{u} \rangle$$

$$\leq \frac{1}{5}\sqrt{\frac{n}{d}} - \frac{4}{10}\sqrt{n}\langle \mathbf{n}, \mathbf{u} \rangle.$$

Therefore, if $\langle \mathbf{n}, \mathbf{u} \rangle > 0$ then the condition

$$\langle \sqrt{d}\mathbf{n}, \mathbf{u} \rangle \geq \frac{1}{2} \tag{8}$$

implies at least one of $\mathbf{x}$ or $-\mathbf{x}$ is misclassified. Suppose $\mathbf{n} \sim U(\mathbb{S}^d \cap \mathrm{span}(\mathbf{v})^\perp)$ is such that (8) holds. Then as $y \sim U(\{-1, 1\})$ it follows given $\mathbf{n}$ that either $\mathbf{x}$ or $-\mathbf{x}$ are sampled each with equal probability and thus the chance of misclassifying is at least $1/2$. As a result, the probability of misclassification is at least

$$\frac{1}{4} \mathbb{P}\left( \langle \sqrt{d}\mathbf{n}, \mathbf{u} \rangle \geq \frac{1}{2} \right) \geq \frac{1}{8}$$

as claimed. We note the final inequality above follows by showing that the two spherical caps corresponding to the set of unit vectors $\mathbf{z}$ satisfying $\langle \mathbf{n}, \mathbf{u} \rangle \geq 1/(2\sqrt{d})$ account for less than half the area of $\mathbb{S}^{d-1}$, which can be derived from formulas provided in (S, 2011). This inequality has also appeared for instance in (Asi & Duchi, 2019). $\qquad\square$

### D.3 Proof of Theorem 3.6

**Theorem D.6** (Theorem 3.6). *Assume Assumption 4 holds with $\rho = \frac{1}{5n}$. With probability at least $1 - \delta$ over the randomness of the dataset and network initialization the following hold.*

1. *There exists a positive constant $C$ such that the training process terminates at an iteration $\mathcal{T}_{end} \leq \frac{Cn}{\eta}$.*

2. *For all $i \in [2n]$ $\ell(\mathcal{T}_{end}, \mathbf{x}_i) = 0$.*

3. *The generalization error satisfies*

$$\mathbb{P}(\mathrm{sgn}(f(\mathcal{T}_{end}, \mathbf{x})) \neq y) \geq \frac{1}{8}.$$

*Proof.* Under Assumption 5 Statement 1 and 2 come from Lemma D.3. The bound on $\mathcal{T}_{end}$ comes from Lemma D.2 applied between iterations 0 and $\mathcal{T}_{end}$, using $\ell(0, \mathbf{x}_i) = 1 + O(\eta)$ for all $i \in [2n]$ and $(\gamma + \rho)(2n - 1) < \frac{2}{3}$. Statement 3 follows from Lemma D.5. We conclude by observing under Assumption 4 that Assumption 5 holds with probability at least $1 - \delta$. $\qquad\square$

## Appendix E   No-overfitting

**Assumption 6.** *With $\delta, \rho \in (0, 1)$ and $C$ a generic, positive constant, we assume the following conditions on the data and model hyperparameters.*

1. $n \geq C \log\left(\frac{2m}{\delta}\right)$,

2. $m \geq 2$

3. $d \geq \left\{ 3, 3\rho^{-2} \ln\left(\frac{6n^2}{\delta}\right) \right\}$,

4. $k < \frac{n}{100}$,

5. $\frac{3}{n} < \gamma < \frac{1}{36} \min\{k^{-1}, 1\}$,

6. $\lambda_w < \eta$.

For our analysis we make two additional assumptions.

**Assumption 7.** *Let $\rho \in (0, 1)$ satisfy $\gamma \geq 5\rho$. In addition to the assumptions detailed in Assumption 6, suppose the following conditions hold.*

1. $\Gamma = [2m]$.

2. *For all $i, l \in [2n]$ such that $i \neq l$ then $|\langle \mathbf{n}_i, \mathbf{n}_l \rangle| \leq \frac{\rho}{1-\gamma}$.*

We remark under these assumptions that for sufficiently large $n$ then $\rho$ satisfies the inequality $\rho < \min\left\{ \frac{\gamma(n-3k)-2}{n+k}, \frac{\gamma}{5}, \frac{n}{11} \right\}$. As shown in the following lemma, these two additional conditions hold with high probability over the randomness of the initialization and training set.

**Lemma E.1.** *There exists a positive constant $C$ such that if $n \geq C \log\left(\frac{2m}{\delta}\right)$ then the extra conditions of Assumption 7 hold with probability at least $1 - \delta$.*

*Proof.* Using Lemma A.3, there exists a positive constant $C$ such that for $n \geq C$ there in turn exists a constant $c$ such that the probability the first condition does not hold is at most $m \exp(-cn)$. Setting $\delta \geq 2m \exp(-cn)$ and rearranging, as long as $n \geq C \log\left(\frac{2m}{\delta}\right)$, then the probability the first condition does not hold is at most $\delta/2$. Using Lemma A.1 and observing $\frac{\rho}{1-\gamma} > \rho$, then under the condition on $d$ stated in Assumption 7, the probability that the second condition does not hold is also at most $\frac{\delta}{2}$. Using the union bound, we therefore conclude that both properties hold with probability at least $\delta$. $\qquad\square$

### E.1 Proof of Lemma 3.9

**Lemma E.2** (Lemma 3.9). *Suppose Assumption 7 holds. Consider an arbitrary $j \in [2m]$ and iteration $t$ satisfying $2 \leq t < \mathcal{T}_0$. Then $i \in \mathcal{A}_j^{(t)}$ iff $i \sim j$.*

*Proof.* First we establish at iteration $t = 1$ that for all $i \in \mathcal{S}_T$, $i \in \mathcal{A}_j^{(t)}$ iff $i \sim j$. The argument here is similar to that of Lemma C.3. Suppose $i \sim j$ and $i \in \mathcal{S}_T$. By Assumption 7 all neurons are in $\Gamma$, therefore from the definition of $\Gamma_p$ for all $j \in [2m]$ we have $G_j^{(i)}(0,1)(\gamma - \rho) - B_j^{(i)}(0,1)(\gamma + \rho) \geq \frac{2\lambda_w}{\eta}$. Using Lemma B.2

$$\langle \mathbf{w}_j^{(1)}, \mathbf{x}_i \rangle \geq \langle \mathbf{w}_j^{(0)}, \mathbf{x}_i \rangle + \eta \left( T_{ij}(0,1) + G_j^{(i)}(0,1)(\gamma - \rho) - B_j^{(i)}(0,1)(\gamma + \rho) \right)$$
$$> \langle \mathbf{w}_j^{(0)}, \mathbf{x}_i \rangle + \eta \left( G_j(0,1)(\gamma - \rho) - B_j(0,1)(\gamma + \rho) \right)$$
$$\geq \langle \mathbf{w}_j^{(0)}, \mathbf{x}_i \rangle + 2\lambda_w$$
$$> \lambda_w.$$

On the other hand, if $i \nsim j$ then again from Lemma B.2

$$\langle \mathbf{w}_j^{(1)}, \mathbf{x}_i \rangle \leq \langle \mathbf{w}_j^{(0)}, \mathbf{x}_i \rangle - \eta \left( T_{ij}(0,1) + G_j^{(i)}(0,1)(\gamma - \rho) - B_j^{(i)}(0,1)(\gamma + \rho) \right)$$
$$\leq \langle \mathbf{w}_j^{(0)}, \mathbf{x}_i \rangle - \eta \left( G_j(0,1)(\gamma - \rho) - B_j(0,1)(\gamma + \rho) \right)$$
$$\leq -\lambda_w.$$

Now we consider $t = 2$. If $i \in \mathcal{S}_T$ and $i \sim j$ then

$$\langle \mathbf{w}_j^{(2)}, \mathbf{x}_i \rangle \geq \langle \mathbf{w}_j^{(1)}, \mathbf{x}_i \rangle + \eta \left( T_{ij}(1,2) + G_j^{(i)}(1,2)(\gamma - \rho) - B_j^{(i)}(1,2)(\gamma + \rho) \right)$$
$$> \eta \left( 1 + (n - k - 1)(\gamma - \rho) - 2k(\gamma + \rho) \right)$$

whereas if $i \in \mathcal{S}_T$ and $i \nsim j$ then

$$\langle \mathbf{w}_j^{(2)}, \mathbf{x}_i \rangle \leq \langle \mathbf{w}_j^{(1)}, \mathbf{x}_i \rangle - \eta \left( T_{ij}(1,2) + G_j^{(i)}(1,2)(\gamma - \rho) - B_j^{(i)}(1,2)(\gamma + \rho) \right)$$
$$\leq -\eta((n - k)(\gamma - \rho) - 2k(\gamma + \rho)).$$

By assumption $\gamma > \frac{2 + (n+k)\rho}{n - 3k} > \frac{(n+k)\rho}{n-3k}$ and therefore for $i \in \mathcal{S}_T$ then $i \in \mathcal{A}_j^{(2)}$ iff $i \sim j$. Again using Lemma B.2, for $i \in \mathcal{S}_F$ and $i \sim j$

$$\langle \mathbf{w}_j^{(1)}, \mathbf{x}_i \rangle \geq \langle \mathbf{w}_j^{(0)}, \mathbf{x}_i \rangle - \eta \left( T_{ij}(0,1) - G_j^{(i)}(0,1)(\gamma - \rho) + B_j^{(i)}(0,1)(\gamma + \rho) \right)$$
$$> -\lambda_w - \eta \left( 1 - \frac{2\lambda_w}{\eta} \right)$$
$$> -\eta,$$

and for $i \in \mathcal{S}_F$ and $i \nsim j$

$$\langle \mathbf{w}_j^{(1)}, \mathbf{x}_i \rangle \leq \langle \mathbf{w}_j^{(0)}, \mathbf{x}_i \rangle + \eta \left( T_{ij}(0,1) - G_j^{(i)}(0,1)(\gamma - \rho) + B_j^{(i)}(0,1)(\gamma + \rho) \right)$$
$$< \lambda_w + \eta \left( 1 - \frac{2\lambda_w}{\eta} \right)$$
$$< \eta.$$

Therefore, as $\gamma > \frac{2+(n+k)\rho}{n-3k}$ then for $i \in \mathcal{S}_F$ and $i \sim j$

$$\langle \mathbf{w}_j^{(2)}, \mathbf{x}_i \rangle \geq \langle \mathbf{w}_j^{(1)}, \mathbf{x}_i \rangle - \eta \left( T_{ij}(1,2) - G_j^{(i)}(1,2)(\gamma - \rho) + B_j^{(i)}(1,2)(\gamma + \rho) \right)$$
$$> -\eta + \eta \left( (n-k)(\gamma - \rho) - 2k(\gamma + \rho) - 1 \right)$$
$$> 0$$

and for $i \in \mathcal{S}_F$ and $i \not\sim j$

$$\langle \mathbf{w}_j^{(2)}, \mathbf{x}_i \rangle \leq \langle \mathbf{w}_j^{(1)}, \mathbf{x}_i \rangle + \eta \left( T_{ij}(1,2) - G_j^{(i)}(1,2)(\gamma - \rho) + B_j^{(i)}(1,2)(\gamma + \rho) \right)$$
$$< \eta - \eta \left( (n-k)(\gamma - \rho) - 2k(\gamma + \rho) - 1 \right)$$
$$< 0.$$

With the base case established we proceed by induction to prove if $i \in \mathcal{A}_j^{(t-1)}$ iff $i \sim j$, then $i \in \mathcal{A}_j^{(t)}$ iff $i \sim j$. By the assumptions on $\gamma$, the induction hypothesis and again using Lemma B.2, for $i \sim j$

$$\langle \mathbf{w}_j^{(t)}, \mathbf{x}_i \rangle \geq \langle \mathbf{w}_j^{(t-1)}, \mathbf{x}_i \rangle - \eta \left( T_{ij}(t-1,t) - G_j^{(i)}(t-1,t)(\gamma - \rho) + B_j^{(i)}(t-1,t)(\gamma + \rho) \right)$$
$$> \eta \left( (n-k)(\gamma - \rho) - k(\gamma + \rho) - 1 \right)$$
$$> 0$$

and for $i \not\sim j$

$$\langle \mathbf{w}_j^{(t)}, \mathbf{x}_i \rangle \leq \langle \mathbf{w}_j^{(t-1)}, \mathbf{x}_i \rangle + \eta \left( T_{ij}(t-1,t) - G_j^{(i)}(t-1,t)(\gamma - \rho) + B_j^{(i)}(t-1,t)(\gamma + \rho) \right)$$
$$< -\eta \left( (n-k)(\gamma - \rho) - k(\gamma + \rho) - 1 \right)$$
$$< 0.$$

Therefore, for an epoch $t$ satisfying $2 \leq t \leq \mathcal{T}_1$ then $i \in \mathcal{A}_j^{(t-1)}$ iff $i \sim j$. $\qquad \square$

## E.2  Proof of Lemma 3.10

**Lemma E.3** (Lemma 3.10). *Suppose Assumption 7 holds, then there is an iteration $\mathcal{T}_1 < \mathcal{T}_0$ such that*

$$\langle \mathbf{w}_j^{(\mathcal{T}_1)}, \mathbf{x}_i \rangle \leq \frac{\gamma(n-2k) + \rho n - 1 + (\gamma - \rho)}{m(1 + \gamma(n-2k) + \rho n - (\gamma - \rho))} + O(\eta) \text{ if } i \in \mathcal{S}_F, i \sim j$$

$$\langle \mathbf{w}_j^{(\mathcal{T}_1)}, \mathbf{x}_i \rangle \geq \frac{1 + \gamma(n-2k) - \rho n - (\gamma - \rho)}{m(1 + \gamma(n-2k) + \rho n - (\gamma - \rho))} + O(\eta) \text{ if } i \in \mathcal{S}_T, i \sim j$$

$$\langle \mathbf{w}_j^{(\mathcal{T}_1)}, \mathbf{x}_i \rangle \leq -\frac{\gamma(n-2k) - \rho n}{m(1 + \gamma(n-2k) + \rho n - (\gamma - \rho))} + O(\eta) \text{ if } i \not\sim j.$$

*Furthermore for $i \in \mathcal{S}_T$*

$$\ell(\mathcal{T}_1, \mathbf{x}_i) \leq \frac{2\rho n}{1 + \gamma(n-2k) + \rho n - (\gamma - \rho)} + O(\eta).$$

*Finally,*

$$\mathcal{T}_1 = \frac{1}{\eta m(1 + \gamma(n-2k) + \rho n - (\gamma - \rho))} + O(1).$$

*Proof.* Let $t < \mathcal{T}_0$. By Lemma E.2 and Lemma B.2 we can bound for $i \in \mathcal{S}_F$ and $i \sim j$ as follows,

$$\langle \mathbf{w}_j^{(t)}, \mathbf{x}_i \rangle \leq \langle \mathbf{w}_j^{(2)}, \mathbf{x}_i \rangle - \eta T_{ij}(2,t) - \eta(\gamma - \rho)B_j^{(i)}(2,t) + \eta(\gamma + \rho)G_j(2,t)$$
$$= \eta t((\gamma + \rho)(n-k) - (\gamma - \rho)(k-1) - 1) + O(\eta)$$
$$= \eta t(\gamma(n-2k) + \rho n - 1 + (\gamma - \rho)) + O(\eta).$$

Similarly, for $i \in \mathcal{S}_F, i \not\sim j$,

$$\langle \mathbf{w}_j^{(t)}, \mathbf{x}_i \rangle \leq \langle \mathbf{w}_j^{(2)}, \mathbf{x}_i \rangle + \eta T_{ij}(2,t) + \eta(\gamma + \rho)B_j^{(i)}(2,t) - \eta(\gamma - \rho)G_j(2,t)$$
$$= -\eta t((\gamma - \rho)(n-k) - (\gamma + \rho)k) + O(\eta)$$
$$= -\eta t(\gamma(n-2k) - \rho n) + O(\eta).$$

For $i \in \mathcal{S}_T, i \sim j$,

$$\langle \mathbf{w}_j^{(t)}, \mathbf{x}_i \rangle \leq \langle \mathbf{w}_j^{(2)}, \mathbf{x}_i \rangle + \eta T_{ij}(2,t) + \eta(\gamma + \rho)G_j^{(i)}(2,t) - \eta(\gamma - \rho)G_j(2,t)$$
$$= \eta t(1 + (\gamma + \rho)(n - k - 1) - (\gamma - \rho)k) + O(\eta)$$
$$= \eta t(1 + \gamma(n - 2k) + \rho n - (\gamma + \rho)) + O(\eta)$$

and

$$\langle \mathbf{w}_j^{(t)}, \mathbf{x}_i \rangle \geq \langle \mathbf{w}_j^{(2)}, \mathbf{x}_i \rangle + \eta T_{ij}(2,t) + \eta(\gamma - \rho)G_j^{(i)}(2,t) - \eta(\gamma + \rho)G_j(2,t)$$
$$= \eta t(1 + (\gamma - \rho)(n - k - 1) - (\gamma + \rho)k) + O(\eta)$$
$$= \eta t(1 + \gamma(n - 2k) - \rho n - (\gamma - \rho)) + O(\eta).$$

Lastly, for $i \in \mathcal{S}_T, i \not\sim j$,

$$\langle \mathbf{w}_j^{(t)}, \mathbf{x}_i \rangle \leq \langle \mathbf{w}_j^{(2)}, \mathbf{x}_i \rangle + \eta T_{ij}(2,t) - \eta(\gamma - \rho)G_j^{(i)}(2,t) + \eta(\gamma + \rho)G_j(2,t)$$
$$= -\eta t((\gamma - \rho)(n - k) - (\gamma + \rho)k) + O(\eta)$$
$$= -\eta t(\gamma(n - 2k) - \rho n) + O(\eta)$$

Therefore, for $i \in \mathcal{S}_T$,

$$f(t, \mathbf{x}_i) = y_i \sum_{j=1}^{2m} (-1)^j \phi(\langle \mathbf{w}_j^{(t)}, \mathbf{x}_i \rangle)$$
$$= \sum_{j \sim i} \langle \mathbf{w}_j^{(t)}, \mathbf{x}_i \rangle$$

from which we conclude

$$\eta mt(1 + \gamma(n - 2k) - \rho n - (\gamma - \rho)) + O(\eta) \leq f(t, \mathbf{x}_i) \leq \eta mt(1 + \gamma(n - 2k) + \rho n - (\gamma - \rho)) + O(\eta).$$

Therefore, as long as

$$\eta mt(1 + \gamma(n - 2k) + \rho n - (\gamma - \rho)) + O(\eta) < 1, \tag{9}$$

then $\ell(t, \mathbf{x}_i) > 0$. Let $\mathcal{T}_1$ be the largest value of $t$ satisfying (9) and $t < \mathcal{T}_0$. We see that

$$\mathcal{T}_1 = \frac{1}{\eta m(1 + \gamma(n - 2k) + \rho n - (\gamma - \rho))} + O(1).$$

From this, the bounds claimed follow. $\qquad\square$

### E.3 Proof of Lemma 3.11

**Lemma E.4** (Lemma 3.11). *Let Assumption 7 hold. Suppose at iteration $t_0$ the following conditions are satisfied.*

    *a. $\ell(t_0, \mathbf{x}_i) \leq a$ for all $i \in \mathcal{S}_T$,*

    *b. $\phi(\langle \mathbf{w}_j^{(t_0)}, \mathbf{x}_i \rangle) \leq b$ for all $i \in \mathcal{S}_F$ and $i \sim j$,*

    *c. For all iterations $\tau$ satisfying $t_0 \leq \tau \leq t$ it holds that $i \in \mathcal{A}_j^{(\tau)}$ only if $i \sim j$,*

    *d. For all iterations $\tau$ satisfying $t_0 \leq \tau \leq t$, $i \in \mathcal{A}_j^{(\tau)}$ if $i \sim j$ and $i \in \mathcal{S}_T$.*

*Then for $j \in [2m]$ and $p \in \{-1, 1\}$ we have*

$$B_j(t_0, \tau) \leq \frac{k}{\eta}\left(\frac{3b}{2} + \frac{2a}{m}\right),$$
$$\sum_{j \sim s} G_j(t_0, t) \leq \frac{1}{\gamma \eta}\left(\frac{3a}{2} + mb\right).$$

*Proof.* Consider an arbitrary neuron $j \in [2m]$, using Lemma B.3 and assumption (b) we bound for $t < \tau \leq t$, $i \in \mathcal{S}_F$, and $i \sim j$

$$\phi(\langle \mathbf{w}_j^{(\tau)}, \mathbf{x}_i \rangle) \leq b - \eta(T_{ij}(t_0, \tau) - (\gamma + \rho)G_j(t_0, \tau) - 1).$$

As $\phi(\langle \mathbf{w}_j^{(\tau)}, \mathbf{x}_i \rangle) \geq 0$ in general we may conclude that

$$T_{ij}(t_0, \tau) \leq \frac{b}{\eta} + (\gamma + \rho)G_j(t_0, \tau) + 1.$$

Summing over all $i \in \mathcal{S}_F$ such that $i \sim j$ then by assumption (c)

$$\sum_{\substack{i \in \mathcal{S}_F \\ i \sim j}} T_{ij}(t_0, \tau) = B_j(t_0, \tau).$$

Combining these expressions it follows that

$$B_j(t_0, \tau) \leq \frac{kb}{\eta} + k(\gamma + \rho)G_j(t_0, \tau) + k.$$

As the number of clean updates on a pair of neurons $\ell$ and $j$ with $\ell \sim j$ is the same by assumptions (c) and (d), then we may rewrite this bound as

$$B_j(t_0, \tau) \leq \frac{kb}{\eta} + \frac{k(\gamma + \rho)}{m} \sum_{\ell \sim j} G_\ell(t_0, t) + k. \tag{10}$$

Let $s \in [2m]$ be arbitrary, we proceed to bound $\sum_{j \sim s} G_j(t_0, t)$. Using Lemma B.2, for $i \in \mathcal{S}_T$ and $i \sim j$

$$\langle \mathbf{w}_j^{(\tau-1)}, \mathbf{x}_i \rangle \geq \langle \mathbf{w}_j^{(t)}, \mathbf{x}_i \rangle + \eta(T_{ij}(t_0, \tau - 1) + (\gamma - \rho)G_j^{(i)}(t_0, \tau - 1) - (\gamma + \rho)B_j(t_0, \tau - 1)).$$

By assumptions (c) and (d)

$$\begin{aligned}
y_i f(\tau - 1, \mathbf{x}_i) &= \sum_{j \sim i} \phi(\langle \mathbf{w}_j^{(\tau-1)}, \mathbf{x}_i \rangle) \\
&\geq \sum_{j \sim i} \left( \langle \mathbf{w}_j^{(t_0)}, \mathbf{x}_i \rangle + \eta T_{ij}(t_0, \tau - 1) \right. \\
&\qquad \left. + \eta(\gamma - \rho)G_j^{(i)}(t_0, \tau - 1) - \eta(\gamma + \rho)B_j(t_0, \tau - 1) \right) \\
&\geq (1 - a) + \eta(1 - (\gamma - \rho))T_i(t_0, \tau - 1) \\
&\qquad + \left( \sum_{j \sim i} ((\gamma - \rho)G_j(t_0, \tau - 1) - (\gamma + \rho)B_j(t_0, \tau - 1)) \right).
\end{aligned}$$

Note either $T_i(t_0, \tau) = T_i(t_0, \tau - 1)$ or $\ell(\tau, \mathbf{x}_i) > 0$. Consider the case where the latter holds, then

$$\eta((1 - (\gamma - \rho))T_i(t_0, \tau - 1) + \sum_{j \sim i} ((\gamma - \rho)G_j(t_0, \tau - 1) - (\gamma + \rho)B_j(t_0, \tau - 1))) < a.$$

Furthermore, suppose $\tau' \leq \tau$ is the first iteration before $\tau$ such that $G_j(\tau', \tau) = 0$ and let $i \in \mathcal{S}_T$, $i \sim s$ be a point that makes an update at iteration $\tau' - 1$. Using the above bound it follows that

$$\eta((1 - (\gamma - \rho))T_i(t_0, \tau' - 1) + \sum_{j \sim i} ((\gamma - \rho)G_j(t_0, \tau' - 1) - (\gamma + \rho)B_j(t_0, \tau' - 1))) < a.$$

This implies

$$\sum_{j \sim i} ((\gamma - \rho)G_j(t_0, \tau' - 1) - (\gamma + \rho)B_j(t_0, \tau' - 1)) < \frac{a}{\eta}$$

and

$$\sum_{j \sim s} ((\gamma - \rho)G_j(t_0, \tau') - (\gamma + \rho)B_j(t_0, \tau')) < \frac{a}{\eta} - 2(n - k)(\gamma - \rho).$$

By the construction of $\tau'$ it follows that

$$\sum_{j\sim s}((\gamma-\rho)G_j(t_0,\tau)-(\gamma+\rho)B_j(t_0,\tau)) < \frac{a}{\eta}-2(n-k)(\gamma-\rho).$$

From this we get the bound

$$\sum_{j\sim s}G_j(t_0,t) \le \frac{1}{\gamma-\rho}\left(\frac{a}{\eta}+(\gamma+\rho)\sum_{j\sim s}B_j(t_0,t)\right)+O(1). \tag{11}$$

Combining (10) summed over $j\sim s$ with (11), then

$$\sum_{j\sim s}G_j(t_0,t) \le \frac{1}{\gamma-\rho}\left(\frac{a}{\eta}+(\gamma+\rho)\left(\frac{kmb}{\eta}+k(\gamma+\rho)\sum_{j\sim s}G_j(t_0,t)+k\right)\right)+O(1)$$

$$\le \frac{1}{\gamma-\rho-k(\gamma+\rho)^2}\left(\frac{a}{\eta}+\frac{kmb(\gamma+\rho)}{\eta}\right)+O(1)$$

and

$$B_j(t_0,\tau) \le \frac{kb}{\eta}+\frac{k(\gamma+\rho)}{m(\gamma-\rho-k(\gamma+\rho)^2)}\left(\frac{a}{\eta}+\frac{kmb(\gamma+\rho)}{\eta}\right)+O(1).$$

Using $\rho\le\frac{\gamma}{5}$, $\eta$ sufficiently small and $\gamma\le\frac{1}{36k}$ (Assumption 7) these bounds simplify to

$$B_j(t_0,\tau) \le \frac{k}{\eta}\left(\frac{3b}{2}+\frac{2a}{m}\right),$$

$$\sum_{j\sim s}G_j(t_0,t) \le \frac{1}{\gamma\eta}\left(\frac{3a}{2}+mb\right). \qquad\square$$

### E.4  Late training

**Lemma E.5.** *Under Assumption 7 the training process terminates at an iteration $\mathcal{T}_{end}$ satisfying*

$$\ell(\mathcal{T}_{end},\mathbf{x}_i) = 0$$

*for all $i\in\mathcal{S}_T$ and*

$$\phi(\langle\mathbf{w}_j^{(\mathcal{T}_{end})},\mathbf{x}_i\rangle) = 0$$

*for all $i\in\mathcal{S}_F$ and $j\in[2m]$.*

*Proof.* By Lemma E.3, at iteration $t=\mathcal{T}_1$ with

$$a = \frac{2\rho n}{1+\gamma(n-2k)+\rho n-(\gamma-\rho)}+O(\eta),$$

$$b = \frac{\gamma(n-2k)+\rho n-1+(\gamma-\rho)}{m(1+\gamma(n-2k)+\rho n-(\gamma-\rho))}+O(\eta)$$

then the first two conditions of Lemma E.4 are satisfied. Next, using Lemma B.2, we see by induction on $t\ge\mathcal{T}_1$ that if

$$B_j(\mathcal{T}_1,t) < \frac{\gamma(n-2k)-\rho n}{\eta m(\gamma+\rho)(1+\gamma(n-2k)+\rho n-(\gamma-\rho))}+O(1)$$

then for $i\not\sim j$ and $\mathcal{T}_1\le\tau\le t$,

$$\langle\mathbf{w}_j^{(\tau)},\mathbf{x}_i\rangle \le \langle\mathbf{w}_j^{(\mathcal{T}_1)},\mathbf{x}_i\rangle+\eta(\gamma+\rho)B_j(\mathcal{T}_1,\tau)$$

$$\le \langle\mathbf{w}_j^{(\mathcal{T}_1)},\mathbf{x}_i\rangle+\eta(\gamma+\rho)B_j(\mathcal{T}_1,t)$$

$$< 0$$

and for $i \in \mathcal{S}_T$, $i \sim j$, and $\mathcal{T}_1 \leq \tau \leq t$,

$$
\begin{aligned}
\langle \mathbf{w}_j^{(\tau)}, \mathbf{x}_i \rangle &\geq \langle \mathbf{w}_j^{(\mathcal{T}_1)}, \mathbf{x}_i \rangle - \eta(\gamma + \rho) B_j(\mathcal{T}_1, \tau) \\
&\geq \langle \mathbf{w}_j^{(\mathcal{T}_1)}, \mathbf{x}_i \rangle - \eta(\gamma + \rho) B_j(\mathcal{T}_1, t) \\
&\geq \frac{1 - (\gamma - \rho)}{m(1 + \gamma(n - 2k) + \rho n - (\gamma - \rho))} + O(\eta) \\
&> 0
\end{aligned}
$$

for $\eta$ sufficiently small. Thus we have shown under an additional assumption on $B_j(\mathcal{T}_1, t)$ that with $t_0 = \mathcal{T}_1$ and $a$ and $b$ as defined above, then all four conditions of Lemma E.4 are satisfied. As a result GD converges or terminates as long as

$$
k\left(\frac{3b}{2\eta} + \frac{2a}{\eta m}\right) < \frac{\gamma(n - 2k) - \rho n}{\eta m(\gamma + \rho)(1 + \gamma(n - 2k) + \rho n - (\gamma - \rho))} + O(1)
$$

which is equivalent to

$$
k(\gamma + \rho)\left(\frac{3}{2}\left(\gamma(n - 2k) - 1 + (\gamma - \rho)\right) + \frac{11}{2}\rho n\right) < \gamma(n - 2k) - \rho n + O(\eta).
$$

This is true by Assumption 7, as

$$
\begin{aligned}
k(\gamma + \rho)\left(\frac{3}{2}\left(\gamma(n - 2k) - 1 + (\gamma - \rho)\right) + \frac{11}{2}\rho n\right) &\leq \frac{1}{30}\left(\frac{3}{2}\gamma(n - 2k) - \frac{3}{2} + \frac{1}{36} + \frac{1}{2}\right) \\
&\leq \frac{1}{20}\gamma(n - 2k) - \frac{1}{2} \\
&< \gamma(n - 2k) - \rho n,
\end{aligned}
$$

where above we used $\gamma \leq \min\left\{\frac{1}{36k}, \frac{1}{36}\right\}$ and $\rho \leq \min\left\{\frac{\gamma}{5}, \frac{n}{11}\right\}$. $\qquad\square$

**Lemma E.6.** *Assume Assumption 7 holds. Let $y \in \{-1, 1\}$ be drawn uniformly at random and $\mathbf{x} := y\sqrt{\gamma}\mathbf{v} + \sqrt{1 - \gamma}\mathbf{n}$ where $\mathbf{n} \sim \text{Uniform}(\mathcal{S}^{d-1} \cap \text{span}\{\mathbf{v}\}^{\perp})$. Suppose that $|\langle \mathbf{n}, \mathbf{n}_\ell \rangle| < \frac{\rho}{1 - \gamma}$ for all $l \in [2n]$, then $y f(\mathcal{T}_{end}, \mathbf{x}) > 0$.*

*Proof.* We proceed as in the proof of Lemma C.13. Following the same steps as in (5), for any $j \in [2m]$

$$
\begin{aligned}
\langle \mathbf{w}_j^{(\mathcal{T}_{end})}, \mathbf{x} \rangle &= \langle \mathbf{w}_j^{(2)}, \mathbf{x}_i \rangle + (-1)^j \eta \sum_{\ell=1}^{2n} T_{\ell j}(1, \mathcal{T}_{end}) y_\ell \langle \mathbf{x}_\ell, \mathbf{x}_i \rangle \\
&= \langle \mathbf{w}_j^{(2)}, \mathbf{x}_i \rangle + (-1)^j y\eta \sum_{\ell=1}^{2n} T_{\ell j}(2, t)(-1)^\ell y\beta(\ell)\langle \mathbf{x}_\ell, \mathbf{x} \rangle \\
&= \langle \mathbf{w}_j^{(2)}, \mathbf{x}_i \rangle + (-1)^j y\eta \sum_{\ell=1}^{2n} T_{\ell j}(2, t)\lambda_{i\ell}',
\end{aligned}
$$

where $\lambda_\ell' := (-1)^l y\beta(\ell)\langle \mathbf{x}_\ell, \mathbf{x} \rangle = \beta(\ell)\gamma + (1 - \gamma)\langle \mathbf{n}_\ell, \mathbf{n} \rangle$. Then as in Lemma B.1

$$
\begin{aligned}
\gamma - \rho \leq &\lambda_\ell' \leq \gamma + \rho, \text{ if } i \in \mathcal{S}_T, \\
-(\gamma + \rho) \leq &\lambda_\ell' \leq -(\gamma - \rho), \text{ if } i \in \mathcal{S}_F,.
\end{aligned}
$$

Recall, from Lemma E.2, for any $j \in [2m]$ then $G_j(2, \mathcal{T}_{end}) \geq G_j(2, \mathcal{T}_1) = (\mathcal{T}_1 - 2)(n - k)$. As a consequence, for $j \in \Gamma_p$ we have

$$
\begin{aligned}
\langle \mathbf{w}_j^{(\mathcal{T}_{end})}, \mathbf{x} \rangle &\geq \langle \mathbf{w}_j^{(2)}, \mathbf{x}_i \rangle + \eta G_j(2, \mathcal{T}_{end})(\gamma - \rho) - \eta B_j(2, \mathcal{T}_{end})(\gamma + \rho) \\
&\geq O(\eta) + \mathcal{T}_1(n - k)(\gamma - \rho) - \eta B_j(2, \mathcal{T}_{end})(\gamma + \rho).
\end{aligned}
$$

For $j$ such that $(-1)^j = y$ then

$$
\begin{aligned}
\phi(\langle \mathbf{w}_j^{(\mathcal{T}_{end})}, \mathbf{x} \rangle) &\leq \phi(\langle \mathbf{w}_j^{(2)}, \mathbf{x}_i \rangle - \eta G_j(2, \mathcal{T}_{end})(\gamma - \rho) + \eta B_j(2, \mathcal{T}_{end})(\gamma + \rho)) \\
&\leq O(\eta) + \eta B_j(2, \mathcal{T}_{end})(\gamma + \rho).
\end{aligned}
$$

As a result

$$yf(\mathcal{T}_{\text{end}}, \mathbf{x}) = \sum_{j \in [2m]} \phi(\langle \mathbf{w}_j^{(\mathcal{T}_{\text{end}})}, \mathbf{x} \rangle)$$

$$\geq \sum_{j \,:\, (-1)^j = y} \langle \mathbf{w}_j^{(\mathcal{T}_{\text{end}})}, \mathbf{x} \rangle - \sum_{j \,:\, (-1)^j \neq y} \phi(\langle \mathbf{w}_j^{(\mathcal{T}_{\text{end}})}, \mathbf{x} \rangle)$$

$$\geq \eta \sum_{j \,:\, (-1)^j = y} (\mathcal{T}_1 - 2)(n - k)(\gamma - \rho) - \eta \sum_{j \in [2n]} B_j(2, \mathcal{T}_{\text{end}})(\gamma + \rho) + O(\eta)$$

$$\geq \eta m \mathcal{T}_1 (n - k)(\gamma - \rho) - \eta B(2, \mathcal{T}_{\text{end}})(\gamma + \rho)) + O(\eta)$$

Decompose $B(2, \mathcal{T}_{\text{end}}) = B(2, \mathcal{T}_1) + B(\mathcal{T}_1, \mathcal{T}_{\text{end}})$ and observe from Lemma E.2 that

$$B(2, \mathcal{T}_1) = 2km(\mathcal{T}_1 - 2) = 2km\mathcal{T}_1 + O(1).$$

From Lemma E.3 and Lemma E.4, using the assumptions $\rho \leq \frac{n}{11}$, $\eta$ sufficiently small and $\gamma \leq \min\{\frac{k}{36}, \frac{1}{36}\}$ then

$$B(\mathcal{T}_1, \mathcal{T}_{\text{end}}) \leq 2mk \left( \frac{3b}{2\eta} + \frac{2a}{\eta m} \right)$$

$$\leq 2mk\mathcal{T}_1 \left( \frac{3(\gamma(n - 2k) + \rho n - 1 + (\gamma - \rho))}{2} + 4\rho n \right) + O(1)$$

$$\leq 2m\mathcal{T}_1 \left( \frac{n - 2k}{24} + \frac{1}{22} - \frac{1}{2} + \frac{1}{72} + \frac{4}{11} \right) + O(1)$$

$$\leq \frac{m\mathcal{T}_1(n - k)}{12}.$$

Using the assumption that $k \leq \frac{n}{100}$ and $\eta$ is sufficiently small we see that

$$B(2, \mathcal{T}_{\text{end}}) \leq \frac{m\mathcal{T}_1(n - k)}{9}.$$

As

$$yf(\mathcal{T}_{\text{end}}, \mathbf{x}) \geq \eta m \mathcal{T}_1 (n - k)((\gamma - \rho) - (\gamma + \rho)/9) + O(\eta),$$

then $yf(\mathcal{T}_{\text{end}}, \mathbf{x})$ is positive provided $(\gamma - \rho) - (\gamma + \rho)/9$ is positive and $\eta$ is sufficiently small. Both these conditions are guaranteed by Assumption 7 and thus the test point is correctly classified. $\square$

## E.5    Proof of Theorem 3.8

**Theorem E.7** (Theorem 3.8). *Let Assumption 6 hold with $\rho = \gamma/5$. There exists a sufficiently small step-size $\eta$ such that with probability at least $1 - \delta$ over the randomness of the dataset and network initialization we have the following.*

1. *There exists a positive constant $C$ such that the training process terminates at an iteration $\mathcal{T}_{end} \leq \frac{Cn}{\eta}$.*

2. *For all $i \in \mathcal{S}_T$ then $\ell(\mathcal{T}_{end}, \mathbf{x}_i) = 0$ while $\ell(\mathcal{T}_{end}, \mathbf{x}_i) = 1$ for all $i \in \mathcal{S}_F$.*

3. *There exists a positive constant $c$ such that the generalization error satisfies*

$$\mathbb{P}(\text{sgn}(f(\mathcal{T}_{end}, \mathbf{x})) \neq y) \leq \exp\left( -cd\gamma^2 \right).$$

*Proof.* Under Assumption 7, Statements 1 and 2 follow from Lemma E.5. The bound on $\mathcal{T}_{\text{end}}$ follows from Lemma E.4 applied at $t_0 = 2$, indeed the number of iterations cannot exceed the number of updates which, as $\gamma > \frac{3}{n}$, is bounded as

$$B(2, \mathcal{T}_{\text{end}}) + G(2, \mathcal{T}_{\text{end}}) \leq k \left( \frac{4}{\eta m} \right) + \frac{2}{\gamma} \left( \frac{3}{2\eta} \right) + O(\eta)$$

Again under Assumption 7 using Lemma E.6 then Statement 3 follows in exactly the same manner as the proof of Statement 3 for Theorem C.14. Finally, Lemma E.1 implies that under Assumption 6 then Assumption 7 holds with probability at least $1 - \delta$. $\square$

## Appendix F    Numerical simulations

**Reproducibility statement:** the code used to generate the following figures can be found at `https://github.com/wswartworth/benign_overfitting`.

To investigate our theory we train two-layer neural networks with ReLU activations using full-batch gradient descent and a fixed step size. We train on a synthetic binary classification dataset generated as per Section 2.1. Finally, we train using both the hinge and logistic loss.

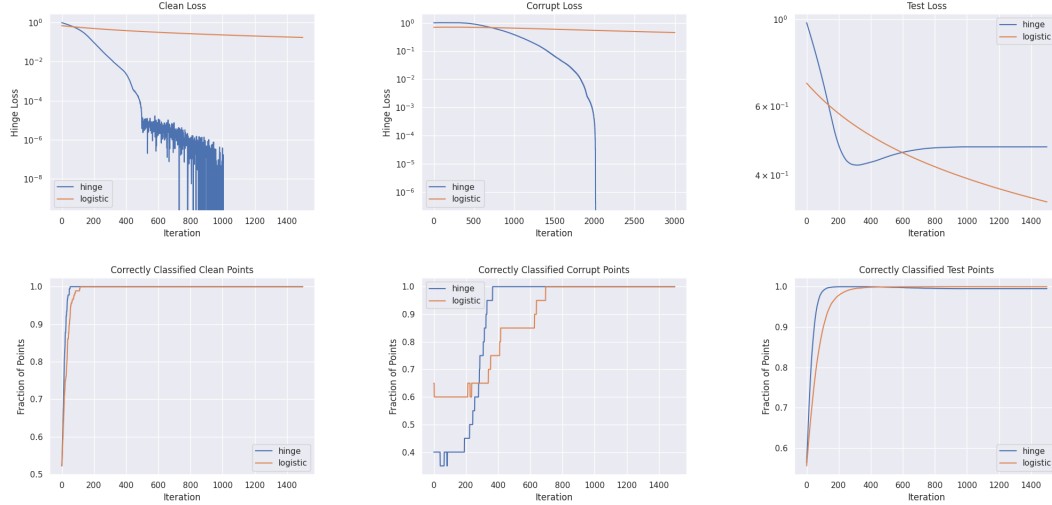

Figure 1: from left to right, the first row shows the clean, corrupt, and test losses as a function of epoch (or iteration). The second row shows the fraction of clean, corrupt, and test points that are classified correctly. These plots were generated with $n = 100$, $d = 800$, $k/n = 0.1$, $m = 100$, $\gamma = 0.015$, and a step size of $\eta = 0.01$.

In Figure 1 we call attention to the difference in the training dynamics of hinge loss versus logistic loss. Perhaps the key difference between the hinge loss and logistic loss is that the contributions from any given point do not get smaller as the point approaches $0$ loss. Furthermore, unlike with the logistic loss, points can actually attain zero hinge loss after a finite number of epochs. While a point has zero loss it ceases to contribute to the update of the network parameters. As a result, points close to zero hinge loss periodically activate and deactivate giving rise to the chaotic behavior observed as the training loss approaches zero. We emphasize that managing this behavior required a careful analysis distinct from that of prior works analysing the logistic loss.

In Figure 2 we call particular attention to the bottom right plot. Our theory predicts a phase transition between benign overfitting and non-benign overfitting when $\gamma \approx c/n$: the phase transition we observe empirically in the bottom-right heatmap suggests this estimate is reasonable. With regard to the hinge loss over the corrupt points, displayed in the top-right heatmap, we observe another phase transition, this time between overfitting and non-overfitting. The top and bottom heatmaps of the left-hand column display the hinge loss over the clean training set and total training set respectively, these appear very similar due to the fact that clean points make up $95\%$ of the training set. The clean points fail to achieve zero, or close to zero, hinge loss only when $\gamma$ is small and $n$ is large. As stated in the caption, in these experiments $d$ is fixed and thus as $n$ increases the near-orthogonality condition we require on the noise components in order to prove convergence to zero clean loss is compromised. As a result, when $\gamma$ is small and the correlations between noise vectors is potentially large it is possible for pairs of points with opposite labels to be significantly correlated.

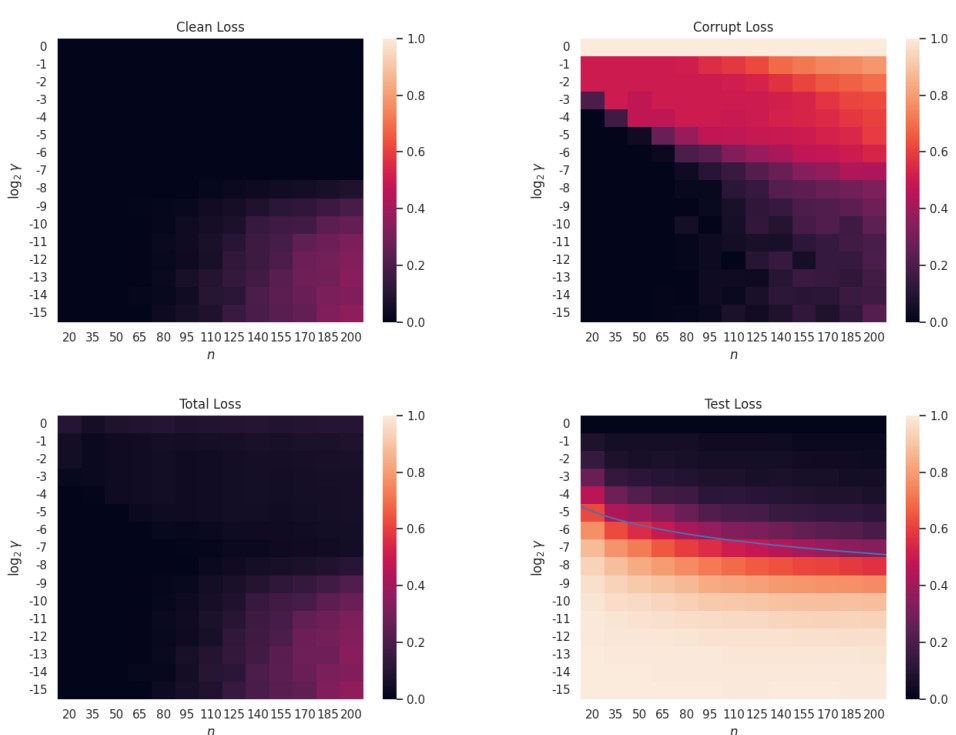

Figure 2: from left to right in the top row we show the loss on clean training and corrupt training points after training. In the bottom row and again from left to right we show the total loss after training and the test loss on 10000 randomly generated points. For each plot we set $d = 1000, m = 30, \eta = 0.005$ and train for 5000 iterations of gradient descent using hinge loss. In each plot we vary $\gamma$ and $n$ and hold the fraction of corrupt points constant at $0.05$. In the bottom right plot we also graph the curve $c/n$ for $c \approx 0.6$

