# Contents

## A  Corrections to the main paper

In the course of preparing the supplementary materials we identified the following two mistakes.

1. First we found a mistake in one of the proofs for a lemma describing the conditions at initialization. We have fixed this issue in the supplementary and highlight below the two errors this has caused in the main paper.

   - An entry in the Table 1 needs to be updated to reflect these changes: in particular the dependence on $n$ is $\frac{1}{\delta}$ not $n \geq 1$ for Theorem 3.1. Aside from this one entry Table 1 is correct. For the convenience of the reader we provide the full, corrected table below.

Table 1: Comparison of results up to constants, note in all cases $d \geq Cn^2 \log(n/\delta)$, $k \leq \frac{n}{C}$ where $C$ is an appropriatly chosen constant.

|  | Frei et al. (2022) | Xu & Gu (2023) | Theorem 3.1 | Theorem 3.6 | Theorem 3.8 |
|---|---|---|---|---|---|
| $n \geq C\cdot$ | $\log\left(\frac{1}{\delta}\right)$ | $\log\left(\frac{m}{\delta}\right)$ | $\frac{1}{\delta}$ | $1$ | $\log\left(\frac{m}{\delta}\right)$ |
| $m \geq C\cdot$ | $1$ | $\log\left(\frac{n}{\delta}\right)$ | $\log\left(\frac{n}{\delta}\right)$ | $\log\left(\frac{n}{\delta}\right)$ | $\log\left(\frac{n}{\delta}\right)$ |
| $\gamma \leq \frac{1}{C}\cdot$ | $\frac{1}{n}$ | $\frac{1}{n}$ | $\frac{1}{n}$ | $\frac{1}{\sqrt{nd}}$ | $\frac{1}{k}$ |
| $\gamma \geq C\cdot$ | $\frac{1}{\sqrt{nd}}$ | $\sqrt{\frac{\log(\frac{md}{n\delta})}{nd}}$ | $\sqrt{\frac{\log(\frac{n^2}{\delta})}{d}}$ | $0$ | $\frac{1}{n}$ |
| Result | Benign[1] | Benign | Benign | Non-benign | No overfit |

   - The same mistake also means that the sentence starting on line 188 "Comparing specifically the results on benign overfitting, we observe a better dependency on n in particular compared with Xu & Gu (2023)..." is incorrect and indeed their work has better dependence on $n$.

2. On line 262 we made a mistake in the explanation of our results: instead of $C\epsilon(1 + n\gamma)$ it should read $\frac{C\varepsilon n\gamma}{(\eta(\gamma+\rho)}$.

# B  Problem setup

## B.1  Notation

In order to provide a convenient reference for the reader, we summarize our notation as follows.

- For $n \in \mathbb{Z}_{\geq 1}$ then $[n] := \{1, 2, 3...n\}$. Furthermore we let $[n]_e := \{i \in [n] : i \text{ is even}\}$ and $[n]_o := \{i \in [n] : i \text{ is odd}\}$.

- For two iterations $t_0, t_1$ with $t_1 > t_0$ we define the following counting functions.

  1. $T_{ij}(t_0, t_1) := \sum_{\tau=t_0}^{t_1-1} \mathbb{1}(i \in \mathcal{A}_j^{(\tau)} \cap \mathcal{F}^{(\tau)})$ is the number of times the $i$-th data point updates the $j$-th neuron between iterations $t_0$ and $t_1$.

  2. $T_i(t_0, t_1) := \sum_{j \in [2m]} T_{ij}(t_0, t_1)$ is the total number of updates from the $i$-th data point to the entire network between iterations $t_0$ and $t_1$.

  3. $G_j(t_0, t_1) := \sum_{i \in \mathcal{S}_T} T_{ij}(t_0, t_1)$ and $B_j(t_0, t_1) := \sum_{i \in \mathcal{S}_F} T_{ij}(t_0, t_1)$ are the number of clean and corrupt updates applied to the $j$-th neuron respectively between iterations $t_0$ and $t_1$. We further define $G_j^{(i)}(t_0, t_1) := G_j(t_0, t_1) - \mathbb{1}(i \in \mathcal{S}_T) T_{ij}(t_0, t_1)$ and $B_j^{(i)}(t_0, t_1) := B_j(t_0, t_1) - \mathbb{1}(i \in \mathcal{S}_F) T_{ij}(t_0, t_1)$.

  4. $H_j^{(i)}(t_0, t_1) := \sum_{\ell \neq i} T_{\ell j}(t_0, t_1)$ is the number of times any data point except the $i$-th updates the $j$-th neuron between iterations $t_0$ and $t_1$.

  5. $G(t_0, t_1) := \sum_{j \in [2m]} G_j(t_0, t_1)$ and $B(t_0, t_1) := \sum_{j \in [2m]} B_j(t_0, t_1)$ are the total number of clean and corrupt updates applied to the entire network between iterations $t_0$ and $t_1$. We similarly define $G^{(i)}(t_0, t_1) = G(t_0, t_1) - \mathbb{1}(i \in \mathcal{S}_T) T_i(t_0, t_1)$ and $B^{(i)}(t_0, t_1) = B(t_0, t_1) - \mathbb{1}(i \in \mathcal{S}_F) T_i(t_0, t_1)$

  6. $T(t_0, t_1) := \sum_{\ell} T_\ell(t_0, t_1)$ is the total number of updates from all points applied to the entire network between iterations $t_0$ and $t_1$. We also define $T^{(i)}(t_0, t_1) = T(t_0, t_1) - T_i(t_0, t_1)$, the number of updates excluding those from point $i$.

  7. $S_i(t_0, t_1) := \sum_{\tau=t_0}^{t_1-1} \mathbb{1}(\exists j \in [2m] : i \in \mathcal{A}_j^{(\tau)} \cap \mathcal{F}^{(\tau)})$ is the number of iterations between $t_0$ and $t_1$ in which the $i$th data point participates. We say a data point participates during an iteration if it contributes to the update of at least one neuron at said iteration.

- We extend each of these definitions to the case $t_0 = t_1$ by letting the empty sum be zero.

- The signal alignment of the $j$-th neuron is defined as $C_j^{(t)} = \langle \mathbf{w}_j^{(t)}, (-1)^j \mathbf{v} \rangle$.

- We use the notation $O(\eta)$ to denote a quantity $f(\eta)$ such that

$$\limsup_{\eta \to 0} \frac{|f(\eta)|}{\eta} < \infty.$$

  Likewise, $f(\eta) = O(1)$ if

$$\limsup_{\eta \to 0} |f(\eta)| < \infty,$$

  $f(\eta) = \Omega(\eta)$ if

$$\liminf_{\eta \to 0} \frac{|f(\eta)|}{\eta} > 0,$$

  and $f(\eta) = \Omega(1)$ if

$$\liminf_{\eta \to 0} |f(\eta)| > 0.$$

  Here the limit is taken as $\eta$ and $\lambda_w$ go to zero while the other parameters of the model and data remain fixed. We will always choose $\lambda_w$ such that $\lambda_w = O(\eta)$.

- Denote $\mathcal{T}_0$ be the first iteration $t$ where $\mathcal{F}^{(t)} \neq [2n]$.

- We use $C \geq 1$ and $c \leq 1$ to generically represent sufficiently large and small constants respectively. Furthermore, we reuse both $C$ and $c$ from one line to another: for example, $2Cx = Cx$ and $0.5cx = cx$.

- Finally, in much of our analysis for particular variables, notably $\gamma$, the constants involved matter and as such we work with explicit constants. For other variables, in particular $m$ and $n$, then typically as long as they are sufficiently large the explicit constants involved are not important. As such we typically resort to using a generically large enough constant $C$.

## B.2 Data model

For the reader's convenience we recap the data model studied in this work. We consider a training sample of $2n$ pairs of points and their labels $(\mathbf{x}_i, y_i)_{i=1}^{2n}$ where $(\mathbf{x}_i, y_i) \in \mathbb{R}^d \times \{-1, +1\}$. Furthermore, we identify two disjoint subsets $\mathcal{S}_T \subset [2n]$ and $\mathcal{S}_F \subset [2n]$, where $\mathcal{S}_T \cup \mathcal{S}_F = [2n]$, which correspond to the clean and corrupt points in our sample respectively. The categorization of a point as clean or corrupted is determined by its label: for all $i \in [2n]$ we assume $y_i = \beta(i)(-1)^i$ where $\beta(i) = -1$ iff $i \in \mathcal{S}_F$ and $\beta(i) = 1$ otherwise. In addition, we assume $|\mathcal{S}_F \cap [2n]_e| = |\mathcal{S}_F \cap [2n]_o| = k$ and $|\mathcal{S}_T \cap [2n]_e| = |\mathcal{S}_T \cap [2n]_o| = n - k$. We remark that this assumption simplifies the exposition of our results but is not actually integral to our analysis. Each data point is assumed to have the form

$$\mathbf{x}_i = (-1)^i(\sqrt{\gamma}\mathbf{v} + \sqrt{1 - \gamma}\beta(i)\mathbf{n}_i). \tag{1}$$

Here $\mathbf{v} \in \mathbb{R}^d$ satisfies $\|\mathbf{v}\| = 1$. We refer to $\mathbf{v}$ as the signal component as the alignment of a clean point with $\mathbf{v}$ determines its sign. Indeed, $\mathrm{sign}(\langle \mathbf{x}_i, \mathbf{v} \rangle) = (-1)^i = y_i$ for $i \in \mathcal{S}_T$ whereas $\mathrm{sign}(\langle \mathbf{x}_i, \mathbf{v} \rangle) = -y_i$ for $i \in \mathcal{S}_F$. Thus we may view the labels of corrupt point as flipped from their clean state. The random vectors $(\mathbf{n}_i)_{i=1}^{2n}$ are mutually independent and identically distributed (i.i.d.) drawn from the uniform distribution over $\mathbb{S}^{d-1} \cap \mathrm{span}\{\mathbf{v}\}^{\perp}$, which we denote $U(\mathbb{S}^{d-1} \cap \mathrm{span}\{\mathbf{v}\}^{\perp})$. This distribution is symmetric, mean zero and for any $\mathbf{n} \sim U(\mathbb{S}^{d-1} \cap \mathrm{span}\{\mathbf{v}\}^{\perp})$ it holds that $\mathbf{n} \perp \mathbf{v}$ and $\|\mathbf{n}\| = 1$. We refer to these vectors as noise components due to the fact that they are independent of the labels of their respective points. We also remark that as the noise distribution is symmetric then clean and corrupt points are identically distributed. Indeed, the multiplication of the noise component by $\beta(i)$ results in the following expression which will prove convenient.

$$y_i\mathbf{x}_i = \beta(i)\sqrt{\gamma}\mathbf{v} + \sqrt{1 - \gamma}\mathbf{n}_i. \tag{2}$$

This expression entails that the only difference between clean and corrupt points during training is that they push neurons in opposite directions along the signal vector. Finally, the real, scalar quantity $\gamma \in [0, 1]$ controls the strength of the signal versus noise, furthermore the clean margin, i.e., the distance from any clean point to the max margin classifier, is $\sqrt{\gamma}$ by construction. Thus far we have discussed only the data in the training sample. We assume test data are drawn mutually i.i.d. from the same distribution as the points in the training sample but with the added proviso that they are always clean: to be clear, at test time a label is sampled as $y \sim U(\{-1, 1\})$ and the corresponding data point has the form

$$\mathbf{x} = y(\sqrt{\gamma}\mathbf{v} + \sqrt{1 - \gamma}\mathbf{n}), \tag{3}$$

where again $\mathbf{n} \sim U(\mathbb{S}^{d-1} \cap \mathrm{span}\{\mathbf{v}\}^{\perp})$.

## B.3 Network architecture, optimization and initialization

Here we also recap the network architecture and optimization and initialization setting. We consider a densely connected, single layer feedforward neural network $f : \mathbb{R}^{2m \times d} \times \mathbb{R}^d \to \mathbb{R}^d$ with the following forward pass map,

$$f(\mathbf{W}, \mathbf{x}) = \sum_{j=1}^{2m} (-1)^j \phi(\langle \mathbf{w}_j, \mathbf{x} \rangle).$$

Here $\phi := \max\{0, z\}$ denotes the ReLU activation function, $\mathbf{w}_j$ the $j$th row of $\mathbf{W}$ and $w_{jc}$ the element of $\mathbf{W}$ on row $j \in [m]$ and column $c \in [d]$. We remark that only the weights of the hidden layer, which we also refer to as the weights of the network, are trainable and the outer weights remain frozen throughout training. The network weights are optimized using full batch gradient descent (GD) with step size $\eta > 0$ to minimize the hinge loss over a training sample $((\mathbf{x}_i, y_i))_{i=1}^{2n} \subset (\mathbb{R}^d \times \{-1, 1\})^{2n}$ as described in Section B.2. After $t' > 0$ iterations this optimization process generates a sequence of weight matrices $(\mathbf{W}^{(t)})_{t=0}^{t'}$. For convenience we overload our

notation for the forward pass map of the network by letting $f(t, \mathbf{x}) := f(\mathbf{W}^{(t)}, \mathbf{x})$. We denote the hinge loss on the $i$-th point at iteration $t$ as

$$\ell(t, i) := \max\{0, 1 - y_i f(t, \mathbf{x}_i)\},$$

the hinge loss over the entire training sample at iteration $t$ is therefore defined as

$$L(t) := \sum_{i=1}^{2n} \ell(t, i).$$

Let $\mathcal{F}^{(t)} := \{i \in [2n] : \ell(t, \mathbf{x}_i) < 1\}$ be the set of points that have nonzero loss at iteration $t$, and $\mathcal{A}_j^{(t)} := \{i \in [2n] : \langle \mathbf{w}_j^{(t)}, \mathbf{x}_i \rangle > 0\}$ the set of points which activate the $j$th neuron at iteration $t$. Combining (2) with

$$\frac{\partial \ell(t, i)}{\partial w_{jr}} = \begin{cases} 0, & \langle \mathbf{w}_j^{(t)}, \mathbf{x}_i \rangle \leq 0, \\ -(-1)^j y_i x_{ir}, & \langle \mathbf{w}_j^{(t)}, \mathbf{x}_i \rangle > 0 \end{cases}$$

gives that the GD update rule[2] for the neuron weights for any iteration $t \geq 0$,

$$\mathbf{w}_j^{(t+1)} = \mathbf{w}_j^{(t)} + (-1)^j \eta \sum_{\ell=1}^{2n} \mathbb{1}(\ell \in \mathcal{A}_j^{(t)} \cap \mathcal{F}^{(t)}) y_\ell \mathbf{x}_\ell. \tag{4}$$

In regard to the initialization of the network weights, for convenience we assume each neuron's weight vector is drawn mutually i.i.d. uniform from the centered sphere with radius $\lambda_w > 0$. We remark that results analogous to the ones presented trivially hold if the weights are instead initialized mutually i.i.d. as $w_{jc}^{(0)} \sim \mathcal{N}(0, \sigma_w^2)$ as long as $\sigma_w^2$ is sufficiently small.

## B.4  Properties of the data and network at initialization

For each of our results to hold we require certain properties on both the network weights and training sample to hold at initialization. Here we bound the probabilities of each of these events in isolation and later will combine them using the union bound.

First, and in order to prove convergence, we require the noise components of the training sample to be approximately orthogonal to one another. A training sample whose noise components satisfy this approximate orthogonality condition we refer to as "good".

**Lemma B.1.** *Let $\rho, \delta \in (0, 1)$. Given a sequence $(\mathbf{n}_i)_{i=1}^{2n}$ of mutually i.i.d. random vectors with $\mathbf{n}_i \sim U(\mathbb{S}^{d-1} \cap span(\mathbf{v})^\perp)$, then assuming $d \geq \max\left\{6, 3\rho^{-2} \ln\left(\frac{2n^2}{\delta}\right)\right\}$*

$$\mathbb{P}\left(\bigcap_{i \neq \ell} \{|\langle \mathbf{n}_i, \mathbf{n}_\ell \rangle| \leq \rho\}\right) \geq 1 - \delta.$$

*Proof.* For any pairs of mutually i.i.d. random vectors $\mathbf{n}, \mathbf{n}' \sim U(\mathbb{S}^{d-1} \cap span(\mathbf{v})^\perp)$ and $\mathbf{u}, \mathbf{u}' \sim U(\mathbb{S}^{d-2})$ observe

$$\langle \mathbf{n}, \mathbf{n}' \rangle \stackrel{d}{=} \langle \mathbf{u}, \mathbf{u}' \rangle.$$

Due to independence of $\mathbf{u}, \mathbf{u}'$ and the rotational invariance of $U(\mathbb{S}^{d-2})$

$$\langle \mathbf{u}, \mathbf{u}' \rangle \stackrel{d}{=} \langle \mathbf{u}, \mathbf{e}_1 \rangle,$$

where here $\mathbf{e}_1 = [1, 0..0]^T$. Let $\mathrm{Cap}(\mathbf{e}_1, \rho) := \{\mathbf{z} \in \mathbb{S}^{d-2} : \langle \mathbf{e}_1, \mathbf{z} \rangle \geq \rho\}$ denote the *spherical cap* of $\mathbb{S}^{d-2}$ centered on $\mathbf{e}_1$. As $d \geq 6$ then from Ball (1997)[Lemma 2.2] it follows that

$$\mathbb{P}(|\langle \mathbf{n}, \mathbf{n}' \rangle| \geq \rho) = \mathbb{P}(\mathbf{u} \in \mathrm{Cap}(\mathbf{e}_1, \rho)) \leq \exp\left(-\frac{(d-2)\rho^2}{2}\right) \leq \exp\left(-\frac{d\rho^2}{3}\right).$$

---

[2]Although the derivative of ReLU clearly does not exist at zero, we follow the routine procedure of defining an update rule that extends the gradient update to cover this event.

Applying the union bound then

$$\mathbb{P}\left(\bigcap_{i,\ell\in[2n],i\neq\ell}\{|\langle \mathbf{n}_i, \mathbf{n}_\ell\rangle| \leq \rho\}\right) = 1 - \mathbb{P}\left(\bigcup_{i,\ell\in[2n],i\neq\ell}\{|\langle \mathbf{n}_i, \mathbf{n}_\ell\rangle| \geq \rho\}\right)$$

$$\geq 1 - 2n^2\mathbb{P}\left(|\langle \mathbf{n}_i, \mathbf{n}_\ell\rangle| \geq \rho\right)$$

$$\geq 1 - 2n^2\exp\left(-\frac{d\rho^2}{3}\right).$$

Setting $\delta \geq 2n^2\exp\left(-\frac{d\rho^2}{3}\right)$ and rearranging we arrive at the result claimed. $\qquad\square$

In addition to requiring the approximate orthogonality property on the training data, our approach also requires at least a large proportion of the neurons at initialization to satisfy certain conditions. To this end, we introduce the following notation, where $p \in \{-1, 1\}$.

- Let $\Gamma_p := \{j \;:\; (-1)^j = p, \; G_j(0,1)(\gamma - \rho) - B_j(0,1)(\gamma + \rho) \geq \frac{2\lambda_w}{\eta}\}$ denote the set of neurons with output weight $(-1)^p$ which have more clean points activating them than corrupt ones at initialization. We will show that these sets of neurons have a *predictable* behavior early during training before any clean points achieve zero loss. Let $\Gamma = \Gamma_1 \cup \Gamma_{-1}$.

- Let $\Theta_p := \{j \sim \Gamma_p \;:\; G_j(0,1)(\gamma + \rho) - B_j(0,1)(\gamma - \rho) < 1 - \gamma + \rho\} \subset \Gamma_p$. We will show that neurons in this subset are able to *carry* corrupt points through training, eventually, at least in the overfitting setting, enabling them to achieve zero loss.. Let $\Theta = \Theta_1 \cup \Theta_{-1}$.

Our goals are two-fold: first show $\Gamma_p$ accounts for a significant proportion of the neurons with output label sign matching $p$, second, and of particular importance for our result on benign overfitting, ensure each corrupt point activates a neuron in $\Theta_p$ where $p$ matches its label. To this end we first provide the following result.

**Lemma B.2.** *Define $\mu := \frac{2k}{n+k}$ and assume $\kappa \in (0, 1)$ satisfies $\kappa > \mu$. Given an arbitrary neuron $\mathbf{w}_j \sim U(\mathbb{S}^{d-1})$, we say that a collection of training points is $(\varepsilon, \kappa)$-good iff both $T_j(0,1) \geq 1$ and $B_j(0,1) < \kappa T_j(0,1)$ with probability at least $1 - \epsilon$ over the randomness of the neuron. Define*

$$\delta := 2\exp\left(-\frac{(n+k)(\kappa-\mu)^2}{16}\right),$$

*then with probability at least $1 - \frac{\delta}{\epsilon}$ the training sample is $(\varepsilon, \kappa)$-good.*

*Proof.* First we establish the notation for what follows: we say a point $\mathbf{x}$ is positive iff $\langle \mathbf{x}, \mathbf{v}\rangle > 0$ and is negative iff $\langle \mathbf{x}, \mathbf{v}\rangle < 0$. We use $\mathcal{S}^+$ and $\mathcal{S}^-$ to denote these sets of points respectively. Note by construction, see (1), clean and corrupt points of the same sign are mutually i.i.d. As here we only ever consider the activations at initialization, we also drop both the subscript $j$ as well as the argument parentheses on the counting functions. We also use $\pm$ superscripts to denote the subsets corresponding to activations from positive and negative points respectively: as indicative examples of this notation, $T$ is therefore used as shorthand for the total number of activations, $B^+$ is the number corrupt positive activations and $G^-$ is the number of clean negative activations.

By the symmetry of the distribution of $\mathbf{w}$, $\mathbb{P}(\langle \mathbf{w}, \mathbf{v}\rangle > 0) = \mathbb{P}(\langle \mathbf{w}, \mathbf{v}\rangle < 0) = \frac{1}{2}$. As a result

$$\mathbb{P}((B < \kappa T)\cap(T > 0)) = \frac{1}{2}\mathbb{P}((B < \kappa T)\cap(T > 0) \mid \langle \mathbf{w}, \mathbf{v}\rangle > 0)$$

$$+ \frac{1}{2}\mathbb{P}((B < \kappa T)\cap(T > 0) \mid \langle \mathbf{w}, \mathbf{v}\rangle < 0).$$

As the analysis and results derived under either condition will prove identical under reversal of the signs involved, without loss of generality we let $\langle \mathbf{w}, \mathbf{v}\rangle > 0$. Using the union bound

$$\mathbb{P}((B < \kappa T)\cap(T > 0) \mid \langle \mathbf{w}, \mathbf{v}\rangle > 0) \geq 1 - \mathbb{P}(T = 0 \mid \langle \mathbf{w}, \mathbf{v}\rangle > 0) - \mathbb{P}(B \geq \kappa T \mid \langle \mathbf{w}, \mathbf{v}\rangle > 0),$$

therefore it suffices to upper bound the two probabilities on the right-hand-side.

Observe if $\langle \mathbf{w}, \mathbf{v} \rangle > 0$ then for $\mathbf{x} \in \mathcal{S}^+$ $\mathbb{P}(\langle \mathbf{x}, \mathbf{w} \rangle) > 1/2$ and for $\mathbf{x} \in \mathcal{S}^-$ $\mathbb{P}(\langle \mathbf{x}, \mathbf{w} \rangle) < 1/2$. By the mutual independence of the preactivations $(\langle \mathbf{x}, \mathbf{w}_j \rangle)_{i=1}^{2n}$ then $\mathbb{P}(T = 0 \mid \langle \mathbf{w}, \mathbf{v} \rangle > 0) \leq (1/2)^n$. To upper bound the other probability of interest we further condition on the following two events: there are no negative clean activations and all corrupt points are positive. Conditioning on these three events, which we denote for convenience $\Lambda$, then $T^+ = T$ and furthermore the event $B < \kappa T$ is equivalent to $B^+ < \kappa T^+$. Again as the preactivations are mutually independent the number of positive activations can be lower bounded using a binomial distribution with probability $1/2$. Applying a Chernoff bound

$$\mathbb{P}\left(T^+ \geq \frac{n+k}{4} \;\middle|\; \Lambda\right) \geq 1 - \exp\left(-\frac{n+k}{16}\right).$$

Furthermore, sampling positive points which activate $\mathbf{w}_j$ is equivalent to uniformly sampling without replacement $T^+$ points from $\mathcal{S}^+$. Let $Z_\ell = 1$ iff the $\ell$-th element sampled from $\mathcal{S}^+$ is corrupt and is $0$ otherwise. Using a variant of Hoeffding's bound for sampling without replacement (see Proposition 1.2 of Bardenet & Maillard (2015) for example)

$$\mathbb{P}\left(B^+ \geq \kappa T^+ \mid \Lambda\right) = \mathbb{P}\left(\frac{1}{T^+}\sum_{\ell=1}^{T^+} Z_\ell - \mu \geq \kappa - \mu\right) \leq \exp\left(-2T^+(\kappa - \mu)^2\right).$$

Therefore

$$
\begin{aligned}
&\mathbb{P}((B < \kappa T) \cap (T > 0) \mid \langle \mathbf{w}, \mathbf{v} \rangle > 0) \\
&\geq 1 - \mathbb{P}(T = 0 \mid \langle \mathbf{w}, \mathbf{v} \rangle > 0) - \mathbb{P}(B \geq \kappa T \mid \langle \mathbf{w}, \mathbf{v} \rangle > 0) \\
&\geq 1 - (1/2)^n - \mathbb{P}\left(B^+ \geq \kappa T^+ \;\middle|\; T^+ \geq \frac{n+k}{4}, \Lambda\right) \mathbb{P}\left(T^+ \geq \frac{n+k}{4} \;\middle|\; \Lambda\right) \\
&\geq 1 - (1/2)^n - \exp\left(-\frac{(n+k)(\kappa - \mu)^2}{16}\right) \\
&\geq 1 - \delta.
\end{aligned}
$$

Now if instead $\langle \mathbf{x}, \mathbf{w} \rangle < 0$ then swapping the roles of the negative and positive points in the argument above gives the same answer. As a result

$$\mathbb{P}((B < \kappa T) \cap (T > 0)) \geq 1 - \delta.$$

For convenience let $X := (\mathbf{x}_i)_{i=1}^{2n}$ and

$$\chi_{\kappa,\epsilon}^c = \{X : \mathbb{P}_w((B \geq \kappa T) \cup (T = 0)) > \epsilon\}.$$

Note here that the subscript $w$ indicates randomness over the neuron alone and in addition clearly by construction

$$\mathbb{P}\left((B \geq \kappa T) \cup (T = 0) \mid X \in X_{\kappa,\epsilon}^c\right) > \epsilon.$$

Furthermore, as

$$\delta \geq \mathbb{P}\left((B \geq \kappa T) \cup (T = 0)\right) \geq \mathbb{P}\left((B \geq \kappa T) \cup (T = 0) \mid X \in X_{\kappa,\epsilon}^c\right) \mathbb{P}(X \in \chi_{\kappa,\epsilon}^c),$$

then it follows that $\mathbb{P}\left(X \in X_{\kappa,\epsilon}^c\right) \leq \frac{\delta}{\epsilon}$. As a result we conclude that the probability of drawing a $(\kappa, \epsilon)$-good training sample is at least $1 - \frac{\delta}{\epsilon}$. $\qquad\square$

Based on Lemma B.2, the following lemma bounds the probability that $\Gamma_p$ is sufficiently large for our purposes. In particular, for our result on non-overfitting we require $|\Gamma_p| = m$, while for our benign overfitting result only that $|\Gamma_p| \geq (1 - \alpha)m$ for some constant $\alpha \in (0, 1)$.

**Lemma B.3.** *Suppose $n \geq 15k$, $2\lambda_w \leq \eta(\gamma - \rho)$, $\gamma \geq 2\rho$ and $p \in \{-1, 1\}$. Then for sufficiently large $n$ there exists a constant $c > 0$ such that the following are true.*

1. $\mathbb{P}\left(|\Gamma_p| = m\right) = 1 - m\exp(-cn)$.

2. *With $\alpha \in (0, 1)$ a constant such that $\alpha m \in [m]$, then*

$$\mathbb{P}\left(|\Gamma_p| \geq (1 - \alpha)m\right) \geq 1 - \exp(-cn).$$

*Proof.* As here we only ever consider the activations at initialization, for convenience we drop the argument parentheses "$(0,1)$" on the counting functions: in particular we write $T_j(0,1)$ as $T_j$ and $B_j(0,1)$ as $B_j$. Suppose $(j)^{-1} = p$, under the assumption $\lambda_w \leq \eta(\gamma - \rho)$ then if

$$G_j(\gamma - \rho) - B_j(\gamma + \rho) > (\gamma - \rho) \geq \frac{2\lambda_w}{\eta}$$

we may conclude $j \in \Gamma_p$. Rearranging this expression, equivalently $j \in \Gamma_p$ if

$$(1 + B_j)\gamma < (\gamma - \rho)T_j.$$

As a result, membership to $\Gamma_p$ is guaranteed as long as $T_j > 0$ and $B_j < \frac{\gamma - \rho}{2\gamma}T_j$. Note by the assumptions of the lemma $\mu := \frac{2k}{n+k} \leq \frac{1}{8}$ and $\frac{\gamma - \rho}{2\gamma} \geq \frac{1}{4}$. Conditioning on the event we draw a $(\varepsilon, \frac{1}{4})$-good training sample then the probability that $j \notin \Gamma_p$ is at most $\epsilon$ by Lemma B.2. Furthermore, with the training sample fixed the activations of each neuron are mutually independent. Let $X = (\mathbf{x}_i)_{i=1}^n$ denote the draw of the training sample and

$$\chi_{\kappa,\epsilon} = \{X : \mathbb{P}_w((B \geq \kappa T) \cup (T = 0)) \leq \epsilon\}$$

the set of $(\varepsilon, \frac{1}{4})$-good training samples. Let $\epsilon = \exp(-cn)$, where $c$ in what follows is a sufficiently small constant. By the assumptions of the lemma

$$\mathbb{P}\left(X \in X_{1/4,\epsilon}\right) \geq 1 - \frac{\delta}{\epsilon} \geq 1 - 2\exp\left(-\frac{n}{1024} + cn\right) \geq 1 - 2\exp\left(-cn\right).$$

For the first result, using the union bound

$$\begin{aligned}
\mathbb{P}\left(|\Gamma_p| = m\right) &\geq \mathbb{P}\left(|\Gamma_p| = m \mid X \in X_{1/4,\epsilon}\right)\mathbb{P}\left(X \in X_{1/4,\epsilon}\right) \\
&\geq (1 - m\exp(-cn))\left(1 - 2\exp\left(-cn\right)\right) \\
&\geq 1 - m\exp(-cn)
\end{aligned}$$

as claimed. For the second result observe

$$\begin{aligned}
\mathbb{P}\left(|\Gamma_p| < (1-\alpha)m \mid X \in X_{1/4,\epsilon}\right) &= \mathbb{P}\left(\exists \mathcal{J} \subset [2m]_p, |\mathcal{J}| = \alpha m : j \notin \Gamma_p \;\forall j \in \mathcal{J} \mid X \in X_{\kappa,\epsilon}\right) \\
&\leq \binom{m}{\alpha m}\epsilon^{\alpha m} \\
&\leq \left(\frac{\epsilon e}{\alpha}\right)^{\alpha m}.
\end{aligned}$$

As $\alpha$ is a constant again there is a sufficiently small constant $c$ such that

$$\frac{\epsilon e}{\alpha} = \exp(-cn + 1 + \log(1/\alpha)) \leq \exp(-cn).$$

Therefore, there exists a sufficiently small constant $c$ such that

$$\begin{aligned}
\mathbb{P}\left(|\Gamma_p| \geq (1-\alpha)m\right) &\geq \mathbb{P}\left(|\Gamma_p| \geq (1-\alpha)m \mid X \in X_{1/4,\epsilon}\right)\mathbb{P}\left(X \in X_{1/4,\epsilon}\right) \\
&\geq (1 - \exp\left(-cmn\alpha\right))\left(1 - 2\exp\left(-cn\right)\right) \\
&\geq 1 - \exp(-cn)
\end{aligned}$$

as claimed. $\qquad\square$

**Lemma B.4.** *Assume $\gamma + \rho \leq \frac{1}{1.02(n-k)}$, $n - k \geq 2 \times 10^6$, $|\Gamma_p| > 0.99m$ for $p \in \{-1, 1\}$ and $m \geq C\log(n)$ for a sufficiently large constant $C$. Then with probability at least $1 - \frac{C}{n}$ for all $i \in \mathcal{S}_F$ there exists a $j \in \Theta_{y_i}$ such that $i \in \mathcal{A}_j^{(0)}$.*

*Proof.* For convenience in what follows we use $G_j$ and $B_j$ for $G_j(0,1)$ and $B_j(0,1)$ respectively. For a neuron to be in $\Theta_p$ it must satisfy the following condition,

$$G_j(\gamma + \rho) - B_j(\gamma - \rho) < 1 - \gamma + \rho,$$

or alternatively

$$G_j < \frac{1 - \gamma + \rho}{\gamma + \rho} = \frac{1}{\gamma + \rho} - \frac{\gamma - \rho}{\gamma + \rho}.$$

232 By our assumptions this is in turn implied by the following condition $G_j < 1.02(n - k) - 1$ or
233 $G_j < 1.01(n - k)$ for $n - k > 100$.

234 Let $P(i, l)$ be the probability that an arbitrary random neuron is active on points $i$ and $l$. Consider
235 independently drawing two points from the distribution over points with either positive or negative
236 signal sign component, and let $p$ be the probability that an arbitrary random neuron is active on both
237 points. Similarly let $q$ be the probability that an arbitrary neuron is active on two points which are
238 drawn with opposite signal signs. By rotational invariance of the weight distribution the probability a
239 random neuron activates on a point is $1/2$, therefore $\mathbb{E}(G_j) = n - k$ and furthermore $P(i, i) = 1/2$.
240 In addition, by writing $G_j$ as a sum of indicator functions, expanding, and using the linearity of
241 expectation we have

$$\mathbb{E}(G_j^2) = \sum_{(i,l) \in [n-k] \times [n-k]} P(i, l) = \frac{1}{2} 2(n - k) + 2(n - k)^2 q + 2(n - k)(n - k - 1)p.$$

242 Recall $i$ and $l$ index over the clean points. Observe by construction that for $i \not\sim l$ then $-\mathbf{x}_l \overset{d}{=} \mathbf{x}_i$. As
243 a result, using an abuse of notation where the index $-j$ indicates the point $-\mathbf{x}_j$, then for $i \sim l \not\sim j$
244 we have $P(i, j) = P(i, -l)$. If a neuron activates on $\mathbf{x}_l$ iff it does not activate on $-\mathbf{x}_l$, therefore
245 $P(i, l) + P(i, -l) = P(i, i) = 1/2$ and hence we conclude $p + q = 1/2$. As a result

$$\mathbb{E}(G_j^2) = (n - k) + 2(n - k)\left((n - k)q + (n - k)p - p\right)$$

$$= (n - k) + 2(n - k)\left(\frac{1}{2}(n - k) - p\right)$$

$$\leq (n - k) + (n - k)^2.$$

246 As $\mathbb{E}(G_j) = n - k$, it follows that $G_j$ has variance $n - k$. Therefore by Chebyshev's inequality

$$\mathbb{P}(G_j \geq 1.01(n - k)) \leq \frac{10^4}{n - k}.$$

247 Therefore a given random neuron $j$ satisfies the condition $G_j < 1.01(n - k)$ with failure probability
248 at most $\frac{10^4}{n-k}$. Applying Markov's inequality

$$\mathbb{P}\left(\sum_{j=1}^{2m} \mathbb{1}\left(G_j \geq 1.01(n - k)\right) \geq \frac{2m}{100}\right) \leq \frac{10^6}{2(n - k)}$$

249 and therefore

$$\mathbb{P}\left(\sum_{j=1}^{2m} \mathbb{1}\left(G_j < 1.01(n - k)\right) \geq 1.998m\right) \geq 1 - \frac{10^6}{2(n - k)}.$$

250 Now by a Chernoff bound, there exists a small constant $c > 0$ such that with probability at least
251 $1 - \exp(-cm)$, a fixed training point is activated by at least $1/3$ of the neurons of each sign. Therefore,
252 using the union bound every training point is activated by at least $m/3$ of the neurons with probability
253 at least $1 - n \exp(-cm) \geq 1 - \exp(-cm)$ using that $\log n \leq O(m)$.

254 Let $\Lambda := \{j \in [2m] : G_j < 1.01(n - k)\}$ and observe that if $j \in \Gamma_p$ and $j \in \Lambda$ then $j \in \Theta_p$.
255 Condition on two further events in addition to $|\Gamma_p| \geq 0.99m$ for $p \in \{-1, 1\}$: $|\Lambda| \geq 1.998m$ and for
256 all $i \in \mathcal{S}_F$ then $|\mathcal{I}_i(0) \cap \{j : (-1)^j = y_i\}| \geq m/3$ and for any $p \in \{-1, 1\}$. Then with probability
257 one we have for all $i \in \mathcal{S}_F$ that there exists a $j \in \Theta_{y_i}$ such that $i \in \mathcal{A}_j^{(0)}$. The probability that these
258 two conditions hold is at least

$$1 - \frac{C}{n} - \exp(-cm).$$

259 Supposing that $m \geq C \log(n)$ for a sufficiently large constant $C$ then this probability can in turn be
260 lower bounded as

$$1 - \frac{C}{n}$$

261 as claimed.

262 $\square$

The final lemma we provide here states, under mild conditions on the network width, that with high probability every point in the training sample activates a neuron whose output weight matches its label in sign. We will use this to prove our result on non-benign overfitting, which is discussed in Section E.

**Lemma B.5.** *Let $\delta \in (0, 1)$, then if $m \geq \log_2(\frac{2n}{\delta})$

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

## D  Benign overfitting

**Assumption 1.** *Let $\delta \in (0,1)$. With $n \geq C/\delta$ and $\eta \leq c$ then for benign overfitting we assume the following conditions on the other data and model hyperparameters.*

1.  $k < \frac{n}{100}$,

2.  $5\sqrt{2\log(2en^2/\delta)/d} \leq \gamma \leq \frac{4}{5n}$

3.  $\lambda_w \leq \frac{\eta\gamma}{4}$,

4.  $m \geq C\log(n)$

5.  $d \geq 3\rho^{-2}\ln\left(\frac{6n^2}{\delta}\right)$ *where* $\rho \leq \frac{\gamma}{5}$

We remark that under these conditions then for sufficiently large $n$, $\rho$ as defined above clearly satisfies the inequalities $\rho \leq \min\left\{\frac{n-3k}{n+k}\gamma, \frac{1}{6(n-k)}\right\}$ and $\gamma + \rho < \min\left\{\sqrt{\frac{1}{4(n-k)k}}, \frac{1}{n-k}, \frac{1}{99k}, \frac{1}{100}\right\}$. In addition to the assumptions detailed in Assumption 1, for convenience we assume three further conditions hold.

**Assumption 2.** *In addition to the assumptions detailed in Assumption 1, assume the following conditions hold.*

1.  $|\Gamma_p| > 0.99m$ *for* $p \in \{-1, 1\}$.

2.  *For all $i \in \mathcal{S}_F$ there is $j \in \Gamma_{y_i}$ such that $i \in \mathcal{A}_j^{(0)}$.*

3.  *For all $i,l \in [2n]$, $i \neq l$ $|\langle \mathbf{n}_i, \mathbf{n}_l \rangle| \leq \frac{\rho}{1-\gamma}$.*

As shown in the following lemma, these two additional conditions hold with high probability over the randomness of the initialization and training set.

**Lemma D.1.** *The extra conditions of Assumption 2 hold with probability at least $1 - \delta$.*

*Proof.* Using Lemma B.3, then for sufficiently large $n$ there exists a constant $c$ such that the probability the first condition does not hold is at most $\exp(-cn)$. Alternatively, setting $\delta \geq 3\exp(-cn)$ and rearranging, as long as $n \geq C\ln\left(\frac{3}{\delta}\right)$ then the probability the first condition does not hold is at most $\delta/3$. Using Lemma B.4 then the probability condition two does not hold is at most $C/n$ for some large constant $C$, therefore as long as $n \geq C/delta$ then the probability the second condition does not hold is at most $\delta/3$. Using Lemma B.1, observing $\frac{\rho}{1-\gamma} > \rho$ and under the conditions of the lemma that $3\rho^{-2}\ln\left(6n^2\right)) > 6$, then as long as

$$d \geq 3\rho^{-2}\ln\left(\frac{6n^2}{\delta}\right)$$

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

$$

*Proof.* Under Assumption 2 Statement 1 and 2 are derived from Lemma D.12. The bound in statement 1 comes from Lemma D.2 applied from iteration 0 to iteration $\mathcal{T}_{\text{end}}$, using $4k(n-k)(\gamma + \rho)^2 < 4/99$ and $\ell(0, \mathbf{x}_i) = 1 + O(\eta)$ for all $i \in [2n]$. Statement 3 comes from Lemma D.13. Finally, under Assumption 1 then by Lemma D.1 Assumption 2 holds probability at least $1 - \delta$. $\qquad \square$

# E  Non-benign overfitting

**Assumption 3.** *Let $\delta \in (0, 1)$. For the non-benign overfitting setting we assume the following conditions on the data and model hyperparameters.*

*1. $\gamma \leq \frac{1}{6\sqrt{dn}}$*

2. $\eta \le \frac{1}{2mn}$,

3. $\lambda_w \le \min\{n\gamma/10, \eta/4\}$,

4. $m \ge \log_2(\frac{4n}{\delta})$

5. $d \ge \max\left\{6, 3\rho^{-2}\ln\left(\frac{4n^2}{\delta}\right)\right\}$ *where* $\rho \le \min\{\frac{1-\gamma}{4n}, \frac{1}{3n} - \gamma\}$.

For convenience, in our analysis we will make two additional assumptions.

**Assumption 4.** *In addition to the conditions detailed in Assumption 4, assume the following two conditions hold.*

1. *For all $i \in [2n]$ there exists a $j \in [2m]$ such that $(-1)^j = y_i$ and $i \in \mathcal{A}_j^{(0)}$.*

2. *For all $i, l \in [2n]$, $i \ne l$ $|\langle \mathbf{n}_i, \mathbf{n}_l\rangle| \le \frac{\rho}{1-\gamma}$.*

As demonstrated in the following Lemma, these additional two conditions hold with high probability over the randomness of the initialization and training set.

**Lemma E.1.** *The additional conditions of Assumption 4 hold with probability at least $1 - \delta$.*

*Proof.* The additional conditions of Assumption 4 over Assumption 3 are as follows.

1. For all $i \in [2n]$ there exists a $j \in [2m]$ such that $(-1)^j = y_i$ and $i \in \mathcal{A}_j^{(0)}$.

2. For all $i, l \in [2n]$, $i \ne l$ $|\langle \mathbf{n}_i, \mathbf{n}_l\rangle| \le \frac{\rho}{1-\gamma}$.

Using Lemma B.5, then as long as $m \ge \log_2(\frac{4n}{\delta})$ the probability the first condition does not hold is at most $\delta/2$. Using Lemma B.1, and observing $\frac{\rho}{1-\gamma} > \rho$, then as long as

$$d \ge \max\left\{6, 3\rho^{-2}\ln\left(\frac{4n^2}{\delta}\right)\right\}$$

the probability that the second condition does not hold is also at most $\delta/2$. Using the union bound we conclude that both properties hold with probability at least $\delta$. $\qquad\square$

### E.1   Proof of Lemma 3.7

**Lemma E.2** (Lemma 3.7)**.** *Assume Assumption4 and that $\ell(t_0, \mathbf{x}_i) \le a$ for all $i$. Then*

$$T(t_0, t) \le \frac{2n}{1 - (2n-1)(\gamma + \rho)}\left(\frac{a}{\eta} + 3m\right).$$

*Proof.* From Lemma C.5, $\phi(\rho - \gamma) \le \rho + \gamma$, and the assumption on $a$,

$$T_i(t_0, t) \le \frac{a}{\eta} + 3m + (\gamma + \rho)T^{(i)}(t_0, t).$$

If we sum over $i$, we get

$$T(t_0, t) \le \frac{2na}{\eta} + 6mn + (2n-1)(\gamma + \rho)T(t_0, t),$$

from which the result follows. $\qquad\square$

### E.2   Additional lemmas

**Lemma E.3.** *If Assumption 4 holds, then the training process converges to zero loss.*

*Proof.* By Lemma E.2, there is an upper bound on the number of updates independent of epoch. This can only happen if there is some epoch after which we make no updates. In turn, this can only happen if every point is either at zero loss or activates no neurons. We prove by induction that every point activates a neuron each epoch $t = 0$. Fix a point $i$.

At $t = 0$ this is true by initialization. Now suppose it is true at epoch $t$. There are two cases to consider to show this for epoch $t + 1$:

1. If $\ell(t, \mathbf{x}_i) > 0$, then let $j$ be such that $\phi(\langle \mathbf{w}_j^{(t)}, \mathbf{x}_i \rangle) > 0$. We can bound

$$
\begin{aligned}
\phi(\langle \mathbf{w}_j^{(t+1)}, \mathbf{x}_i \rangle) &\geq \phi(\langle \mathbf{w}_j^{(t)}, \mathbf{x}_i \rangle) + \eta - \eta(\gamma + \rho)H_j^{(i)}(t, t+1) \\
&\geq \phi(\langle \mathbf{w}_j^{(t)}, \mathbf{x}_i \rangle) + \eta[1 - (\gamma + \rho)(2n - 1)] \\
&> 0,
\end{aligned}
$$

   since $\gamma + \rho < \frac{1}{2n-1}$ (Assumption 4).

2. If $\ell(t, \mathbf{x}_i) = 0$, then

$$
\begin{aligned}
y_i f(t, \mathbf{x}_i) = y_i \sum_{j=1}^{2m} (-1)^j \phi(\langle \mathbf{w}_j^{(t)}, \mathbf{x}_i \rangle) \\
\leq \sum_{j\,:\,(-1)^j = y_i} \phi(\langle \mathbf{w}_j^{(t)}, \mathbf{x}_i \rangle)
\end{aligned}
$$

   is bounded below by 1. This means that there is some $j$ such that $\phi(\langle \mathbf{w}_j^{(t)}, \mathbf{x}_i \rangle) \geq \frac{1}{m}$. We bound

$$
\begin{aligned}
\phi(\langle \mathbf{w}_j^{(t+1)}, \mathbf{x}_i \rangle) &\geq \phi(\langle \mathbf{w}_j^{(t)}, \mathbf{x}_i \rangle) - \eta(\gamma + \rho)H_j^{(i)}(t, t+1) \\
&\geq \frac{1}{m} - \eta(\gamma + \rho)(2n - 1) \\
&> 0
\end{aligned}
$$

   since $\eta < \frac{1}{2mn}$ (Assumption 4). $\qquad\square$

**Lemma E.4.** *Suppose that at epoch $\tau$ every point is at zero loss. Then we can bound*

$$
\eta T(0, \tau) \geq n + O(\eta).
$$

*Proof.* If $\ell(\tau, \mathbf{x}_i) = 0$ for all $i$, then $y_i f(\tau, \mathbf{x}_i) \geq 1$ for all $i$ as well. We bound

$$
\begin{aligned}
y_i f(\tau, \mathbf{x}_i) &\leq \sum_{j\,:\,(-1)^j = y_i} (\phi(\langle \mathbf{w}_j^{(\tau)}, \mathbf{x}_i \rangle) - \phi(\langle \mathbf{w}_j^{(0)}, \mathbf{x}_i \rangle)) + O(\eta) \\
&\leq \sum_{j\,:\,(-1)^j = y_i} \eta[T_{ij}(0, \tau) + (\gamma + \rho)H_j^{(i)}(0, \tau)] + O(\eta) \\
&\leq \eta T_i(0, \tau) + \eta(\gamma + \rho)T^{(i)}(0, \tau) + O(\eta).
\end{aligned}
$$

If we sum over $i$, we see that

$$
2n \leq \eta[1 + (2n - 1)(\gamma + \rho)]T(0, \tau) \leq 2\eta T(0, \tau) + O(\eta),
$$

from which the desired result follows. $\qquad\square$

**Lemma E.5.** *Let $y \in \{-1, 1\}$ chosen uniformly and $\mathbf{x} := (y\sqrt{\widetilde{\gamma}}\mathbf{v}, \sqrt{1 - \widetilde{\gamma}}\mathbf{n})$, where $\mathbf{n} \sim \mathrm{Uniform}(\mathcal{

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

## F  Non-overfitting

**Assumption 5.** *Let $\delta \in (0, 1)$. With $n \geq C$ and $\eta \leq c$ then for no overfitting we assume the following conditions on the data and model hyperparameters.*

    *1. $k < \frac{n}{100}$,*

    *2. $\frac{3}{n} < \gamma < \frac{1}{36}\min\{k^{-1}, 1\}$,*

    *3. $\lambda_w \leq \frac{\eta\gamma}{4}$*

    *4. $m \geq C\ln\left(\frac{2n}{\delta}\right)$,*

    *5. $d \geq 3\rho^{-2}\ln\left(\frac{4n^2}{\delta}\right)$ where $\rho \leq \frac{\gamma}{2}$.*

We remark that under these assumptions $\rho$ as given above satisfies the inequality $\rho < \min\left\{\frac{\gamma(n-3k)-2}{n+k}, \frac{\gamma}{5}, \frac{n}{11}\right\}$. For convenience, we will also make two additional assumptions.

**Assumption 6.** *In addition to the assumptions detailed in Assumption 5, assume the following conditions hold.*

    *1. $\Gamma = [2m]$.*

    *2. For all $i, l \in [2n]$ such that $i \neq l$ then $|\langle \mathbf{n}_i, \mathbf{n}_l \rangle| \leq \frac{\rho}{1-\gamma}$.*

As shown in the following lemma, these two additional conditions hold with high probability over the randomness of the initialization and training set.

**Lemma F.1.** *The extra conditions of Assumption 6 hold with probability at least $1 - \delta$*

 *Proof.* Recall the additional conditions of Assumption 6 over Assumption 5 are as follows,

1. $\Gamma = [2m]$,

2. for all $i, l \in [2n]$, $i \neq l$ $|\langle \mathbf{n}_i, \mathbf{n}_l \rangle| \leq \frac{\rho}{1-\gamma}$.

Using Lemma B.3, then for sufficiently large $n$ there exists a constant $c$ such that the probability the first condition does not hold is at most $m \exp(-cn)$. Alternatively, setting $\delta \geq 2m \exp(-cn)$ and rearranging, as long as $m \geq C \ln \left( \frac{2n}{\delta} \right)$ then the probability the first condition does not hold is at most $\delta/2$.

Using Lemma B.1, observing $\frac{\rho}{1-\gamma} > \rho$ and under the conditions of the lemma that $3\rho^{-2} \ln \left( 4n^2 \right) > 6$, then as long as

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

## F.3 Proof of Lemma 3.11

**Lemma F.4** (Lemma 3.11). *Let Assumption 6 hold. Suppose there is a time $t_0$ so that:*

  *a. $\ell(t_0, \mathbf{x}_i) \leq a$ for all $i \in \mathcal{S}_T$,*

  *b. $\phi(\langle \mathbf{w}_j^{(t_0)}, \mathbf{x}_i \rangle) \leq b$ for all $i \in \mathcal{S}_F$ and $i \sim j$,*

  *c. For all iterations $\tau$ satisfying $t_0 \leq \tau \leq t$ it holds that $i \in \mathcal{A}_j^{(\tau)}$ only if $i \sim j$,*

720      *d. For all iterations $\tau$ satisfying $t_0 \le \tau \le t$, $i \in \mathcal{A}_j^{(\tau)}$ if $i \sim j$ and $i \in \mathcal{S}_T$.*

721   *We can bound*

$$B_j(t_0, \tau) \le k\left(\frac{3b}{2\eta} + \frac{2a}{\eta m}\right)$$

$$\sum_{j \sim s} G_j(t_0, t) \le \frac{1}{\gamma}\left(\frac{3a}{2\eta} + \frac{mb}{\eta}\right).$$

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

 are guaranteed by Assumption 6, so the point is correctly classified. $\qquad\square$

## F.5 Proof of Theorem 3.8

**Theorem F.7** (Theorem 3.8). *Assume Assumption 5 holds. With probability at least $1 - \delta$ over the randomness of the dataset and network initialization we have the following.*

    *1. The training process terminates at an iteration $\mathcal{T}_{end} \leq \frac{Cn}{\eta}$.*

2. *For all $i \in \mathcal{S}_T$ then $\ell(\mathcal{T}_{end}, \mathbf{x}_i) = 0$ while for all $i \in \mathcal{S}_F$ $\ell(\mathcal{T}_{end}, \mathbf{x}_i) = 1$.*

3. *The generalization error satisfies*

$$\mathbb{P}(\text{sgn}(f(\mathcal{T}_{end}, \mathbf{x})) \neq y) \leq C \exp\left(-\frac{d}{2n^2}\right).$$

*Proof.* Under Assumption 6, statements (1) and (2) follow from Lemma F.5. The bound on $\mathcal{T}_{end}$ follows from Lemma F.4 applied at $t_0 = 2$:

$$B(2, \mathcal{T}_{end}) \leq k\left(\frac{4}{\eta m}\right) + O(\eta)$$

$$B(2, \mathcal{T}_{end}) \leq \frac{2}{\gamma}\left(\frac{3}{2\eta}\right) + O(\eta)$$

with $\gamma > \frac{3}{n}$. Statement (3) is derived from Lemma F.6. Finally, Lemma F.1 implies that under Assumption 5 then Assumption 6 holds with probability at least $1 - \delta$. □

# G    Numerical simulations

**Reproducibility statement:** the code used to generate the following figures can be found at `https://anonymous.4open.science/r/benign_overfitting-4A4C/BO_experiments.ipynb`.

To investigate our theory we train two-layer neural networks with ReLU activations using full-batch gradient descent and a fixed step size. We train on the synthetic binary classification dataset, detailed in Section B.2, that we have studied throughout the paper. Finally we train using both hinge and logistic loss.

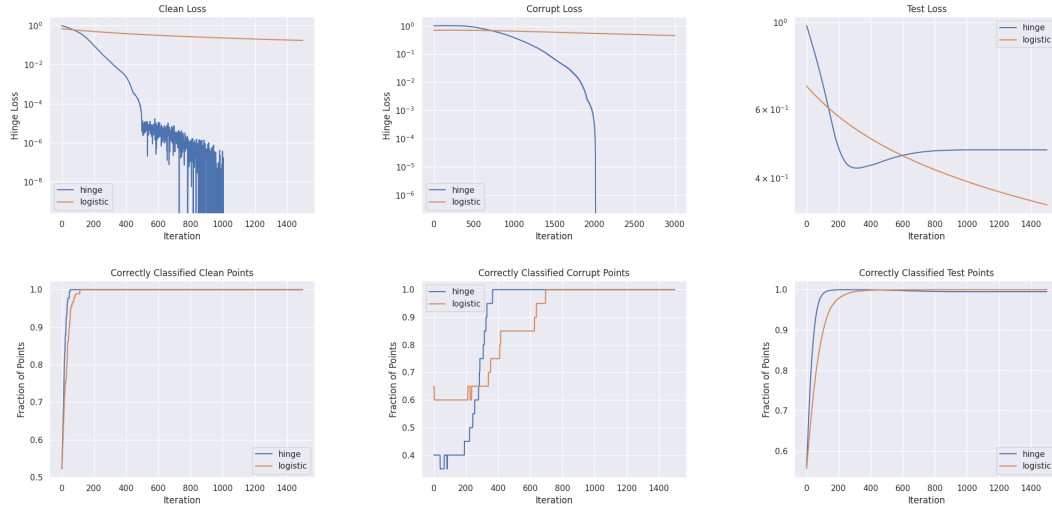

Figure 1: From left to right, the first row shows the clean, corrupt, and test losses as a function of epoch. The second row shows the fraction of clean, corrupt, and test points that are classified correctly. These plots were generated for $n = 100$, $d = 800$, $k/n = 0.1$, $m = 100$, $\gamma = 0.015$, and using gradient descent with a step size of $0.01$.

In Figure 1 we call attention to the difference in the training dynamics of hinge loss versus logistic loss. Perhaps the key difference between hinge loss and logistic loss is that the contributions from any given point do not get smaller as the point approaches $0$ loss. Furthermore, unlike with the logistic loss points can actually attain zero hinge loss after a finite number of epochs. Once they do attain zero loss, they cease to contribute to the update of the network parameters. If the remaining active points push the parameters in such a way as to increase the loss on a given point then it will reactivate. As a result points close to zero hinge loss periodically activate and deactivate giving rise to the chaotic

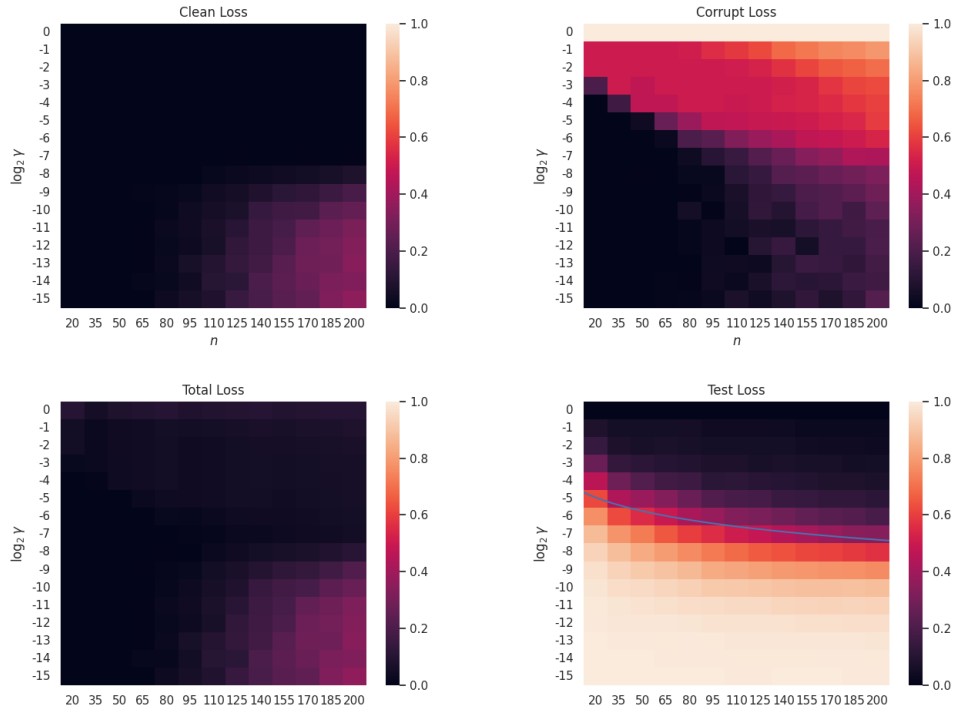

Figure 2: In the top row we show the loss on the clean training points and the corrupt training points after training. In the bottom row we show the total loss after training, along with the test loss on 10000 random generated points after training. For each plot we set $d = 1000, m = 30, \eta = 0.005$ and train for 5000 iterations of gradient descent using hinge loss. In each plot we vary $\gamma$ and $n$ and hold the fraction of corrupt points constant at $0.05$. In the bottom right plot we also graph $c/n$ for $c \approx 0.6$

behavior observed as the training loss approaches zero. We note that managing this behavior required a careful analysis that is distinct from the analysis for logistic loss.

In Figure 2 we call particular attention to the bottom right plot. Our theory predicts a phase transition between benign overfitting and non-benign overfitting when $\gamma \approx c/n$: the phase transition we observe empirically in the bottom-right heatmap suggests this estimate is reasonable. With regard to the hinge loss over the corrupt points, displayed in the top-right heatmap, we observe another phase transition, this time between overfitting and non-overfitting. The top and bottom heatmaps of the left-hand column display the hinge loss over the clean training set and total training set respectively, these appear very similar due to the fact that clean points make up $95\%$ of the training set. The clean points fail to achieve zero, or close to zero, hinge loss only when $\gamma$ is small and $n$ is large. As stated in the caption, in these experiments $d$ is fixed and thus as $n$ increases the near-orthogonality condition we require on the noise components in order to prove convergence to zero clean loss is compromised. As a result, when $\gamma$ is small and the correlations between noise vectors is potentially large it is possible for pairs of points with opposite labels to be significantly correlated.