# OpenReview forum: "Training shallow ReLU networks on noisy data using hinge loss: when do we overfit and is it benign?"
_NeurIPS.cc/2023/Conference — NeurIPS 2023 spotlight_

### Official Review · Reviewer_NX3e · 2023-06-17

**Soundness:** 3 good
**Presentation:** 2 fair
**Contribution:** 3 good
**Rating:** 7
**Confidence:** 3

**Summary:**

The paper studies overfitting when training shallow ReLU networks with the hinge loss using noisy data (i.e., with label flipping noise). They consider a setting where the data points consist of a direction $\mathbf{v}$ that determines the (clean) labels, and noise from a sphere orthogonal to $\mathbf{v}$. This distribution is similar to the two-clusters setting that has been studied in previous works [Frei et al. 2022, Xu and Gu 2023, Chatterji and Long 2021], except that the assumptions on the noise in the current work is stronger. The authors identify conditions on the signal strength that result in benign overfitting, harmful overfitting, and “non overfitting” (where only the clean training examples are classified correctly, and the test error on clean data is small). The results follow by analyzing the trajectory of gradient descent.

**Strengths:**

Understanding the conditions where benign overfitting occurs is an intriguing question that has be studied extensively in recent years. While the primary motivation for studying benign overfitting is to better understand interpolation learning in neural networks, most of the known results consider linear models (or kernel regression). The current paper helps understanding benign/harmful overfitting in shallow fully-connected ReLU networks trained with the hinge loss. Given our limited understanding of benign overfitting in nonlinear networks, I believe that the contribution of the current paper is significant.

I did not verify the correctness of the proofs, which are in the appendix.

**Weaknesses:**

- Assumption 1 in the main text is different from Assumption 1 in the appendix. Specifically, condition 5 of Assumption 1 in the appendix includes a parameter $\rho$ that does not appear in the assumption in the main text. I think that the authors should discuss the relationship between these two versions of Assumption 1.
- Related to the above: Note that in lines 221-222 in the main text, the authors claim that the condition on the correlations between $\mathbf{n}_i,\mathbf{n}_l$ follows from Assumption 1 by letting $\rho \leq \frac{\gamma}{C}$. This is true if we use Assumption 1 from the appendix, but it does not follow from Assumption 1 in the main text. Please explain how/whether the proof holds using the assumption from the main text. This issue appears also in the other proof sketches.
- The authors claim that there are only two papers about benign overfitting in nonlinear fully-connected neural networks: Frei et al. 2022, and Xu \& Gu 2023. The authors provide a nice comparison between the current results and these prior results. It is worth noting that there is an additional prior work on benign overfitting in nonlinear fully-connected networks: “Benign Overfitting in Linear Classifiers and Leaky ReLU Networks from KKT Conditions for Margin Maximization” by Spencer Frei, Gal Vardi, Peter Bartlett and Nati Srebro. This paper appeared in arxiv only 2.5 months before the NeurIPS deadline, so it is reasonable that the authors were not aware of it when writing the current paper, but I think that a comparison to this paper would be helpful.
- The writing is quite sloppy. Below I specify some typos and minor comments:
    - Line 130, in the definition of $\mathcal{F}^{(t)}$: should be $>0$ instead of $<1$.
    - Line 132: “j=th”
    - Line 155 — “constant that depends on $n,m,k,\gamma,d$”: if we think about all of these parameters as constants, then what isn’t a constant?
    - Line 156: (5) should be (4)
    - Line 227: is the notation $[m]_p$ defined? Also, you probably meant $[2m]_p$.
    - Line 385 in the appendix: “delta”
    - The word “overfitting” sometimes means “interpolating also the noisy examples” and sometimes means “harmful overfitting”. I would write explicitly “harmful overfitting” when talking about the latter.


**Questions:**

See the “weaknesses” section.

**Limitations:**

The authors discussed the limitations.

---

> ### Author Rebuttal · Authors · 2023-08-08
>
> We thank the reviewer for their positive review and thoughtful questions which we address below.
>
> - *"Assumption 1 in the main text is different from Assumption 1 in the appendix. Specifically, condition 5 of Assumption 1 in the appendix includes a parameter that does not appear in the assumption in the main text."* The two assumptions labeled Assumption 1 were not supposed to be directly related, and we apologize for the confusion. Assumption 1 in the main text sketches the common setting for the three main results in the paper, while Assumption 1 in the supplementary details specific relationships between the parameters of the neural network and data model that result in our benign overfitting result. We will clarify this better in future versions.
>
> - *"in lines 221-222 in the main text, the authors claim that the condition on the correlations between $n_i, n_l$ follows from Assumption 1 by letting $\rho \leq \frac{\gamma}{C}$. This is true if we use Assumption 1 from the appendix, but it does not follow from Assumption 1 in the main text. Please explain how/whether the proof holds using the assumption from the main text. This issue appears also in the other proof sketches."* We apologize for this confusion and will clarify it in future versions. In the lines in question, as well as the corresponding parts in the other proof sketches, we only mean that there exists a $\rho$ that simultaneously satisfies $\rho \leq \frac{\gamma}{C}$ and $|\langle n_i, n_l \rangle | \leq \frac{\rho}{1-\gamma}$ for all $i$ and $l$.  This follows from Assumption 1 in the main text by taking $\rho = \frac{\gamma}{C}$ and applying inequality (4) in Assumption 1 to Lemma B.1 in the supplementary materials.
>
> - *"It is worth noting that there is an additional prior work on benign overfitting in nonlinear fully-connected networks: 'Benign Overfitting in Linear Classifiers and Leaky ReLU Networks from KKT Conditions for Margin Maximization'... it is reasonable that the authors were not aware of it when writing the current paper, but I think that a comparison to this paper would be helpful."* Thank you for bringing this very interesting, recent and highly relevant paper to our attention. This work studies a setting similar to that considered in the papers we make a direct comparison against. In the context of Leaky ReLU networks, this work shows that gradient flow exhibits an implicit bias towards solutions that satisfy KKT conditions. Under certain assumptions on the data, similar to those in our work and the existing literature, they are able to show that parameters satisfying the KKT points result in networks which exhibit benign overfitting. With regard to differences we remark that while we consider hinge loss the work highlighted considers an exponentially tailed loss. Furthermore while they study gradient flow we consider gradient descent.
>
> - *"Line 155 — 'constant that depends on $n, m, k, \gamma, d$': if we think about all of these parameters as constants, then what isn't a constant?"* Apologies this was not well put, we mean only that $\eta$ depends on these parameters in an as of yet unspecified way. We will clarify this in future versions.
>
> - *"The word 'overfitting' sometimes means 'interpolating also the noisy examples' and sometimes means 'harmful overfitting'. I would write explicitly 'harmful overfitting' when talking about the latter."* We agree with the reviewer that this vocabulary is probably preferable to the one we used. As it currently stands we use non-benign (harmful does seem a better choice) overfitting and benign overfitting to distinguish bad interpolation from benign interpolation.  While regrettably in some instances we have used ``overfitting'' to refer to harmful overfitting, we have corrected this in our current draft.
>
> - *"Line 155 — 'constant that depends on $n, m, k, \gamma, d$': if we think about all of these parameters as constants, then what isn't a constant?"* None of the listed parameters should be thought of as constant and we apologize for the confusion.  We only mean that the upper bound for $\eta$ depends on these parameters in an unspecified way.
>
> - In regard to typos, the comments with regards to lines 130, 132, 156 and 385 are correct, we apologize for these and will fix them. In regard to line 227 it should be $[2m]$.

---

> > ### Comment · Reviewer_NX3e · 2023-08-12
> >
> > Thanks for the response

---

### Official Review · Reviewer_8QyX · 2023-06-29

**Soundness:** 3 good
**Presentation:** 3 good
**Contribution:** 3 good
**Rating:** 7
**Confidence:** 3

**Summary:**

The paper studies training a noisy dataset with label noise using GD on a 2-layer network, without biases, and where the second layer is fixed. The main results are, that under the data distribution defined in the paper (namely, linearly separable with orthogonal noise), different types of overfitting (or no fitting) occur when changing the margin.

**Strengths:**

- I think the main three results are very interesting, and shed light on the role of the margin for determining the types of overfitting which occurs.
- AFAIK the main three results are novel, as they study benign overfitting for neural networks under hinge loss, whereas previous works either studied classification losses (e.g. log loss) or kernel methods.
- The proof techniques also seem interesting and employ combinatorial arguments about how the iterations change the activation pattern of the network.
- The proof intuitions are very helpful and elaborative.


**Weaknesses:**

I think overall this is a nice paper that should be accepted, however, there are some weaknesses/questions which I would be happy to see the author’s response about.

- I think the main weakness of the paper is the limited and very specific setting to which it is applied. This is acknowledged by the authors, however, I still think it should be further addressed.
- The network doesn’t contain biases and only the weights of the first layer are trained. Regarding the bias terms, I am not sure how adding them would change the results and would be happy to see the author’s comment about this.
- Regarding the data distribution, would it be possible to achieve a similar result where the noise n_i are sampled from a Gaussian distribution? If so, I think this is a much more standard setting to study (signal plus Gaussian noise). I guess also here, the noise would be almost orthogonal to the signal v. If something breaks here (rather than just make the computations more tedious), then I think the result is very brittle since it requires the noise to be exactly orthogonal to the signal.
- I think the paper “Benign, Tempered, or Catastrophic: A Taxonomy of Overfitting” by Mallinar et al. should also be cited, as it gives a taxonomy of the different possible types of overfitting (which is studied also here).
- It would be nice to see an upper bound for Theorem 3.6. Currently when the margin is very small then the overfitting is either catastrophic (i.e random guess) or tempered (something between benign and catastrophic). Can the authors provide such an upper bound, or at least remark on what they think it should be? I think that it could depend on k, which depicts the noise level in the labels.
- Small note: Assumption 1, should it be k <= n/C ?


**Questions:**

- How the results would change if the network had bias terms? or the second layer would also be trained?
- Will the result still hold if the noise n_i are drawn from a Gaussian distribution? (or other symmetric distribution such as uniform over a sphere)
- What should the upper bound on the loss in Theorem 3.6 be?


**Limitations:**

I think the authors adequately addressed the limitations of their work.

---

> ### Author Rebuttal · Authors · 2023-08-08
>
> We thank the reviewer for their positive review and insightful questions which we address below.
>
> - *"How the results would change if the network had bias terms? or the second layer would also be trained?"* We believe our arguments can readily be extended to cover the case of bias terms and sketch some ideas as follows: a bias term is equivalent to modifying the data so that each training and test point contains an extra dimension with a fixed value of 1. One can think of the associated one hot vector as defining a common or bias component.  The effect of this change is increased correlation in the noise components, which has an effect of increasing $\rho$.  It is possible some way to control this effect may need to be introduced, e.g. varying the norms of the noise components or the individual learning rate of the bias terms.  We note that the argument in this paper does not immediately apply when the second layer weights are trained, and we see this as a direction to be explored in future work. We remark however that fixed outer weights is also a limitation in the two benign overfitting papers we directly compare our results to, as well as the third reference Frei, Vardi, Bartlett, and Srebo suggested by the other reviewers.
>
> - *"Will the result still hold if the noise $n_i$ are drawn from a Gaussian distribution? (or other symmetric distribution such as uniform over a sphere)"* In short yes primarily as long as the noise distribution entails that w.h.p. the noise components of the training sample are approximately orthogonal. In Lemma C.1 in the supplementary, we derive bounds on the inner product between two training points.  These bounds, along with the conditions on $\gamma$ and $\rho$, are what we use to prove our key results. This lemma would hold under e.g., Gaussian or isotropic noise distribution using a sufficient concentration result. We remark in our work that we also use the norm of each data point being exactly 1, this makes the analysis easier as each point provides an exactly equivalent push to the weights of a neuron at each iteration. However, again this is an assumption we make only to simplify the argument: some variation in the norm of the data points is certainly tolerable and furthermore a concentration result will yield this in the case of a sub-gaussian noise model for instance.
>
> - *"I think the paper “Benign, Tempered, or Catastrophic: A Taxonomy of Overfitting” by Mallinar et al. should also be cited, as it gives a taxonomy of the different possible types of overfitting"* We thank the reviewer for highlighting this interesting and important work, we will cite it in future versions of our work.
>
> - *"I guess also here, the noise would be almost orthogonal to the signal v. If something breaks here (rather than just make the computations more tedious), then I think the result is very brittle since it requires the noise to be exactly orthogonal to the signal."* As we stated in another response, the primary requirement we need is that the bounds in Lemma C.1 of the supplementary hold. These bounds can hold even in the case where the noise is not orthogonal to the signal, provided the component of the noise in the direction of the signal is not too strong.  We do note, however, that if the noise is allowed to be not orthogonal to the signal then the generalization bound in the non-benign overfitting result for small $\gamma$ may be a consequence of the data model and not the training dynamics.
>
> - *"It would be nice to see an upper bound for Theorem 3.6. Currently when the margin is very small then the overfitting is either catastrophic (i.e random guess) or tempered (something between benign and catastrophic). Can the authors provide such an upper bound, or at least remark on what they think it should be? I think that it could depend on k, which depicts the noise level in the labels."* Thanks for raising this important and challenging question. In the small $\gamma$ regime each point is primarily fitted with respect to its noise component, the activation patterns are therefore hard to track and as a result deriving a lower bound on the projection of the weights onto the signal is challenging. We believe the reviewer is correct in their conjecture as a larger $k$ means the contributions from clean and corrupt points, which are equally likely to activate a neuron at initialization, are more likely to cancel one another out, thus resulting in a small if negligible signal component being learned by the network. We leave a more in-depth study of this issue to future work.
>
> - *"Assumption 1, should it be $k \leq n/C$?"* Yes, this is a typo in the main text, thank you for highlighting!

---

> > ### Comment · Reviewer_8QyX · 2023-08-13
> > **Re:Rebuttal**
> >
> > Thank you for the response.

---

### Official Review · Reviewer_ZHgu · 2023-07-07

**Soundness:** 4 excellent
**Presentation:** 3 good
**Contribution:** 4 excellent
**Rating:** 8
**Confidence:** 4

**Summary:**

In this work a simple model of linear classification is analyzed to study the phenomenon of benign overfitting and generalization of neural networks with noisy data. A theoretical analysis of the trajectory of GD shows that it can be in three regimes (1) benign overfitting (2) non-benign overfitting and (3) no overfitting.  The analysis relies on a novel proof of the nonlinear dynamics which shows a bound on the number of non-trivial updates of GD until convergence.


**Strengths:**

1. Strong theoretical results in a very challenging setup of learning with nonlinear NNs.
2. Novel proof. Most works consider the logistic loss without a thorough analysis of the trajectory. Here the hinge loss is considered and a detailed analysis of the trajectory is provided. The key result is a bound on the number of non-trivial updates which is independent of the total number of iterations. This is highly non-trivial: a “worst-case” dynamics can result in oscillating behavior or non-convergence. However, a clever argument is presented which derives inequalities of clean and corrupt updates that together imply convergence.  Hopefully such reasoning can be helpful in more complex settings.
3. Full understanding of the dynamics. While the proof is lengthy, it provides a unique understanding of NN dynamics, their inner workings and the impact on generalization.
4. The paper is mostly clearly written.



**Weaknesses:**

1. The theoretical result holds for a bounded number of samples (i.e., $n$ cannot go to infinity).
2. Simple experiments which show the theoretical predictions could strengthen the paper.


**Questions:**

In Eq. (1) why is $\beta(i)$ needed in the definition of $\boldsymbol{x}_i$. Isn’t the distribution of $\beta(i)\boldsymbol{n}_i$ the same as the distribution of $\boldsymbol{n}_i$?

**Limitations:**

Limitations are discussed.

---

> ### Author Rebuttal · Authors · 2023-08-08
>
> We thank the reviewer for their positive review and for appreciating the novel contributions of our work. We are happy to answer the questions raised.
>
> - *"The theoretical result holds for a bounded number of samples (i.e.,  cannot go to infinity)."* As long as $d$ scales appropriately we do not require an upper bound on $n$ for our results to hold. The assumption on $d$ is used to ensure the noise components of the data points are approximately orthogonal and is standard in the literature.
>
> - *"Simple experiments which show the theoretical predictions could strengthen the paper."* We remark that preliminary experimental results are included in the supplementary materials, in particular we direct the reviewer's attention to Section G, Figures 1 and 2 and the discussion around them. For space reasons we were unable to include these in the main section of the paper.
>
> - *"In Eq. (1) why is $\beta(i)$ needed in the definition of $x_i$. Isn’t the distribution of $n_i$ the same as the distribution of $\beta(i) n_i$?"* Correct, we emphasize that this choice is for notational convenience only. Recall that we define $y_i = \beta(i)(-1)^i$, therefore defining $x_i$ in this way means that the update to each weight, which involves terms of the form $y_ix_i$, has a neater expression as $\beta(i)^2= 1$.

---

### Official Review · Reviewer_dAPs · 2023-07-15

**Soundness:** 3 good
**Presentation:** 3 good
**Contribution:** 3 good
**Rating:** 6
**Confidence:** 4

**Summary:**

This paper studies the benign overfitting (or not) of two-layer ReLU neural networks under the hinge loss. Under a new noisy (almost orthogonal) data setting without separable data setting,
- Theorem 3.1 shows that benign overfitting exists occurs the signal-level parameter $\gamma$ is small in $\Theta(1/n)$;
- catastrophic overfitting (the generalization error is bounded below by a constant) occurs when $\gamma$ is small enough at the order of $n^{-2/3}$.
- Instead, if $\gamma$ is large, no overfitting occurs, i.e., training error cannot be zero but generalizes well.


**Strengths:**

- provide solid analysis of generalization guarantees of two-layer ReLU neural networks under a new data setting
- The results on benign overfitting, catastrophic overfitting, and no overfitting heavily depend on the choice of the signal-level parameter $\gamma$


**Weaknesses:**

- The motivation of this paper is to fix the benign overfitting problem under the hinge loss. Since the analysis of previous work on the logistic loss is invalid, the authors focus on a new data generation setting and new proof framework. However, the motivation and significance of this research appear somewhat limited, particularly when considering the hinge loss. It would be beneficial to emphasize the differentiation between the results obtained with the logistic loss and the hinge loss, highlighting the theoretical advantages of employing the hinge loss. Additionally, exploring potential implicit biases related to benign overfitting and engaging in further discussions, particularly in comparison to the work conducted by Shamir (COLT2022) on the squared hinge loss, would strengthen the paper's arguments.

- The problem setting and data assumption are a bit unrealistic (also as suggested by the authors). It’s ok for me but there are some assumptions missing, e.g., in the benign overfitting part, $n - k > 2*10^6$ in Lemma B.4 is needed. I think this should be clearly mentioned in the main text.

Minior:
- The early phase in line 224 is undefined.


**Questions:**

- I understand the data setting with $d \geq n^2$ for the orthogonal condition, which is widely used in the benign overfitting literature for two-layer neural networks. But currently there exists some work that has fixed this issue:

Frei et al. Benign Overfitting in Linear Classifiers and Leaky ReLU Networks from KKT Conditions for Margin Maximization. COLT2023.

So I kindly ask the authors to think about whether their assumption can be relaxed or not.

- Since I’m an emergency reviewer, I don’t have enough time to check the proof. I high level read the proof framework, and find that line 236-237 is they key proof idea:”every training point activates one neuron after the last training update”. Could you please detail this?
The generalization performance (whether benign overfitting occurs) only depends on the signal-level parameter $\gamma$ in the data generation process. However, intuitively, the difference on generalization guarantees is based on the changes of model complexity. So how to understand this issue?

- Continuing the above question, we can find that, if $\gamma$ is small (enough) at the $O(1/n)$ order, we have the benign overfitting. However, it means that the signal level $v$ is quite small and the noise level $n_i$ is quite large. That means, the model can fit such large random noise? If $\gamma$ is large in Theorem 3.8, the model cannot fix the noise to achieve zero training loss if the noise level decreases. When checking these two results, I find that there exists some contradictions here? Maybe I’m missing something.

- The width $m$ is not required to be over-parameterization?  More discussions are needed.

---

> ### Author Rebuttal · Authors · 2023-08-08
>
> We would like to thank the reviewer for their time and comments and hope we can address each of them below. We hope that if you are satisfied with our responses you might consider increasing your score.
>
> - *"...emphasize the differentiation between ... the logistic loss and the hinge loss"* A comparison of the hyperparameter regimes required for our results versus those of related works is provided in Table 1. We also display preliminary experimental results in Figure 1, Section G of the supplementary: as discussed in the text of Section G the network trained with hinge loss appears to fit the data faster and exhibits an oscillatory behavior in the loss dynamics not present in the loss dynamics of the network trained with logistic loss. We remark that the primary goal of our work was to investigate benign overfitting away from the exponentially tailed losses setting, thereby diversifying our understanding of the phenomenon.  In revisions, we will call further attention to the results in Section G in the main text.
>
> - *"Additionally, exploring potential implicit biases related to benign overfitting and engaging in further discussions, particularly in comparison to the work conducted by Shamir (COLT2022) on the squared hinge loss, would strengthen the paper's arguments."* We remark that we do cite this interesting work but did not include a detailed discussion due to space constraints: in particular, while our work and those we use for detailed comparison in Section 1.1 consider non-linear models, the work highlighted considers linear models.  We will give more attention to this paper on future revisions.
>
> - *"...there are some assumptions missing, e.g., in the benign overfitting part..."* Thank you for highlighting this.  While that assumption is in Table 1, it was mistakenly left out of Assumption 1. This will be corrected in future versions.
>
> - *"I understand the data setting with $d \geq n^2$ for the orthogonal condition, which is widely used in the benign overfitting literature for two-layer neural networks. There exists some work that has fixed this issue: Frei et al. ..."*. To clarify, we emphasize that, like the paper highlighted and indeed all other works in this space we are aware of, we do not require strict orthogonality. Instead an approximate notion of orthogonality suffices: we require that pairwise all inner products between the noise components of data points are small. Likewise, in the paper highlighted by the reviewer, the authors also require a condition of approximate orthogonality, controlled by a parameter $p$. In Lemma B.1 of our paper, one may observe this is analogous to our variable $\rho$. Depending on the distribution from which points are drawn then as long as $d$ is sufficiently large whp an approximate orthogonality condition on the dataset will be satisfied. The $O(n^2)$ scaling of $d$ required across the broader literature (in the highlighted work see condition CL2 on page 10 for instance) arises from a union bound over all possible pairings of inputs.  We will clarify this in the revision.
>
> - *"... every training point activates one neuron after the last training update, could you please detail this?"* Lemma 3.2 and its analogues in other settings state an iteration-independent bound on the total number of updates. This implies that there must be an iteration after which no further updates are made and therefore the network must reach a steady state. This is only possible if the gradient with respect to each training point is zero. For a given training point $i$, a zero gradient is possible only in two situations with a ReLU network and hinge loss: 1) that point achieves zero hinge loss (i.e., is fully and correctly classified) or 2) that point activates no neurons (i.e., $\langle x_i, w_j \rangle \leq 0$ for all $j$).  We wish to rule out the second of these cases, so our goal is to show for each training point $i$ there exists a neuron $j$ such that $\langle x_i, w_j \rangle > 0$.
>
> - *"the difference on generalization guarantees is based on the changes of model complexity. So how to understand this issue?... The width $m$ is not required to be over-parameterization? More discussions are needed."* Large dimensionality of the input data ($d > n^2$) implies that the number of learnable parameters of the neural network is large relative to the size of the training sample. This in turn yields a relatively high model complexity even when $m$ is relatively small. Indeed, as shown in Table 1, our dependence on the number of neurons is very similar to that in prior work.
>
> - *"... if $\gamma$ is small (enough) at the $O(1/n)$ order, we have the benign overfitting. However, it means that the signal level $v$ is quite small and the noise level $n_i$ is quite large... If $\gamma$ is large in Theorem 3.8, the model cannot fix the noise to achieve zero training loss if the noise level decreases. When checking these two results, I find that there exists some contradictions here?''* In the small signal, high noise setting the network is able to fit the noise because the noise components are pairwise close to being orthogonal to one another. Therefore the projections of the network weights onto each noise component evolve approximately independently of each other during training. Indeed, with a large noise level the updates from the noise components dominate and each point is fitted by the network learning its noise component (harmful overfitting). In the large signal setting the signal dominates the training dynamics and points are fitted based on their signal component. Furthermore, fitting the clean points decreases the pre-activation of each corrupted point on neurons of the same label sign. Eventually this leads to corrupted points not activating neurons of the same label sign, the corrupted point then zeroed by the network, and an equilibrium position then reached.  This results in a nonzero loss at the end of training.

---

> > ### Comment · Reviewer_dAPs · 2023-08-11
> >
> > I appreicate the authors' feedback. It addressed most of my concern.
> >
> > About the (nearly) orthogonal condition, I fully understand the authors' concern, and the following paper (it posted after NeurIPS ddl) would be benifical to this question though the problem setting is different and one exrtra assumption is also needed.
> >
> > [1] Chistikov, D., Englert, M., & Lazic, R. (2023). Learning a Neuron by a Shallow ReLU Network: Dynamics and Implicit Bias for Correlated Inputs. arXiv preprint arXiv:2306.06479.
> >
> > Besides, I suggest the authors to think about how to improve the significance/impact of this work in the introduction, not just Table 1.
> > For example, why we focus on the hinge loss?
> >
> > Researchers outside benign overfitting will think the extension from exponential-tail type loss to hinge loss is neither significant nor novel. Though this is not true, we cannot ensure the readers are experts in benign overfitting.

---

> > > ### Author Response · Authors · 2023-08-20
> > >
> > > We thank the reviewer for highlighting this work which as highlighted was released after the NeurIPS deadline. Although the setup is quite different, in particular this paper studies the implicit bias of gradient flow with MSE loss for a clean regression task instead of noisy classification with hinge loss, we agree with the reviewer that the approach for handling correlated data is of great interest. We will certainly investigate such techniques in future work. We will happily improve our discussion of the motivation of this work in future versions. We respectfully disagree with the statement *"Researchers outside benign overfitting will think the extension from exponential-tail type loss to hinge loss is neither significant nor novel”*. Indeed, until now, benign and tempered overfitting have only been theoretically analyzed for exponentially tailed loss, despite the fact that the phenomenon is more universal than this. The training dynamics we observe and thus the proof techniques we use are quite different to prior works. Our work compliments our nascent understanding of benign overfitting and provides a path for assessing the impact of using one loss function versus another.

---

### Decision · Program_Chairs · 2023-09-21

**Decision:**

Accept (spotlight)

**Comment:**

In this work, a simple model of linear classification is analyzed to study the phenomenon of benign overfitting and the generalization of neural networks with noisy data. A theoretical analysis of the trajectory of GD shows that it can be in three regimes (1) benign overfitting (2) non-benign overfitting and (3) no overfitting. The analysis relies on a novel proof of the nonlinear dynamics which shows a bound on the number of non-trivial updates of GD until convergence.

The reviewers all agree this paper makes solid contributions toward understanding benign overfitting in challenging settings, i.e., non-linear networks. This is a clear acceptance.